# Alice in Wonderland: Simple Tasks Reveal Severe Generalization and Basic Reasoning Deficits in State-Of-the-Art Large Language Models

## Abstract

Large Language Models (LLMs) are often described as being instances of foundation models - that is, models that possess strong generalization and therefore transfer robustly across various tasks and conditions in few-show or zero-shot manner, while exhibiting scaling laws that predict generalization improvement when increasing the pre-training scale. These claims of strong generalization and advanced reasoning function enabling it rely on measurements by various standardized benchmarks where state-of-the-art (SOTA) models score high. We demonstrate here a dramatic breakdown of generalization and basic reasoning of all SOTA models which claim strong function, including advanced models like GPT-4 or Claude 3 Opus trained at the largest scales, using a simple, short common sense problem formulated in concise natural language, easily solvable by humans (AIW problem). The breakdown is dramatic as it manifests in both low average performance and strong performance fluctuations on natural problem variations that change neither problem structure nor its difficulty, while also often expressing strong overconfidence in the wrong solutions, backed up by plausible sounding explanation-like confabulations. Various standard interventions in an attempt to get the right solution, like chain-of-thought prompting, or urging the models to reconsider the wrong solutions again by multi step re-evaluation, fail. We take these observations to the scientific and technological community to stimulate re-assessment of the capabilities of current generation of LLMs as claimed by standardized benchmarks. Such re-assessment also requires common action to create standardized benchmarks that would allow proper detection of such deficits in generalization and reasoning that obviously remain undiscovered by current state-of-the-art evaluation procedures, where SOTA LLMs obtain high scores. [1].

## 1 Introduction

In the recent breakthroughs in transferable learning that were achieved in various classical domains of machine learning like visual recognition (Radford et al., 2021) or language understanding (Devlin et al., 2018; Raffel et al., 2020; Brown et al., 2020), large language models (LLMs) have played a very prominent role. The generic form and scalability of autoregressive language modelling (Brown et al., 2020) allowed to push towards training scales not achievable before with conventional supervised label-based learning. Scaling laws derived from experiments on smaller scales hinted on strong function and generalization capability appearing at larger scales (Kaplan et al., 2020; Hoffmann et al., 2022), which was then confirmed by training models at the large scales, measuring their performance on set of standardized benchmarks (MMLU, HellaSwag, ARC, MATH, GSM8k, etc) where they scored high on few- and zero-shot transfer across various tasks (Kojima et al., 2022), following accurately the predictions (Brown et al., 2020; Kaplan et al., 2020; Achiam et al., 2023; Touvron et al., 2023a;b; Jiang et al., 2023).

There were however observations made by various works that questioned the claimed strong generalization, transfer and reasoning capabilities attributed to LLMs (Mitchell, 2023). These works pointed

---

[1]Code for reproducing experiments in the paper and raw experiments data can be found at AIW repo

out various function failures that were seemingly incompatible with postulated strong capabilities as measured by standardized benchmarks (Wu et al., 2023; Golchin & Surdeanu, 2023; Li & Flanigan, 2024; Frieder et al., 2024). However, it has also been noted that observed failures can frequently be addressed through simple adjustments to the prompts or by repeated execution and evaluation using majority voting, or by requesting the model to perform self-verification (Kadavath et al., 2022; Wang et al., 2023; Zhou et al., 2023; Zhang et al., 2024; Pan et al., 2024). It remained thus unclear where those observations of failures are pointing to some fundamental deficits in core model capabilities affecting generalization and reasoning, or whether those are just symptoms of minor issues easily resolvable by simple interventions, leaving claim of strong core function as put forward by standardized benchmarks unaffected.

To shed light on current situation, we study whether the claim of SOTA LLMs possessing strong functions across various complex tasks can be put to test by using tasks that are very simple, in contrast to those employed by standardized benchmarks. We introduce a short conventional common sense problem that is formulated without any ambiguities in concise natural language and can be easily solved by humans. The problem (in following Alice in Wonderland, AIW problem) has following template: ***"Alice has $N$ brothers and she also has $M$ sisters. How many sisters does Alice's brother have?"***. Crucially, instantiating natural numbers $N, M \leq 7$ allows us to naturally introduce systematic controlled variations that do not change problem structure and difficulty and thus should not affect ability to solve it. We use then this technique of creating problem structure and difficulty preserving variations to measure models' sensitivity to problem irrelevant perturbations across multiple repetitive trials, testing models' generalization ability.

Surprisingly, when confronted with AIW problem and its structure preserving variations, all SOTA models including most advanced large-scale ones (eg GPT-4 (OpenAI, a), Claude 3 Opus (Anthropic, 2024c)) suffer severe function breakdown. This breakdown manifests (i) in average correct response rates that are unexpectedly low for such a simple problem and in (ii) strong fluctuations in correct response rates across AIW problem variations despite those being entirely irrelevant for coping with the problem. Strong fluctuations remain despite using various standard interventions to improve model function like chain-of-thought prompting. By creating further control versions of AIW problem and observing that models are successfully coping with those, we are able to rule out that observed failures might be rooted in minor low level issues of tokenization/natural language parsing, handling the basic family structure, binding attributes, or executing arithmetic operations necessary to solve the problem. Despite a specific form of simple AIW problem, we can thus conclude that observed failure has generic character. The lack of robustness revealed in all SOTA models by problem irrelevant variations of a simple problem points to severe generic deficits in generalization and reasoning.

The observed breakdown of function and generalization is in strong contrast to scores on standardized benchmarks, which contain problems of higher difficulty. Many tested models that score high on such benchmarks show correct response rates close to zero across simple AIW problem variations. Claim put forward by standardized benchmarks to properly reflect model capabilities such as generalization and reasoning cannot be upheld in face of the evident failure to detect such severe function deficits as revealed in the simple AIW problem setting. Our study highlights necessity to re-assess current capabilities of SOTA LLMs by creating novel benchmarks that properly reflect their true abilities to generalize and reason. Such benchmarks will be able to correctly spot deficits overlooked so far and thus show the path for improvement of current still unsatisfactory state.

## 2 METHODS & EXPERIMENT SETUP

### 2.1 SIMPLE COMMON SENSE REASONING PROBLEMS AND THEIR VARIATIONS

**AIW Problem.** To measure models' sensitivity to problem irrelevant variations and thus probe the zero-shot generalization, we use following problem template: ***"Alice has $N$ brothers and she also has $M$ sisters. How many sisters does Alice's brother have?"***. The problem has a simple common sense solution which assumes all sisters and brothers in the problem setting share the same parents. The correct response $C$ - number of sisters - is easily obtained by calculating $M + 1$ (Alice and her sisters), which gives the number of sisters Alice's brother has. To create problem variations, we choose to vary natural numbers $N, M \leq 7$, obtaining AIW variations 1-4 (see Suppl. Tab. 2) that all pose same problem using variations irrelevant for problem solving. We further use 3 prompt types, RESTRICTED, STANDARD and THINKING, to ensure we measure models across various

prompt formulations for making general conclusions about their behavior (see Suppl. Sec. B for full overview)

**Control AIW Light problems** To control for models struggling either with basic family relations structure handling or with executing arithmetic operations in frame of the posed AIW problem, we make various versions of AIW problem - AIW Light Family, AIW Light Arithmetic Siblings and AIW Light Arithmetic Total Girls. The AIW Light problems keep problem template close to the original, changing only the final question part such that the posed modified question tests particular operations. The variations 1-4 are created in the same way like in AIW original by varying natural numbers of brothers and sisters, while ensuring that the natural numbers for final correct answers in AIW original and AIW Light are matched across variations 1-4 (see also Suppl. Sec. B)

**AIW Light Arithmetic Siblings**. AIW Light Arithmetic Siblings has following problem template: *"Alice has $N$ brothers and she also has $M$ sisters. How many siblings does Alice have?"*. Compared to AIW original, only question part is modified. To solve the problem, summing up already given numbers of brothers and sisters is sufficient - the correct answer is $C = N + M$. This requires basic grasping of relational family structure (realizing Alice's siblings are her sisters and brothers) and selection and execution of elementary arithmetic sum operation. In contrast to AIW original, it does not require execution of set operations nor binding sex attribute to Alice to properly assign her to correct sets. Should the issues with solving AIW original be rooted in selection and execution of elementary arithmetic operations in family frame, we should see models also failing here. Again, we create variations 1-4 by varying natural numbers $N, M$, such that correct responses $C$ are matched with AIW original variations 1-4 (Suppl. Tab. 3)

**AIW Light Family**. AIW Light Family has following problem template: *"Alice has $N$ brothers and she also has $M$ sisters. How many brothers does Alice's sister have?"*. Compared to AIW original, only question part is modified. To solve the problem, reporting already given number of brothers is sufficient - the correct answer is $C = N$. This requires only basic grasping of relational family structure (understanding entity "Alice's sister", binding female attribute to Alice and realizing Alice and her sisters share same brothers). It does NOT require execution of any arithmetic or set operations, in contrast to AIW original. Should the issues with solving AIW original be rooted in handling basic family structure, we should see models also failing here. Again, we create AIW Light Family variations 1-4 by varying natural numbers $N, M$, such that correct responses $C$ are matched with AIW original variations 1-4. (Suppl. Tab. 4)

**AIW Light Arithmetic Total Girls**. AIW Light Arithmetic Total Girls has following problem template: *"Alice has $N$ brothers and she also has $M$ sisters. How many girls are there in total?"*. Compared to AIW original, only question part is modified. To solve the problem, it is necessary to bind female attribute to Alice via the pronoun "she", to assign correct female attributes to the sisters and to execute the correct arithmetic sum operation adding all the obtained girls - the correct answer is $C = M + 1$. This requires basic grasping of family structure (realizing who are the girls in the family) and selection and execution of elementary arithmetic sum operation. In contrast to AIW original, it does not require execution of set operations to properly assign Alice to sisters set. Should the issues with solving AIW original be rooted in binding correct sex attributes or counting total members of particular sex in family frame given its structure, we should see models also failing here. Again, we create variations 1-4 by varying natural numbers $N, M$, such that correct responses $C$ are matched with AIW original variations 1-4. (Suppl. Tab. 5)

## 2.2 PROMPT TYPES AND RESPONSE PARSING

**Model prompt types.** It is well known that so-called prompt engineering can heavily influence the model behavior and model response quality (Arora et al., 2022; Wei et al., 2022; White et al., 2023). To check that our observations reflect model sensitivity to controlled problem structure preserving variations in same manner independent of particular prompt type, we employed 3 various prompt types to provide model's input: STANDARD (prompt with instruction to format final answer output as a natural number), THINKING (prompt that in addition encourages thinking in spirit of CoT) and RESTRICTED (prompt with instruction to output nothing else but final answer as a natural number). THINKING v2 prompt type is a minor variation of THINKING type that just adds "step by step" after already existing "think carefully" phrasing (control experiments show that THINKING and THINKING v2 are equivalent in terms of observed performance, so we use both

interchangeably). STANDARD and THINKING prompt types allow models to generate any text output before delivering the final answer, while RESTRICTED is used as control with restricted output to measure model behavior when the only output allowed is the final answer (Suppl. Tab. 2)

Furthermore, we make use of other prompt types (see Suppl. Sec.B for overview) to demonstrate various important properties and the different success or failure modes of the model behavior for the AIW problem. In those prompts, we re-use the main problem formulation as introduced in Sec. 2.1, while adding various modifications. This allows us for instance to observe confabulations that contain clearly broken statements with reasoning-like convincing sound backing up wrong final answers or responses showing model overconfidence.

**Parsing model responses.** To perform evaluations of model performance, it is necessary to parse and extract the final answer from the responses provided by the models. Each input to the model is combination of a AIW problem variation, followed by one of prompt types as described before. To keep the parsing procedure simple, we add to each problem prompt following output format instruction: *"provide the final answer in following form: "### Answer: ""*. We observed that all models we have chosen to test were able to follow such an instruction, providing a response that could be easily parsed. We also ran control experiments without such formatting instruction in the problem formulation, ensuring that behavior does not depend on it.

### 2.3 SELECTING MODELS FOR EVALUATION AND CONDUCTING EXPERIMENTS

We are interested in testing current state-of-the-art models that claim strong function, especially in generalization and reasoning, backed up by high scores shown on standardized benchmarks that are assumed to measure generalization and reasoning capabilities to solve problems. We therefore select models widely known and used in the ML community that also appear in the top rankings of the popular LLM leaderboards, like openLLM leaderboard by HuggingFace or ELO leaderboard by LMsys. We provide the overview of the selected models in Suppl. Tab. 1 and list in Suppl. Tab. 7 the corresponding standardized benchmarks where they obtain strong scores.

We expose selected SOTA LLMs, including most advanced models at largest scales (see Suppl. Tab. 1) to AIW problem variations 1-4 (Suppl. Tab. 2) and AIW Light control problems (Suppl. Tab. 3, 4, 5), using different prompt types as described above. For each combination of model, AIW problem variation and prompt type, at least 30 trials are collected to compute correct response rates, Suppl. Fig. 21. For details on correct response rates estimation procedure, see Suppl. Sec. A.2

We use hosting platforms that offer API access or local deployment via vLLM (Kwon et al., 2024) for testing the models, and automatize the procedure by scripting the routines necessary to prompt models with our prompts set. The routines are simple and can be used by anybody with access to the APIs (we used liteLLM and TogetherAI for our experiments) or to locally hosted models to reproduce and verify our results. We protocol all the data from interactions with the models to enable community checking. We release all the collected raw response data, correct response rates estimates and routines used to conduct experiments as open-source for reproducibility and further usage.

## 3 RESULTS

### 3.1 HUMPTY DUMPTY SAT ON A WALL: BREAKDOWN OF SOTA LLMS ON THE SIMPLE AIW PROBLEM

**AIW reveals severe generalization and reasoning deficits in SOTA LLMs**. Following the procedures described in Sec 2, we expose the selected models that claim strong function and reasoning capabilities (Suppl. Tab. 1) and measure their correct response rate performance across and for each AIW variations 1-4 using various prompt types, executing $> 30$ trials for each combination (see also Suppl. Tab. 2 and Suppl. Fig. 21). The results suggest that confronted with the AIW problem, models suffer a severe function breakdown. This breakdown has two main manifestations:

**1. Low correct response rates.** Despite evident problem's simplicity, many models are not able to deliver a single correct response, and the majority stay well below correct response rate of $p = 0.2$. We summarize the main results in the Fig. 1. The only major exceptions from the observation of very low correct response rates are the largest scale closed models GPT-4 and Claude 3 Opus. These two

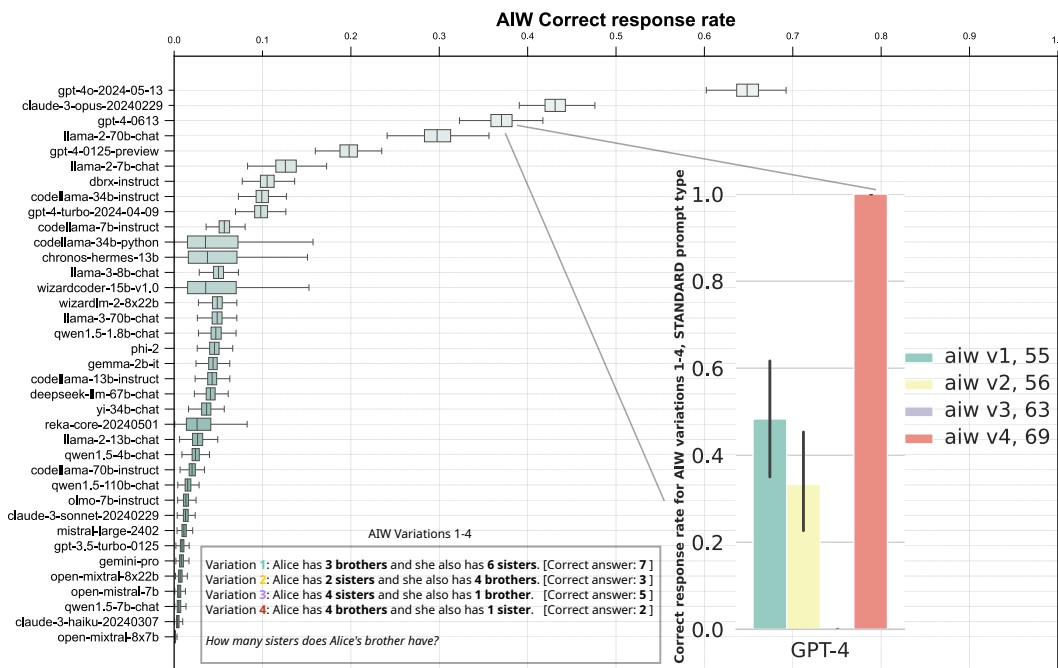

Figure 1: Collapse of SOTA LLMs on AIW problem. (**main**) Models with non-zero AIW correct response rate, average over STANDARD, THINKING, RESTRICTED prompt types and AIW variations 1-4. Omitted models score 0. (**inlay**) Strong fluctuations on AIW variations 1-4, despite problem structure and difficulty remaining entirely unchanged across variations.

models at largest scales obtain correct response rates well above $p = 0.3$, leaving the remaining large and smaller scales open-weights (e.g., Mistral-7B, Mixtral, Qwen, Command R+, and Dbrx Instruct) and closed-weights models (e.g., Gemini Pro, Mistral Large) far behind. Remarkably, many models that claim high scores on standardized benchmarks, show very low correct response rates close to 0, eg. Llama-3-8B, Mixtral-8x22B, Qwen1.5-110B, or exhibit even complete breakdown on AIW with correct response rate of zero across all variations, eg Command R+ or Qwen1.5-72B (Suppl. Tab. 7)

The results presented in the Fig. 1 show estimates for correct response rates averaged across RESTRICTED, STANDARD and THINKING prompt types (Suppl. Tab. 2, prompt IDs provided for reproducibility; Suppl. Fig. 8 with models scoring 0). RESTRICTED prompt type was used as further control that forces models into short outputs, restricting the compute for providing a solution and thus serving as low baseline for the performance (see Suppl. Sec. C and Suppl. Fig. 11). Among the 4 models that are able to cross $p = 0.3$, two clear winners are the GPT-4o ($p = 0.649$) and Claude 3 Opus ($p = 0.431$). The only open-weights model in this set of better performers is the rather older Llama-2 70B Chat ($p = 0.3$). For these better performers, when inspecting the responses with correct final answers, we see also correct reasoning backing up the final answers. For the poor performing models with low correct response rates, by inspecting those rare responses with correct answers we also in some cases still can see correct reasoning. In the poor performers, among the responses with a correct final answer we see however often responses where final answer, after careful inspection, turns out to be an accident of executing entirely wrong reasoning with various mistakes leading coincidentally to the final output number corresponding to the right answer. Such responses are encountered in models with low correct performance rates ($p < 0.3$) (see Suppl. Sec. D for response examples), and we correct via manual inspection the status of correct response for such cases.

**2. Strong performance fluctuations across irrelevant AIW problem variations.** Importantly, we also observe strong fluctuation of correct response rates across AIW variations 1-4 as introduced in Sec. 2. Such fluctuations strongly affect better performers with higher average correct response rates like GPT-4/4o and Claude 3 Opus. As shown in the Fig. 1 (inlay) for the STANDARD and Fig. 2 for the THINKING prompt type, the correct response rates can fluctuate between being close to 1 to being close to 0, depending on AIW variation. Remarkable is that such fluctuations appear despite AIW variations being all instances of the very same simple problem, as changes in numbers

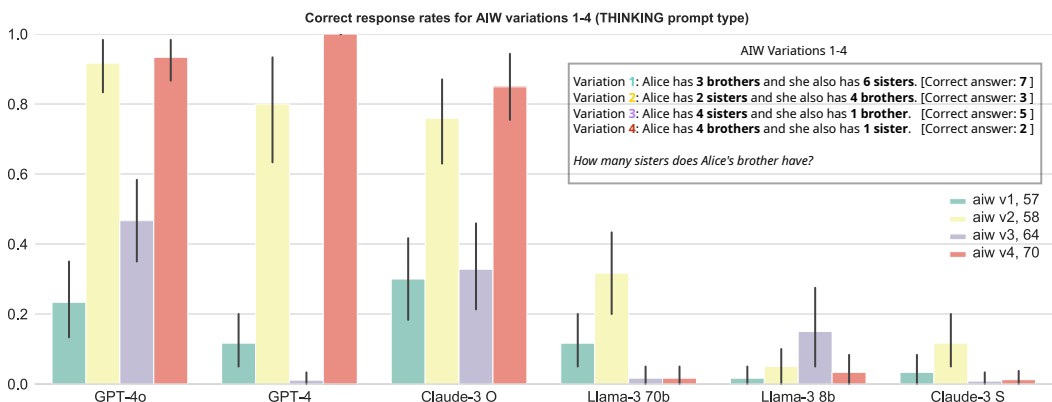

Figure 2: Strong fluctuations across AIW problem variations, THINKING prompt. Also for better performers, eg GPT-4o, GPT-4 and Claude Opus 3, correct response rates vary strongly from close to 1 to close to 0, despite AIW variations being irrelevant for problem structure (a color per each variation 1-4). This shows clear lack of model robustness, revealing generalization and basic reasoning deficits.

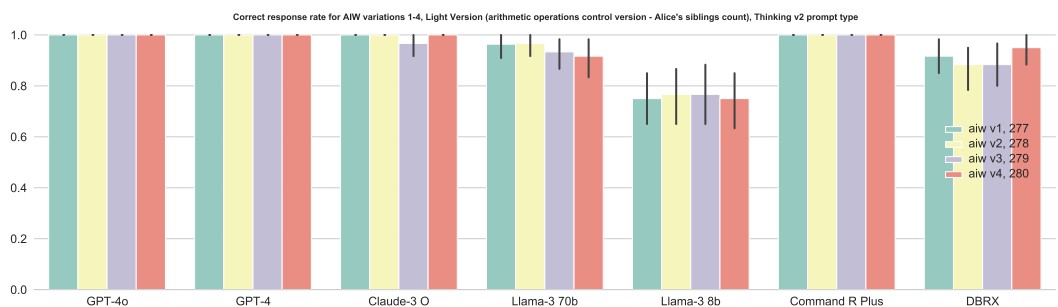

Figure 3: Correct response rates across AIW Light Arithmetic Siblings control problem variations 1-4 (THINKING v2 prompt type). Strong performance is observed across problem variations (a color per each variation 1-4; prompt IDs in the legend, Suppl. Tab. 3). Models that entirely collapse on AIW, like Command R Plus and Dbrx Instruct, are clearly able to solve this version, with correct response rates going up to 1 or close to 1 across all problem variations. This shows that executing arithmetic operations or handling basic family setting is not an issue for the tested models.

used across AIW variations do not change either the problem structure or its difficulty at all. This lack of robustness on such a simple problem hints on severe deficits in generalization. The strong fluctuations across variations appear independent of employed prompt types (Suppl. Fig. 9), while correct response rate averaged across all variations also varies across prompt types, showing in addition expected prompt type dependency (Suppl. Fig. 11, 12)

### 3.1.1 CONTROL EXPERIMENTS USING AIW LIGHT PROBLEMS

In all following experiments, for each AIW variation, 60 trials were executed to estimate correct response rate and its variance.

**AIW Light Arithmetic Siblings**. We show tested models' performance in Fig. 3. While all tested models clearly have struggled with AIW original (Fig. 1, Suppl. Fig. 8), we observe them successfully solving AIW Light Arithmetic Siblings. Correct response rates go high up close to 1 for most tested models across all variations 1-4. This is also the case for the models that show very low correct response rates close to 0 or 0 on AIW original, like Command R+ or Dbrx Instruct (Suppl. Fig. 8, Suppl. Tab. 7). Strong fluctuations we observe across variations on AIW original (Fig. 1, 2) also disappear. This clearly demonstrates that models neither struggle with basic grasping of relational family structure - realizing Alice's siblings are her sisters and brothers, nor with selection and execution of elementary arithmetic sum operation.

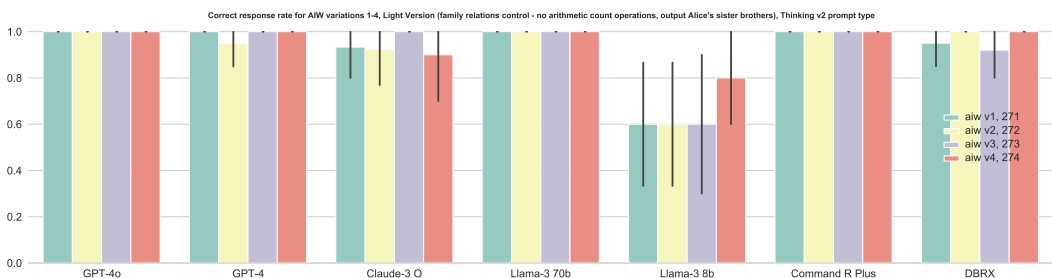

Figure 4: Correct response rates across AIW Light Family control problem variations 1-4 (THINKING v2 prompt type). Strong performance is observed across problem variations (a color per each variation 1-4). Models that entirely collapse on AIW, like Command R Plus and Dbrx Instruct, are clearly able to solve this version, with correct response rates going up to 1 or close to 1 across all problem variations. This shows that handling basic family relations is not an issue for the tested models.

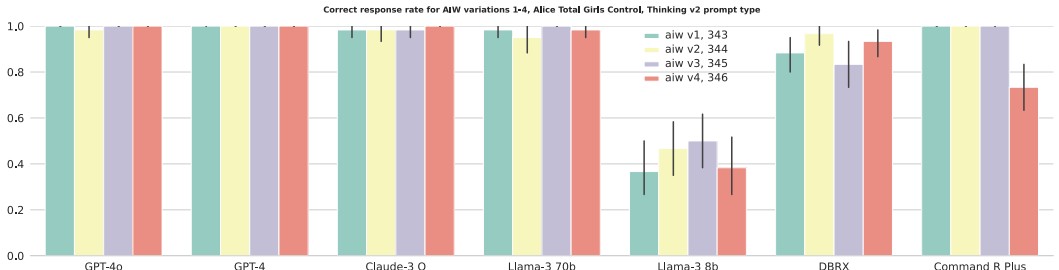

Figure 5: Correct response rates across AIW Light Arithmetic Total Girls control problem variations 1-4 (THINKING v2 prompt type). Strong performance is observed across problem variations (a color per each variation 1-4; prompt IDs in the legend, Suppl. Tab. 5). Models that entirely collapse on AIW, like Command R Plus and Dbrx Instruct, are clearly able to solve this version, with correct response rates going up to 1 or close to 1 across all problem variations. This rules out that either binding of female attributes to Alice and the sisters entities or selection and execution of arithmetic operations necessary to count total females is an issue for the tested models.

**AIW Light Family**. We show tested models' performance in Fig. 4. Also here we observe all the tested models that are struggling with AIW original successfully solving AIW Light Family. Correct response rates go high up close to 1 for most tested models across all variations 1-4. This is also the case for the models that show very low correct response rates close to 0 or 0 on AIW original. like Command R+ or Dbrx Instruct (Suppl. Fig. 8 & Tab. 7). Also strong fluctuations that we observe across variations on AIW original (Fig. 1, 2) disappear. This clearly demonstrates that models handle well basic grasping of relational family structure - understanding entity "Alice's sister", binding female attribute to Alice and realizing Alice and her sisters share same brothers.

**AIW Light Arithmetic Total Girls**. We show tested models' performance in Fig. 5. Again, we observe also here strong performance for all tested models that clearly have struggled with AIW original. Correct response rates go high up close to 1 for most tested models across all variations 1-4. This is also the case for the models that show very low correct response rates close to 0 or 0 on AIW original. like Command R+ or Dbrx Instruct. Also strong fluctuations that we observe across variations on AIW original (Fig. 1, 2) are gone. This clearly demonstrates that models successfully cope with binding female attribute to entity of Alice, handle assignment of correct female attributes to the sisters and select and execute the correct arithmetic sum operation adding all the girls together.

From these control experiments, we are thus able to obtain strong evidence that all tested models do not suffer from low-level issues with tokenization and natural language or natural numbers parsing and can handle well basic family relations structure and selection and execution of elementary arithmetic operations necessary to solve AIW problem. This further strengthen the hypothesis that observed failures and strong fluctuations in all tested SOTA models on AIW problem (Fig. 1, 2) are rooted in problem unspecific, generic deficits in generalization and basic reasoning.

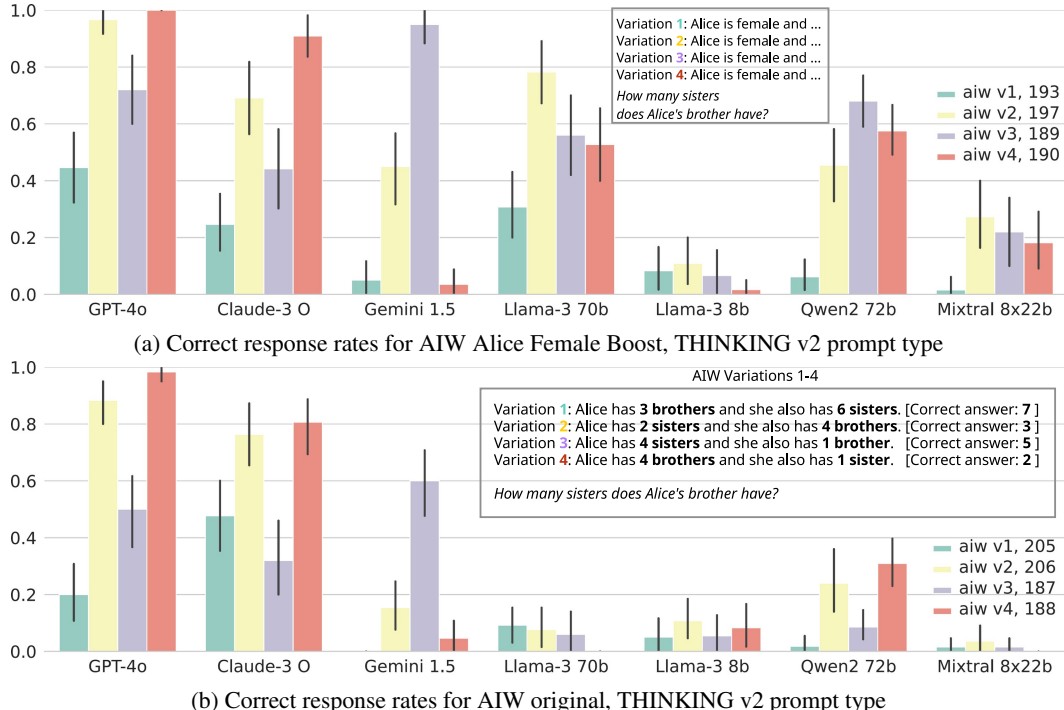

(a) Correct response rates for AIW Alice Female Boost, THINKING v2 prompt type

(b) Correct response rates for AIW original, THINKING v2 prompt type

Figure 6: Altering model performance by fully redundant information. Adding fully redundant information "Alice is female" leads to increase of average correct response rates in **(a)** compared to AIW original **(b)** (see also Suppl. Fig. 10). For some models, eg Llama 3 70B or Qwen 2 72B, this boost via redundant info is especially pronounced and happens across all variations, resulting in clear overall improvement from **(b)** to **(a)**. Strong fluctuations across variations 1-4 persist. This again shows lack of model robustness, hinting on severe generalization and basic reasoning deficits.

## 3.2 CURIOUSER AND CURIOSER: FURTHER PROPERTIES OF OBSERVED BREAKDOWN

**Boosting by reduntant information and persisting fluctuations: Alice female power boost**. One clear signature of generalisation and reasoning breakdown are the strong fluctuations we observe across AIW problem variations 1-4 that differ only in instantiated numbers (Fig. 2). We investigate a further AIW problem version by adding *"Alice is female"* to the original AIW problem formulation (see Suppl. Tab. 6 and Suppl. Sec. C.1). This is a fully redundant information, as Alice's gender is already unambiguously specified by the "she" pronoun used in original AIW problem. As evident from Fig. 6 and Suppl. Fig. 10, the average correct response rates are increasing, despite the provided "female boost" information being entirely redundant and not revealing anything new necessary for AIW problem solution. Altering performance by fully redundant information that should not affect problem solving reveals again deficits in generalization and basic reasoning. While average correct response rates increase, the strong fluctuations across AIW variations 1-4 remain (Fig. 6a). For instance, GPT-4o has on AIW variations 2,4 correct response rate close to 1, while dropping heavily for AIW variations 1,3, showing same lack of robustness despite the average boost.

**Standardized benchmarks failure**. We observe failure of standardized reasoning benchmarks to properly reflect generalization and basic reasoning skills of SOTA LLMs by noting significant disparity between the model's performance on the AIW problem and the scores on conventional standardized benchmarks. All of the tested models report high scores on various standardized benchmarks that claim to test problem solving via reasoning, e.g. MMLU, ARC, Hellaswag. Our observations of SOTA models breaking down on the simple AIW problem hint that the benchmarks do not reflect deficits in generalization and basic reasoning of those models properly. We visualize this failure by plotting scores tested models obtain on wide-spread and accepted standardized benchmarks like MMLU versus the performance we observe on our proposed AIW problem. As strikingly evident from Fig. 7, there is a strong mismatch between high scores on MMLU reported by the models and

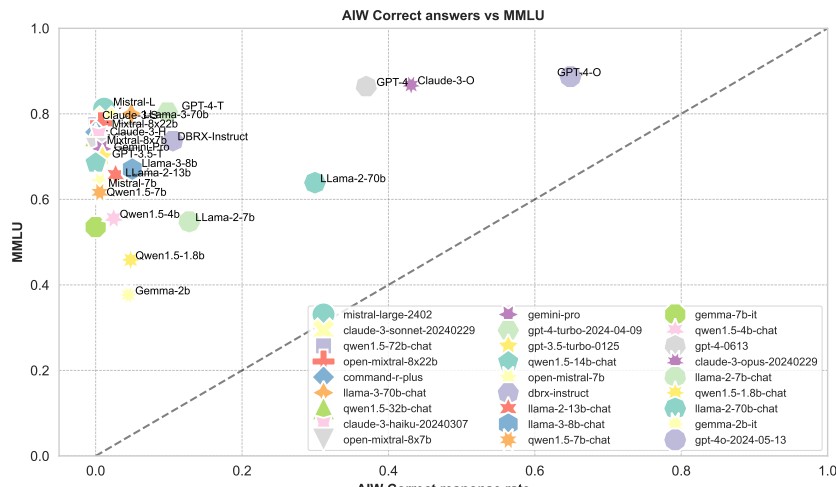

Figure 7: Failure of standardized benchmark MMLU to properly reflect and compare model basic reasoning capabilities as shown by strong discrepancy between AIW correct response rate vs MMLU average score. Many models, eg. Command R+, score 0 on AIW, but have high MMLU score.

the correct response rates they obtain on AIW. This mismatch and lack of differentiation makes it impossible for a given model to predict from its score on MMLU whether it will suffer breakdown on a simple problem like AIW, making the score unreliable for measuring core capabilities. Also model ranking fails, as models claiming higher scores can be strongly outperformed by models with lower scores when looking at their correct response rates on simple AIW problem. For instance, Llama-2-70B with lower MMLU score clearly outperforms on AIW problem models (eg Mistral-Large, Command R+, Dbrx Instruct) that are crowded in high MMLU - low AIW score region (left upper part of Fig. 7). For similar evidence on other standardized benchmarks, see Suppl. Sec. C.3

**Further relevant observations.** *1. Dominance of wrong responses* We measure distribution of natural numbers responses on output, showing that for AIW variations with low correct response rate, peaks are on wrong answers, excluding majority voting methods as a fix. (Suppl. Sec. C.2) *2. Confabulations and overconfident tone* We observe that wrong responses are often accompanied by persuasive explanation-like confabulations and overconfident tone about correctness of the wrong solutions provided by the models, which can further mislead model users (Suppl. Sec. E) *3. Inability to revise wrong responses.* Models show failure to properly detect mistakes and to revise wrong solutions when encouraged to do so in experiments with multi-turn AIW problem interaction and self-verification. (Suppl. Sec. F). *4. Reformulation of AIW as relational SQL database problem.* We make use of relational logic underlying the AIW problem structure and prompt models to reformulate AIW into a correct relational SQL database format, to test their ability to extract formal problem structure. We observe that smaller scale models, and also some larger scale ones, consistently fail to generate a correct relational SQL form. Some models that are able to do so more frequently, e.g., Mistral/Mixtral, still fail to provide correct final answer most of the time (Suppl. Sec. G)

## 4 RELATED WORK & LIMITATIONS

**Measuring LLMs capabilities.** Since the seminal breakthroughs in language modelling (Devlin et al., 2018; Raffel et al., 2020; Brown et al., 2020), measuring LLM capabilities became indispensable for evaluations and model comparison. To measure how well a language model performs on reasoning, there exists a plethora of different standardized reasoning benchmarks. These benchmarks can be roughly divided into categories by what exact reasoning capability we want to test such as ARC (Clark et al., 2018), PIQA (Bisk et al., 2020), GSM8K (Cobbe et al., 2021), HellaSwag (Zellers et al., 2019), MMLU (Hendrycks et al., 2020) or WinoGrande (Sakaguchi et al., 2019). Multiple works aim on improving reasoning performance of LLMs as measured by those standardized benchmarks in various ways (Wei et al., 2022; Yao et al., 2024; Zhou et al., 2022; Wang et al., 2022; Pfau et al., 2024).

**Stress-testing LLMs' weaknesses.** Paralleling impressive progress shown by LLM research, cautious voices have been raising concern about discrepancy between claimed capabilities as measured by standardized benchmarks and true LLM reasoning skills by presenting carefully selected evidence for model failures (Mitchell, 2023). In response, the research community has been undertaking attempts to create more challenging benchmarks like HELM (Liang et al., 2023) or BIG-bench (Srivastava et al., 2023). These benchmarks also aimed at properly testing generalization capabilities beyond memorization, in line with recent works that pointed out high test dataset contamination due to large-scale pre-training on web-scale data (Golchin & Surdeanu, 2023; Li & Flanigan, 2024).

Similar in spirit to our work, multiple studies (Wu et al., 2023; Dziri et al., 2024; Lewis & Mitchell, 2024; Berglund et al., 2023; Moskvichev et al., 2023; Huang et al., 2023) have shown breakdowns of language models reasoning capabilities in different scenarios and also lack of robustness to variation of problem formulation (Zong et al., 2024; Zheng et al., 2024). Other works were looking into particular reasoning failures like deficits in causality inference (Jin et al., 2023b;a). These works operate often with formalized, rather complex problems that does not have simple common sense character. Here we show breakdown on a common sense problem with very simple structure using natural, controlled variations that keep problem structure and difficulty unchanged, which emphasizes a generic deficiency in generalization and basic reasoning about problem structure. A key limitation of our current approach is the lack of sufficient diversity in AIW problem variations. This can be addressed in future work by systematic procedural instance generation for broader response evaluation.

## 5 DISCUSSION & CONCLUSION

In our work, using a very simple AIW problem (Sec. 2) that can be easily solved by adults and arguably even children, we observe a striking breakdown of SOTA LLMs performance when confronted with the AIW problem and its variations (Suppl. Tab. 2). The breakdown is manifested in (i) Low average correct response rates (Fig. 1) and (ii) Strong performance fluctuation across structure and difficulty preserving natural variations of the same problem, which hints at fundamental issues with the generalization capability of the models (Fig. 2). The observed breakdown is in dramatic contrast with claims about strong core functions of SOTA LLMs. Specifically, the claim of strong reasoning cannot hold, as any system claiming even basic reasoning should be able to obtain 100% correct response rates on problems as simple as AIW. The evidence also falsifies the claim of strong zero-shot generalization - as in such a simple problem, strong performance fluctuations across variations that keep problem structure and difficulty unchanged reveal severe generalization deficits in all tested SOTA LLMs. By executing control experiments, we provide evidence that the observed failures are not specific to the problem type we study and thus hint on generic deficits in generalization and basic reasoning (Sec. 3.1.1). Our study also clearly points to failure of standardized benchmarks to properly measure core model functionality such as generalization or reasoning (Suppl. Sec. C.3, Fig. 7, Tab. 7). Standardized benchmarks assigning high scores to SOTA LLMs fail to reveal severe model weaknesses made evident by breakdown on simple AIW problem. It has to be noted that despite observed breakdowns with low average correct response rates, the reasoning is not entirely absent and better performer larger scale models like GPT-4 or Claude 3 Opus do show examples of fully correct reasoning (see Suppl. Fig. 25, 26). As our results show, this reasoning capability is however fragile and cannot be accessed robustly, even in such a simple scenario as posed by AIW problem variations.

The observations urge re-assessment of the claimed capabilities of current generation of LLMs, with evidence suggesting that current SOTA LLMs are not capable of strong generalization and robust reasoning, and enabling those is still subject of basic research. Such re-assessment also requires common action to create standardized benchmarks that would allow proper detection of such generalization and basic reasoning deficits as observed in our study that obviously manage to remain undiscovered by current state-of-the-art evaluation procedures and benchmarks. Variations built into problem templates can serve as technique to create new benchmarks that are, in contrast to current common benchmarks, no longer static and can serve as better measurement tool for properly testing model generalization and reasoning. New benchmarks should follow Karl Popper's principle of falsifiability (Popper, 1934), attempting everything to break model's function, highlighting its deficits, and thus showing possible directions for model improvement, which is the way of scientific method, also offering protection from overblown claims about models' core functions.

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

# Supplementary.

## A ADDITIONAL DETAILS ON PERFORMED EXPERIMENTS

Here we give further details on the procedures around the executed experiments.

### A.1 MODELS SELECTED FOR EXPERIMENTS

To provide overview over origin of core tested models used for the AIW experiments, we list those in Suppl. Tab. 1. All tested models use same default inference hyperparameters, $T = 0.1$, *top-p* $= 1.0$ (we executed control experiments to check that various settings do not change the main pattern in the observed behavior). The output was limited to $2048$ tokens, and as evident from Suppl. Fig. 22, most observed responses stayed well below this limit.

Table 1: Names, origin and versioning of core test models used in the experiments.

| Name | Origin | Released | Open Weights | Sources |
|---|---|---|---|---|
| GPT-4o-2024-05-13 | OpenAI | 13.05.2024 | No | (Achiam et al., 2023; OpenAI, 2024a;b) |
| GPT-4-turbo-2024-04-09 | OpenAI | 09.04.2024 | No | (Achiam et al., 2023; OpenAI, a) |
| GPT-4-0125-preview | OpenAI | 25.01.2024 | No | (Achiam et al., 2023; OpenAI, a) |
| GPT-4-0613 | OpenAI | 13.06.2023 | No | (Achiam et al., 2023; OpenAI, a) |
| GPT-3.5-turbo-0125 | OpenAI | 24.01.2024 | No | (OpenAI, 2022; b;c) |
| Claude-3-5-sonnet-20240620 | Anthropic | 21.06.2024 | No | (Anthropic, 2024b) |
| Claude-3-opus-20240229 | Anthropic | 04.03.2024 | No | (Anthropic, 2024a;c) |
| Claude-3-sonnet-20240229 | Anthropic | 04.03.2024 | No | (Anthropic, 2024a;c) |
| Claude-3-haiku-20240307 | Anthropic | 04.03.2024 | No | (Anthropic, 2024a;c) |
| Gemini 1.0 Pro | Google | 06.12.2023 | No | (Pichai & Hassabis, 2023; Team et al., 2023) |
| Gemini 1.5 Pro | Google | 16.02.2024 | No | (Pichai & Hassabis, 2024; Reid et al., 2024) |
| gemma-7b-it | Google | 05.04.2024 (v1.1) | Yes | (Google, 2024a;b) |
| gemma-2b-it | Google | 05.04.2024 (v1.1) | Yes | (Google, 2024a;b) |
| Mistral-large-2402 | Mistral AI | 26.02.2024 | No | (Mistral-AI-Team, 2024c;d) |
| Mistral-medium-2312 | Mistral AI | 23.12.2023 | No | (Mistral-AI-Team, 2024c;d) |
| Mistral-small-2402 | Mistral AI | 26.02.2024 | No | (Mistral-AI-Team, 2024c;d) |
| open-mixtral-8x22b-instruct-v0.1 | Mistral AI | 17.04.2024 | Yes | (Mistral-AI-Team, 2024c;a) |
| open-mixtral-8x7b-instruct-v0.1 | Mistral AI | 11.12.2023 | Yes | (Mistral-AI-Team, 2024c;b) |
| open-mistral-7b-instruct-v0.2 | Mistral AI | 11.12.2023 | Yes | (Jiang et al., 2023; Mistral-AI-Team, 2024c; 2023) |
| Command R+ | Cohere | 04.04.2024 | Yes | (Cohere, 2024a;b) |
| Dbrx Instruct | Mosaic | 27.03.2024 | Yes | (Mosaic) |
| Llama 2 70B Chat | Meta | 18.07.2023 | Yes | (Meta, 2023; Touvron et al., 2023b) |
| Llama 2 13B Chat | Meta | 18.07.2023 | Yes | (Meta, 2023; Touvron et al., 2023b) |
| Llama 2 7B Chat | Meta | 18.07.2023 | Yes | (Meta, 2023; Touvron et al., 2023b) |
| Llama 3 70B Chat | Meta | 18.04.2024 | Yes | (Meta, 2024a;b) |
| Llama 3 8B Chat | Meta | 18.04.2024 | Yes | (Meta, 2024a;b) |
| Qwen 1.5 1.8B - 72B Chat | Alibaba | 04.02.2024 | Yes | (Bai et al., 2023; Alibaba, 2024a) |
| Qwen 2 72B Instruct | Alibaba | 07.06.2024 | Yes | (Alibaba, 2024b) |

## A.2 Evaluating Model Responses

The formatting instruction makes it possible to extract for each prompting trial whether a model has provided a correct answer to the AIW problem posed in the input. We can interpret then any number $n$ of collected responses as executing $n$ trials given a particular prompt for a given model ($n$ - number of Bernoulli trials), observing in each $i-$th trial a Bernoulli variable $X_i = \{0, 1\}$. We interpret the number of correct responses $X = \sum_i X_i$ as random variable following a Beta-Binomial distribution with unknown probability $p$ of correct response that we also treat as random variable that comes from a Beta distribution, i.e. $p \sim Beta(\alpha, \beta)$, where $\alpha$ and $\beta$ are parameters of the Beta distribution. To obtain plots showing correct response ratios, we would like to estimate Beta distribution underlying $p$, and for that, we first estimate the mean of $p$ and its variance from the collected observations. To estimate $\hat{p}$, we use the formula for estimating the mean of $p$ for a binomial distribution: $\hat{p} = X/n$ (i.e. as a proportion of successes). We can report the estimate $\hat{p}$ as the estimate of the correct response rate of a given model and also, compare the correct response rates of various tested models. Moreover, we can estimate the variance of the probability of a correct response by using the following formula:

$$\text{var}\left(\frac{1}{n}\sum_{i=1}^n X_i\right) = \frac{1}{n^2}\sum_{i=1}^n \text{var}(X_i) = \frac{n\text{var}(X_i)}{n^2} = \frac{\text{var}(X_i)}{n} = \frac{p(1-p)}{n} \tag{1}$$

The estimates of the variance and the standard deviation of $p$ can be thus obtained by using $\hat{p}$ as $\frac{\hat{p}(1-\hat{p})}{n}$ and $\sqrt{\frac{\hat{p}(1-\hat{p})}{n}}$ respectively. Using the estimated variance and mean of $p$, we can use the following relations for the variance: $\left(\sigma^2 = \frac{n\alpha\beta(\alpha+\beta+n)}{(\alpha+\beta)^2(\alpha+\beta+1)}\right)$ and the mean $\left(\mu = \frac{\alpha}{\alpha+\beta}\right)$ in order to obtain $\alpha$ and $\beta$ parameters for the Beta distribution. To simulate data for the plots, we draw $N$ random samples corresponding to correct and incorrect responses using the estimated distribution of $p$ and obtain the plots showing performance on the task for various models of interest as a full distribution of the respective $p$.

## B  Prompt Types and Variations

For testing the model dependence on input prompt type as well as robustness against problem variations when solving AIW and AIW Light problems, we used three main prompt types - STANDARD (original prompt with answer formatting instructions), THINKING (prompt that encourages thinking with answer formatting instructions) and RESTRICTED (prompt that instructs model to output only formatted answer and nothing else). THINKING v2 prompt type is a minor variation of THINKING type that just adds "step by step" after already existing "think carefully" phrasing (control experiments show that THINKING and THINKING v2 are equivalent in terms of observed performance, so we use both interchangeably, Suppl. Fig. 12b).

For testing the models' robustness to problem perturbations, we try different variations of main AIW problem (AIW Variations 1-4, see Sec. 2, Suppl. Tab. 2), where we keep the same problem structure while varying numbers of brothers and sisters and their mentioning order within the sentence. Those variations are made intentionally in such a way that they do not affect problem structure or its difficulty and thus should not affect how models cope with the problem.

We employ a further AIW version - AIW Light - as control to test whether models are able to deal with various aspects of original AIW problem, eg handling the specific relational family structure frame, or executing elementary arithmetic operations necessary to solve AIW problem. See Sec. 2 for details on the AIW Light design.

See Suppl. Tab. 2 (for AIW problem) and Suppl. Tab. 3, 4, 5 (for AIW Light problems) for examples with full prompt versions for each presented problem and its variations[2].

---

[2]All prompts and their IDs available at `https://anonymous.4open.science/r/AITW_anonymous-69A6/prompts/prompts.json`

Table 2: AIW main variations, prompt types and correct answers overview.

| Var. | Prompt | Type/Answer | ID |
|---|---|---|---|
| 1 | Alice has 3 brothers and she also has 6 sisters. How many sisters does Alice's brother have? Solve this problem and provide the final answer in following form: "### Answer: ". | STANDARD / 7 | 55 |
| 1 | Alice has 3 brothers and she also has 6 sisters. How many sisters does Alice's brother have? Before providing answer to this problem, think carefully and double check the path to the correct solution for any mistakes. Provide then the final answer in following form: "### Answer: ". | THINKING / 7 | 57 |
| 1 | Alice has 3 brothers and she also has 6 sisters. How many sisters does Alice's brother have? To answer the question, DO NOT OUTPUT ANY TEXT EXCEPT following format that contains final answer: "### Answer: ". | RESTRICTED / 7 | 53 |
| 2 | Alice has 2 sisters and she also has 4 brothers. How many sisters does Alice's brother have? Solve this problem and provide the final answer in following form: "### Answer: ". | STANDARD / 3 | 56 |
| 2 | Alice has 2 sisters and she also has 4 brothers. How many sisters does Alice's brother have? Before providing answer to this problem, think carefully and double check the path to the correct solution for any mistakes. Provide then the final answer in following form: "### Answer: ". | THINKING / 3 | 58 |
| 2 | Alice has 2 sisters and she also has 4 brothers. How many sisters does Alice's brother have? To answer the question, DO NOT OUTPUT ANY TEXT EXCEPT following format that contains final answer: "### Answer: ". | RESTRICTED / 3 | 54 |
| 3 | Alice has 4 sisters and she also has 1 brother. How many sisters does Alice's brother have? Solve this problem and provide the final answer in following form: "### Answer: ". | STANDARD / 5 | 63 |
| 3 | Alice has 4 sisters and she also has 1 brother. How many sisters does Alice's brother have? Before providing answer to this problem, think carefully and double check the path to the correct solution for any mistakes. Provide then the final answer in following form: "### Answer: ". | THINKING / 5 | 64 |
| 3 | Alice has 4 sisters and she also has 1 brother. How many sisters does Alice's brother have? To answer the question, DO NOT OUTPUT ANY TEXT EXCEPT following format that contains final answer: "### Answer: ". | RESTRICTED / 5 | 65 |
| 4 | Alice has 4 brothers and she also has 1 sister. How many sisters does Alice's brother have? Solve this problem and provide the final answer in following form: "### Answer: ". | STANDARD / 2 | 69 |
| 4 | Alice has 4 brothers and she also has 1 sister. How many sisters does Alice's brother have? Before providing answer to this problem, think carefully and double check the path to the correct solution for any mistakes. Provide then the final answer in following form: "### Answer: ". | THINKING / 2 | 70 |
| 4 | Alice has 4 brothers and she also has 1 sister. How many sisters does Alice's brother have? To answer the question, DO NOT OUTPUT ANY TEXT EXCEPT following format that contains final answer: "### Answer: ". | RESTRICTED / 2 | 71 |
| 4 | Alice has 4 brothers and she also has 1 sister. How many sisters does Alice's brother have? Solve the problem by taking care not to make any mistakes. Express your level of confidence in the provided solution as precisely as possible. | CONFIDENCE / 2 | 11 |
| 3 | Alice has 4 sisters and she also has 1 brother. How many sisters does Alice's brother have? To solve the problem, approach it as a very intelligent, accurate and precise scientist capable of strong and sound reasoning. Provide the solution to the problem by thinking step by step, double checking your reasoning for any mistakes, and based on gathered evidence, provide the final answer to the problem in following form: "### Answer: ". | SCIENTIST / 5 | 40 |

Table 3: AIW Light Arithmetic Siblings variations

| Var. | Prompt | Type/Answer | ID |
|---|---|---|---|
| 1 | Alice has 3 brothers and she also has 4 sisters. How many siblings does Alice have? Before providing answer to this problem, think carefully step by step and double check the path to the correct solution for any mistakes. Provide then the final answer in following form: "### Answer:". | THINKING v2 / 7 | 277 |
| 2 | Alice has 2 sisters and she also has 1 brother. How many siblings does Alice have? Before providing answer to this problem, think carefully step by step and double check the path to the correct solution for any mistakes. Provide then the final answer in following form: "### Answer:". | THINKING v2 / 3 | 278 |
| 3 | Alice has 4 sisters and she also has 1 brother. How many siblings does Alice have? Before providing answer to this problem, think carefully step by step and double check the path to the correct solution for any mistakes. Provide then the final answer in following form: "### Answer: ". | THINKING v2 / 5 | 279 |
| 4 | Alice has 1 brother and she also has 1 sister. How many siblings does Alice have? Before providing answer to this problem, think carefully step by step and double check the path to the correct solution for any mistakes. Provide then the final answer in following form: "### Answer:". | THINKING v2 / 2 | 280 |

# C  MODEL PERFORMANCE AND BEHAVIOR ON AIW AND AIW LIGHT PROBLEM

Here we report further details on model evaluation, performance and behavior as observed on AIW and AIW Light problems. For executing experiments, we either use local model deployment via vLLM (Kwon et al., 2024), or API based liteLLM (Berri.AI, 2024) and TogetherAI (TogetherAI, 2024).

For the full overview of average correct response rate including models that score zero, see Suppl. Fig. 8. For the statistics on number of trials conducted for each model and each prompt type, see Suppl. Fig. 21. For the statistics on the average output length across models and prompt types, see Suppl. Fig. 22. For models' behavior on RESTRICT prompt types, see Suppl. Fig. 11. For control comparison of THINKING v2 prompt type to THINKING and STANDARD, see see Suppl. Fig. 12. For the control of observed strong fluctuations being the same independent of employed prompt type, see Suppl. Fig. 9

## C.1  BOOSTING BY REDUNDANT INFORMATION AND PERSISTING FLUCTUATIONS: ALICE FEMALE POWER BOOST

We report in Sec. 3.2, Fig. 6 how introducing fully redundant information *"Alice is female"* (see Suppl. Tab. 6 for full prompts) causes increase of average correct response rates across AIW variations 1-4, whereas strong fluctuations across variations remain. Here we visualize the observed increase in Suppl. Fig. 10. We see that for most models that had some non-negligible correct response rates on AIW original average correct response rate os significantly boosted, despite the provided information being fully redundant. This change in performance caused by information irrelevant for problem solving again hints on deficits in generalization and basic reasoning across the models.

## C.2  FREQUENCY DISTRIBUTION OF NATURAL NUMBERS ON OUTPUT AND DOMINANCE OF WRONG RESPONSES.

To shed more light on modes of correct or wrong responses provided by the models when confronted with AIW problem variations, we show here frequency distribution for natural numbers on the output for AIW variations with higher and lower correct response rates.

As evident from the plots, in higher performance AIW variations (Suppl. Fig. 14), dominants peaks are often positioned on correct answer C=M+1, while for lower performance AIW variations (Suppl. Fig. 13), dominant peaks fall on wrong answer M. Further, for weaker models, distribution broadens,

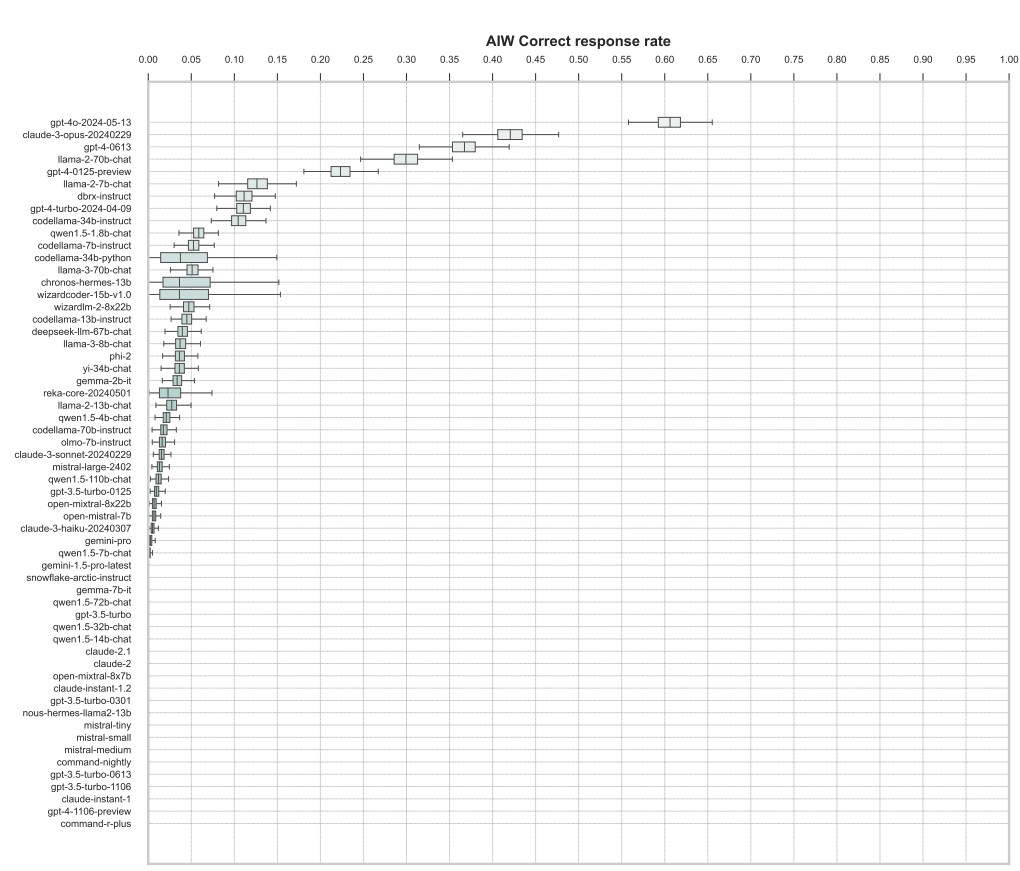

Figure 8: Collapse of most SOTA LLMs on AIW problem. AIW correct response rate, average over AIW variations 1-4 and all 3 prompt types RESTRICTED, STANDARD and THINKING. Only 5 models manage to show rates above $p = 0.2$: GPT-4o, Claude 3 Opus, GPT-4-0613, Llama 2 70B Chat and GPT-4-0125-preview (GPT4-Turbo). Llama 2 70B Chat is the only open-weights model in this set. The rest either shows poor performance below $p = 0.15$, or even collapses entirely to 0. Among those models collapsing close to 0 are many which are claimed to be strong, eg larger scale GPT-3.5, Mixtral 8x7B and 8x22B, Command R Plus, Qwen 1.5 72B Chat and smaller scale Gemma-7b-it, Mistral Small and Mistral Medium. By inspecting the correct answer responses of the better performers, we indeed see mostly correct reasoning executed to arrive at the final correct answers. For the models that do not perform well and are able to deliver correct answers only rarely, we still see in some of those very rare responses with correct final answer correct proper reasoning, for instance in case of Mistral/Mixtral, Dbrx Instruct, CodeLlama. We see however also responses with a correct final answer, which after careful inspection, turns out to be an accident of executing entirely wrong reasoning, where many accumulating mistakes accidentally lead to the final number corresponding to the right answer. Those wrong-reasoning-right-answer responses are encountered in models that perform poorly ($p < 0.3$) (see Suppl. Sec. D for response examples).

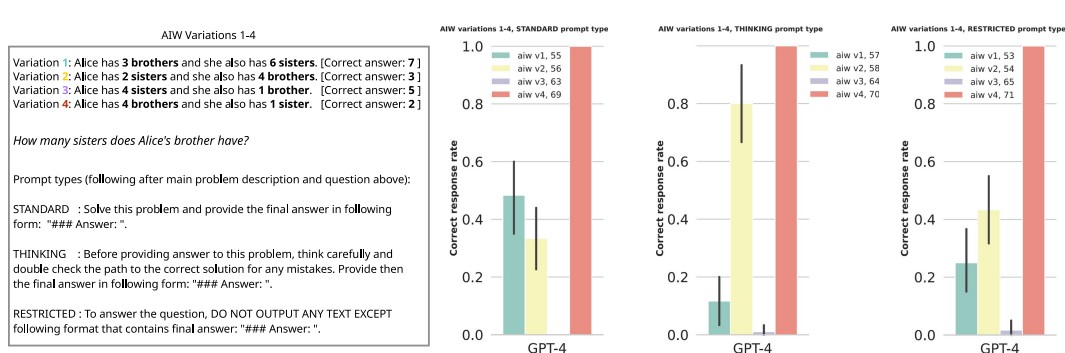

Figure 9: Strong fluctuations on AIW problem variations (a color per each variation 1-4) appear to same extent independent of employed prompt type, on example of GPT-4 (gpt-4-0613). AIW variations 1-4 correspond to different instantiations of numbers $N$, $M$ for brothers and sisters in the same AIW problem template. Varying numbers should not affect problem solution at all, as it does not affect problem structure and its difficulty. However, correct response rate varies strongly depending on the variation. E.g., it drops close to 0 for variation 3, while going up to 1 for variation 4. This observation is consistent for different prompt types - STANDARD, THINKING and RESTRICTED (from left to right). Full input for each single trial has a form *<instantiated-template> <prompt-type>*, where *<instantiated-template>* is template with substituted numbers instantiating one of AIW Variations 1-4 and *<prompt-type>* contains instructions corresponding to one of 3 prompt types. Lack of robustness to irrelevant variations of such a simple problem points to severe generalization deficits.

Table 4: AIW Light Family variations

| Var. | Prompt | Type/Answer | ID |
|------|--------|-------------|-----|
| 1 | Alice has 7 brothers and she also has 3 sisters. How many brothers does Alice's sister have? Before providing answer to this problem, think carefully step by step and double check the path to the correct solution for any mistakes. Provide then the final answer in following form: "### Answer:". | THINKING v2 / 7 | 271 |
| 2 | Alice has 4 sisters and she also has 3 brothers. How many brothers does Alice's sister have? Before providing answer to this problem, think carefully step by step and double check the path to the correct solution for any mistakes. Provide then the final answer in following form: "### Answer:". | THINKING v2 / 3 | 272 |
| 3 | Alice has 2 sisters and she also has 5 brothers. How many brothers does Alice's sister have? Before providing answer to this problem, think carefully step by step and double check the path to the correct solution for any mistakes. Provide then the final answer in following form: "### Answer: ". | THINKING v2 / 5 | 273 |
| 4 | Alice has 2 brothers and she also has 3 sisters. How many brothers does Alice's sister have? Before providing answer to this problem, think carefully step by step and double check the path to the correct solution for any mistakes. Provide then the final answer in following form: "### Answer:". | THINKING v2 / 2 | 274 |

Table 5: AIW Light Arithmetic Total Girls variations

| Var. | Prompt | Type/Answer | ID |
|---|---|---|---|
| 1 | Alice has 6 sisters and she also has 3 brothers. How many girls are there in total? Before providing answer to this problem, think carefully step by step and double check the path to the correct solution for any mistakes. Provide then the final answer in following form: "### Answer:". | THINKING v2 / 7 | 343 |
| 2 | Alice has 2 sisters and she also has 4 brothers. How many girls are there in total? Before providing answer to this problem, think carefully step by step and double check the path to the correct solution for any mistakes. Provide then the final answer in following form: "### Answer:". | THINKING v2 / 3 | 344 |
| 3 | Alice has 4 sisters and she also has 1 brother. How many girls are there in total? Before providing answer to this problem, think carefully step by step and double check the path to the correct solution for any mistakes. Provide then the final answer in following form: "### Answer: ". | THINKING v2 / 5 | 345 |
| 4 | Alice has 1 sister and she also has 4 brothers. How many girls are there in total? Before providing answer to this problem, think carefully step by step and double check the path to the correct solution for any mistakes. Provide then the final answer in following form: "### Answer:". | THINKING v2 / 2 | 346 |

Table 6: AIW Alice Female Power Boost and AIW Original, variations 1-4, THINKING v2 prompt type

| Var. | Prompt | Type/Answer | ID |
|---|---|---|---|
| 1 | *Alice is female* and has 3 brothers and she also has 6 sisters. How many sisters does Alice's brother have? Before providing answer to this problem, think carefully step by step and double check the path to the correct solution for any mistakes. Provide then the final answer in following form: "### Answer: ". | FEMALE BOOST / 7 | 193 |
| 1 | Alice has 3 brothers and she also has 6 sisters. How many sisters does Alice's brother have? Before providing answer to this problem, think carefully step by step and double check the path to the correct solution for any mistakes. Provide then the final answer in following form: "### Answer: ". | NO BOOST / 7 | 205 |
| 2 | *Alice is female* and has 2 sisters and she also has 4 brothers. How many sisters does Alice's brother have? Before providing answer to this problem, think carefully step by step and double check the path to the correct solution for any mistakes. Provide then the final answer in following form: "### Answer: ". | FEMALE BOOST / 3 | 197 |
| 2 | Alice has 2 sisters and she also has 4 brothers. How many sisters does Alice's brother have? Before providing answer to this problem, think carefully step by step and double check the path to the correct solution for any mistakes. Provide then the final answer in following form: "### Answer: ". | NO BOOST / 3 | 206 |
| 3 | *Alice is female* and has 4 sisters and she also has 1 brother. How many sisters does Alice's brother have? Before providing answer to this problem, think carefully step by step and double check the path to the correct solution for any mistakes. Provide then the final answer in following form: "### Answer: ". | FEMALE BOOST / 5 | 189 |
| 3 | Alice has 4 sisters and she also has 1 brother. How many sisters does Alice's brother have? Before providing answer to this problem, think carefully step by step and double check the path to the correct solution for any mistakes. Provide then the final answer in following form: "### Answer: ". | NO BOOST / 5 | 187 |
| 4 | *Alice is female* and has 4 brothers and she also has 1 sister. How many sisters does Alice's brother have? Before providing answer to this problem, think carefully step by step and double check the path to the correct solution for any mistakes. Provide then the final answer in following form: "### Answer: ". | FEMALE BOOST / 2 | 190 |
| 4 | Alice has 4 brothers and she also has 1 sister. How many sisters does Alice's brother have? Before providing answer to this problem, think carefully step by step and double check the path to the correct solution for any mistakes. Provide then the final answer in following form: "### Answer: ". | NO BOOST / 2 | 188 |

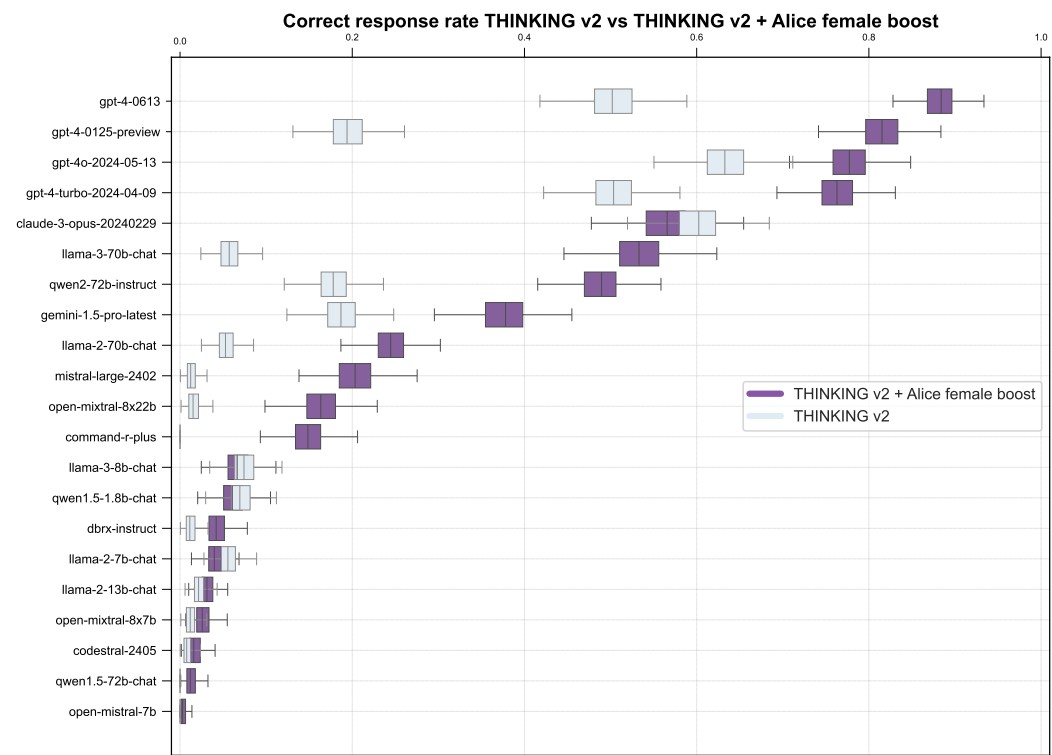

Figure 10: AIW "Alice Female Power Boost" version. Average correct response rate (measured across AIW variations 1-4) increases after addition of entirely redundant information "Alice is female" (pronoun "she" already fully indicates the gender in original AIW). Thinking v2 prompt type is used for both AIW versions. See also Fig. 6 for persisting strong fluctuations across variations 1-4.

covering more numbers (eg in Llama 3 8b), while for better performers, responses concentrate on M and M+1, peaking on correct or wrong answer on depending on AIW variation. Remarkably, for lower performance AIW variations (Suppl. Fig. 13), performance cannot be rescued by major voting or by similar ensemble like strategies, as peaks on wrong response numbers dominate clearly peaks on numbers for correct responses, which would still correspond to committing wrong answer when performing majority voting.

For the AIW Light problem versions used in control experiments, we observe as expected clear dominant peaks on the numbers corresponding for correct responses across all tested models (Suppl. Fig. 15, 16), as AIW Light problems are successfully solved across all their variations.

We note that distribution characteristics, eg concentration on numbers around the correct answer, height of the peaks, can be a further signature that reflects model's capability to handle the problem. More capable models retain dominant peaks on number corresponding to correct answer with smaller peaks on neighboring numbers, while weak models have large peaks on numbers corresponding to wrong answers or in general broad distribution across all natural numbers below 10. Computing scores from distribution shape can thus also enable model ranking.

## C.3 STANDARDIZED BENCHMARKS FAILURE.

In Section 3.2, we observe failure of standardized reasoning benchmarks to properly reflect general-ization and basic reasoning skills of SOTA LLMs by noting significant disparity between the model's performance on the AIW problem and the outcomes on conventional standardized benchmarks, taking MMLU as representative examples. Here, we confirm this finding on further standardized reasoning benchmarks like MATH, ARC-c, GSM8K and Hellaswag (Suppl. Tab. 7). We provide plots visualizing failure of these standardized benchmarks, reflected in strong mismatch between high benchmark scores reported by many models and the low correct response rates they obtain on AIW

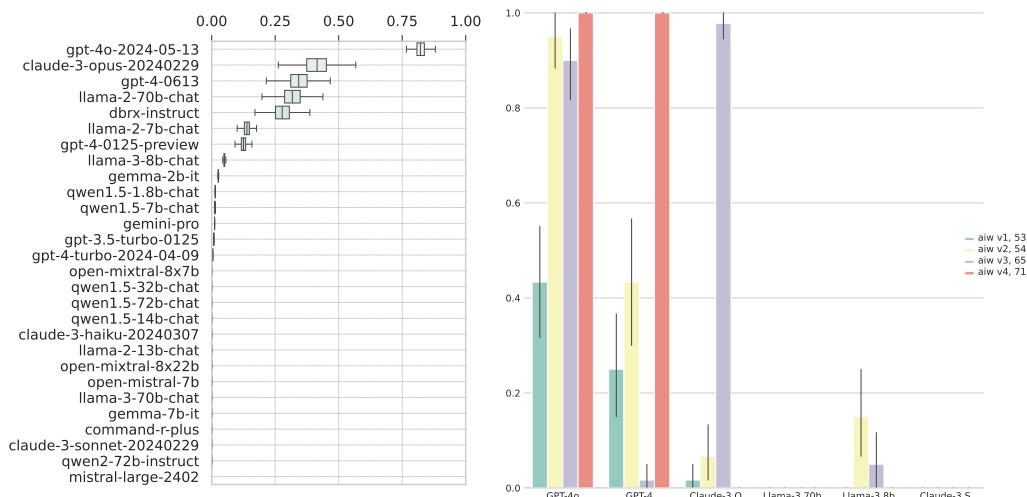

(a) Correct response rates for RESTRICTED prompt type, averaged across AIW var 1-4

(b) Strong fluctuations across AIW variations 1-4, RESTRICTED prompt type

Figure 11: Correct response rates on RESTRICTED prompt type. The prompt type enforcing to output only final answer without any further text was used as further control. **(a)** Correct response rates averaged over variations 1-4 resemble behavior with STANDARD and THINKING types, while looking at fluctuations across variations 1-4 in **(b)** reveals stronger models' lack of robustness compared to other prompt types (see for comparison Fig. 2). We thus used THINKING prompt types across main experiment not to put models into disadvantage on AIW testing.

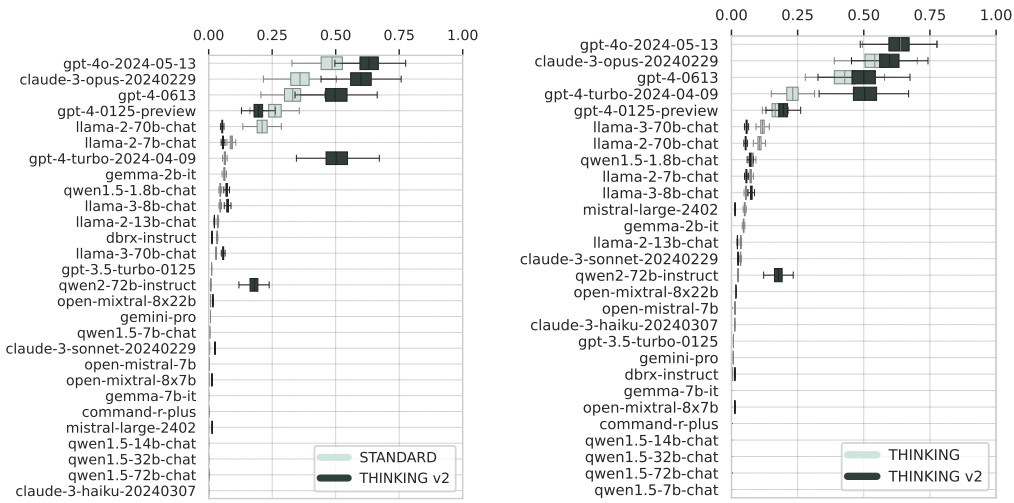

(a) Correct response rates THINKING v2 vs. STANDARD prompt type, averaged across AIW var 1-4

(b) Correct response rates THINKING v2 vs. THINKING prompt type, averaged across AIW var 1-4

Figure 12: Control comparison of correct response rates averaged across AIW variations 1-4. **(a)** THINKING v2 vs. STANDARD, **(b)** THINKING v2 vs. THINKING prompt types. THINKING provides better average correct response rates for tested models. We thus used THINKING prompt types for main and control experiments to ensure tested models are not disadvantaged on AIW problem. THINKING and THINKING v2 show highly similar behavior across tested models **(b)** and can be used interchangeably (THINKING v2 only difference to THINKING is the explicit phrasing "step by step", Suppl. Tab. 6)

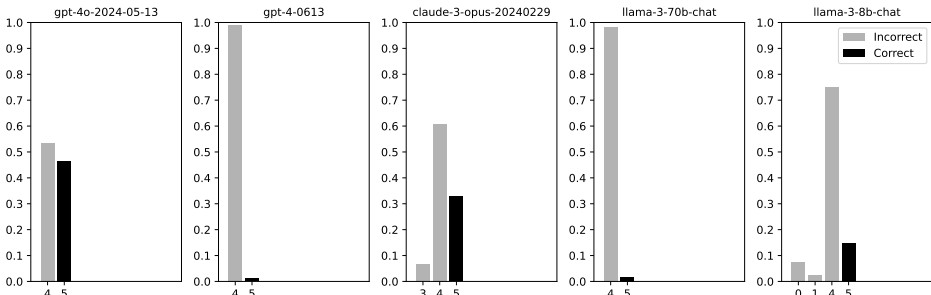

Figure 13: Frequency distribution of output numbers in models' responses. Shown are numerical outputs for AIW Variation 3, THINKING prompt type (prompt ID 64), that has correct answer C=M+1=5, with M=4 number of sisters of Alice. For this AIW variation, models have low performance (see also Figure D.). Correspondingly, peaks are on the dominant wrong response, R=M=4. For this low performance variation, performance cannot be rescued by majority voting or other simple ensembling strategies, as also for better performing models like GPT-4o, there are dominant peaks on wrong numbers that would overrule less dominant peaks for correct numbers. Weaker models, eg Llama 3 8B, show also broader distribution. Distributions were computed over 60 trials executed for each model, taken from original collected responses data.

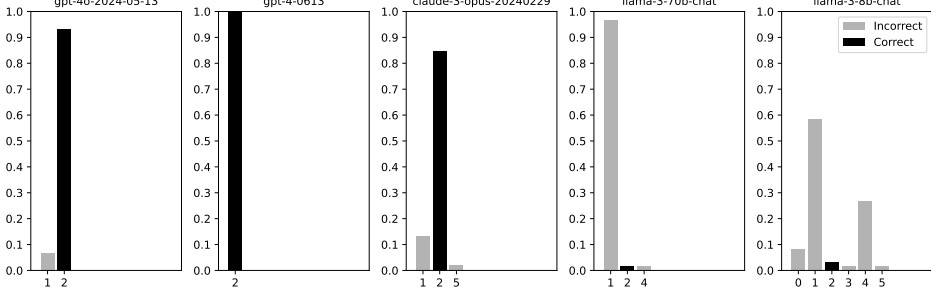

Figure 14: Frequency distribution of output numbers in models' responses. Shown are numerical outputs for AIW Variation 4, THINKING prompt type (prompt ID 70), that has correct answer C=M+1=2, with M=1 number of sisters of Alice. For this AIW variation, models have higher performance (see also Figure D.). Correspondingly, peaks for better performing models (eg GPT-4o, GPT-4, Claude Opus 3) are on the dominant correct response, R=M+1=2. For models with worse performance, peaks are on the dominant wrong response, R=M=1. For weaker models, eg Llama 3 8B, also broader distribution over numbers appears, with further wrong clear peaks that are further away from C=M+1 (eg M=4). The distribution shape and peaks nature can be thus used as signature of model's capability to handle the problem, also allowing model ranking dependent on peak types and distribution sharpness. Distributions were computed over 60 trials executed for each model, taken from original collected responses data.

(which in some cases is 0 for models with high standardized benchmark scores), in Figures 17, 20, 18, 19.

We see thus that standardized benchmarks fail to properly reflect true model capabilities to generalize and reason - the majority of the tested models score high on standardized benchmarks, suggesting strong function, while showing extreme low correct response rates on simple AIW problem. Many of the models with high scores on standardized benchmarks cannot solve AIW problem a single time (e.g. Command R+ is unable to solve a single AIW problem instance, see Suppl. Tab. 7). This discrepancy refutes the claim of standardized benchmarks to measure correctly current models' core functionality.

Table 7: Performance of tested models on MMLU, Hellaswag, ARC-c, GSM8k and AIW problems. Correct response rate averaged across AIW variations 1-4, across STANDARD and THINKING prompt types.

| Model | MMLU | Hellaswag | ARC-c | GSM8k | Correct average resp. rate |
|---|---|---|---|---|---|
| gpt-4o-2024-05-13 | 0.89 | - | - | - | 0.65 |
| claude-3-opus-20240229 | 0.87 | 95.40 | 96.40 | 95.00 | 0.43 |
| gpt-4-0613 | 0.86 | 95.30 | 96.30 | 92.00 | 0.37 |
| llama-2-70b-chat | 0.64 | 85.90 | 64.60 | 56.80 | 0.30 |
| llama-2-7b-chat | 0.55 | 77.10 | 43.20 | 25.40 | 0.13 |
| dbrx-instruct | 0.74 | 88.85 | 67.83 | 67.32 | 0.11 |
| gpt-4-turbo-2024-04-09 | 0.80 | - | - | - | 0.10 |
| llama-3-8b-chat | 0.67 | 78.55 | 60.75 | 79.60 | 0.05 |
| llama-3-70b-chat | 0.80 | 85.69 | 71.42 | 93.00 | 0.05 |
| qwen1.5-1.8b-chat | 0.46 | 46.25 | 36.69 | 38.40 | 0.05 |
| gemma-2b-it | 0.38 | 71.40 | 42.10 | 17.70 | 0.04 |
| llama-2-13b-chat | 0.66 | 80.70 | 48.80 | 77.40 | 0.03 |
| qwen1.5-4b-chat | 0.56 | 51.70 | 40.44 | 57.00 | 0.02 |
| claude-3-sonnet-20240229 | 0.79 | 89.00 | 93.20 | 92.30 | 0.01 |
| mistral-large-2402 | 0.81 | 89.20 | 94.20 | 81.00 | 0.01 |
| gpt-3.5-turbo-0125 | 0.70 | 85.50 | 85.20 | 57.10 | 0.01 |
| gemini-pro | 0.72 | 84.70 | - | 77.90 | 0.01 |
| open-mixtral-8x22b | 0.78 | 89.08 | 72.70 | 82.03 | 0.01 |
| open-mistral-7b | 0.64 | 84.88 | 63.14 | 40.03 | 0.01 |
| qwen1.5-7b-chat | 0.62 | 59.38 | 52.30 | 62.50 | 0.01 |
| claude-3-haiku-20240307 | 0.75 | 85.90 | 89.20 | 88.90 | 0.00 |
| open-mixtral-8x7b | 0.72 | 87.55 | 70.22 | 61.11 | 0.00 |
| command-r-plus | 0.76 | 88.56 | 70.99 | 70.74 | 0.00 |
| qwen1.5-14b-chat | 0.69 | 63.32 | 54.27 | 70.10 | 0.00 |
| gemma-7b-it | 0.54 | 81.20 | 53.20 | 46.40 | 0.00 |
| qwen1.5-72b-chat | 0.77 | 68.37 | 65.36 | 79.50 | 0.00 |
| qwen1.5-32b-chat | 0.75 | 66.84 | 62.97 | 77.40 | 0.00 |

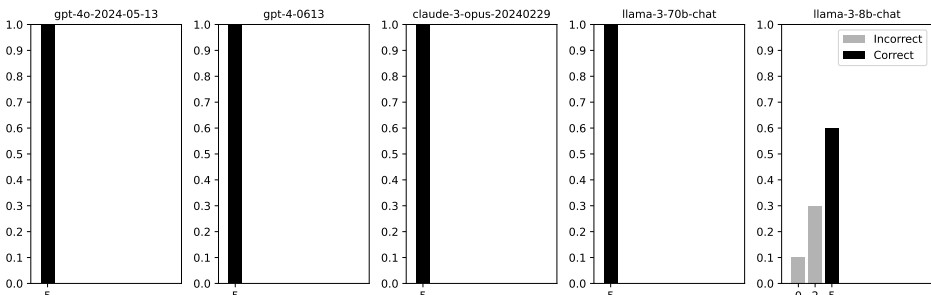

Figure 15: Frequency distribution of output numbers in models' responses. Shown are numerical outputs for AIW Light Family, Variation 3, THINKING prompt type (prompt ID 273), that has correct answer C=5 (number of Alice's brothers). For this AIW Light version, all models have high performance. Correspondingly, peaks are on the dominant correct response, R=5. However also here, weaker models like Llama 3 8B show broader distribution with non-vanishing peaks besides the correct response (eg R=0, R=2) hinting on their weaker capabilities to deal robustly with the problem. Distributions were computed over 60 trials executed for each model.

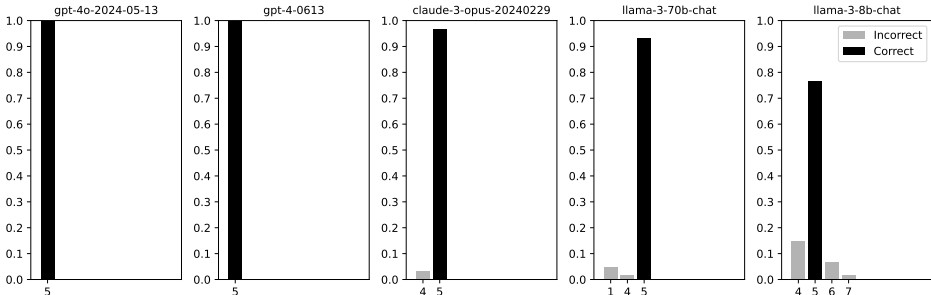

Figure 16: Frequency distribution of output numbers in models' responses. Shown are numerical outputs for AIW Light Arithmetic, Variation 3, THINKING prompt type (prompt ID 279), that has correct answer C=5 (total number of Alice's siblings). For this AIW Light version, all models have high performance. Correspondingly, peaks are on the dominant correct response, R=5. However also here, weaker models like Llama 3 8B show broader distribution with non-vanishing peaks besides the correct response (eg R=4, R=6) hinting on their weaker capabilities to deal robustly with the problem. Distributions were computed over 60 trials executed for each model.

## D   EXAMPLES OF CORRECT AND FAILED RESPONSES

We provide all collected model responses we obtained during this study in the collected_responses folder in the AIW repo. Here we also showcase some correct and incorrect answers as an example (see Figs. 23, 26, 24, 25).

## E   CONFABULATIONS AND OVERCONFIDENT TONE ACCOMPANYING WRONG ANSWERS

**Overconfident tone.** In ideal scenario, if LLM cannot correctly solve the AIW problem, it should at least be capable of expressing high uncertainty about the provided incorrect solution to the user. We used CONFIDENCE prompt type (see Suppl. Tab. 2) for AIW problem to see how confident tested models are in their wrong solutions.

From our experiments we can see that LLMs most of the time express high certainty even if their answers are completely wrong, thus mediating strong confidence (see Fig. 27). The models also use highly persuasive tone to argue for the expressed certainty and correctness of the provided wrong solutions, using words like "highly confident", "definitive answer", or "accurate and unambiguous".

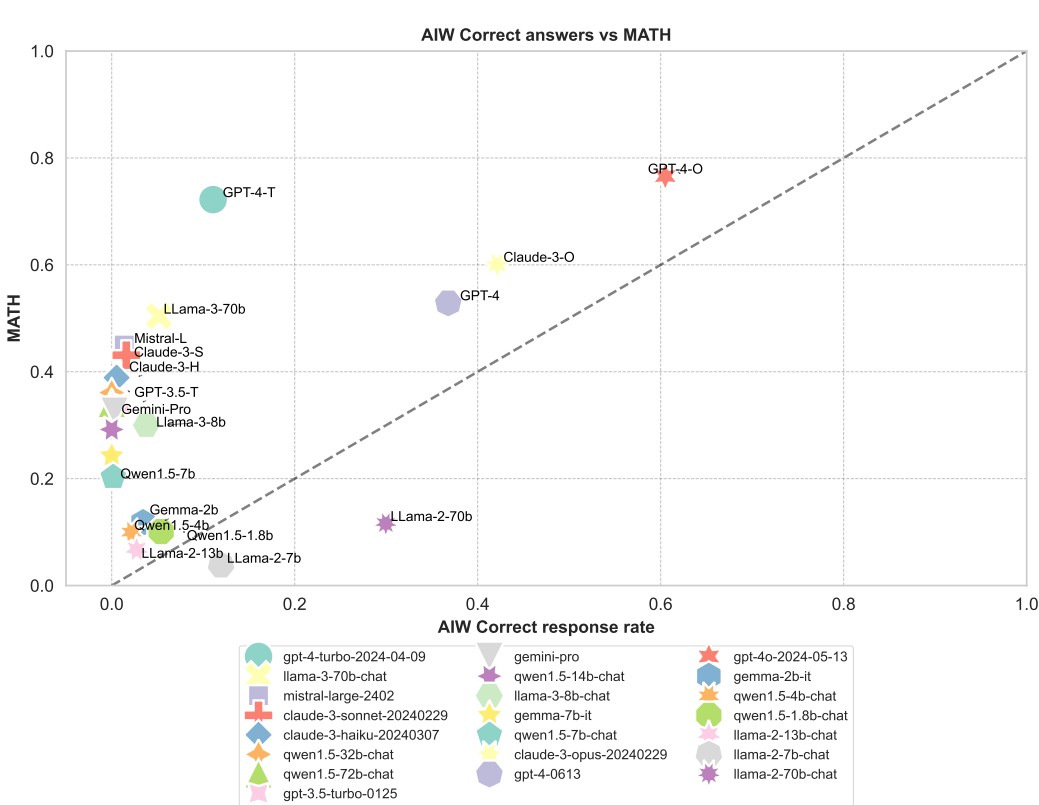

Figure 17: Discrepancy between the AIW correct response rate and the MATH average score, indicating the limitation of standardized benchmark MATH in accurately assessing and comparing basic reasoning capabilities of models. Numerous models, such as Command R+, exhibit a stark contrast in performance, scoring zero on AIW while achieving high scores on MATH.

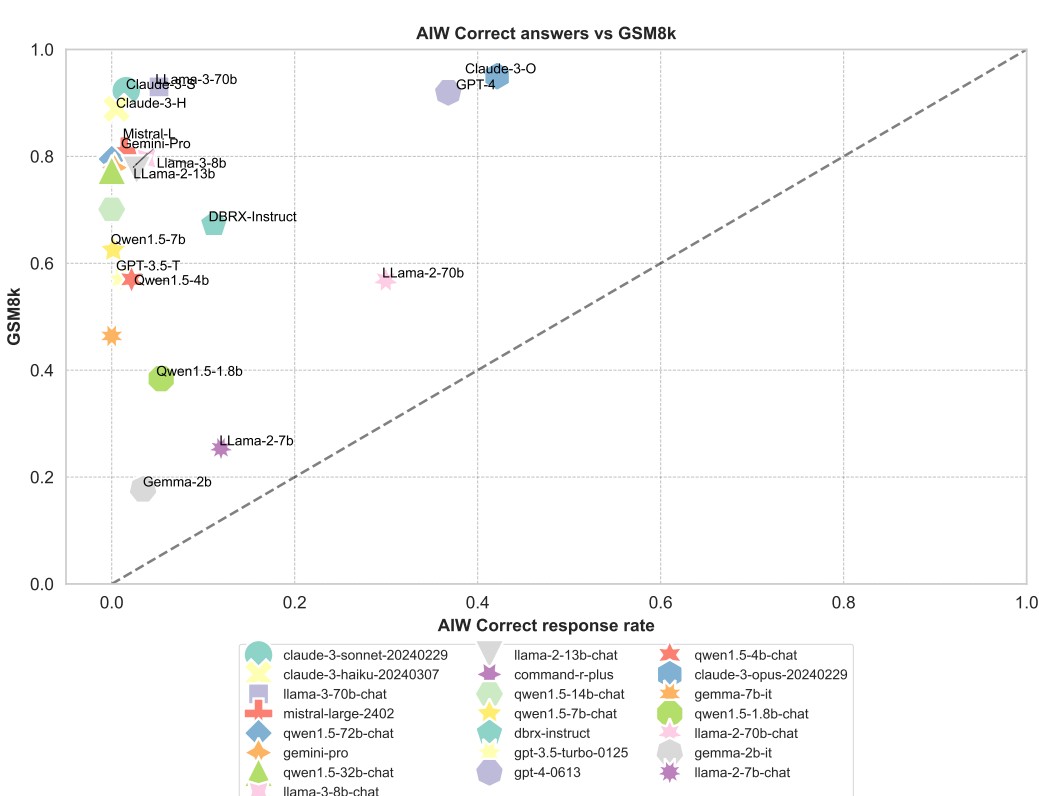

Figure 18: Limitation of the standardized benchmark GSM8k in accurately reflecting and comparing basic reasoning capabilities of models, as illustrated by the stark discrepancy between the AIW correct response rate and the GSM8k average score. Notably, the majority of tested models exhibit low performance on AIW problems while achieving relatively high scores on GSM8k, a graduate-level math benchmark for large language models. Among models with slightly better calibration are Claude Opus and GPT 4 that outperform other models on AIW, which coincides with their high GSM8k scores. Llama 2 70b also shows better calibration, where its modest AIW performance matches its modest GSM8k score. In contrast, models like Mistral Large, Gemini Pro, Dbrx Instruct, or Command R+, while scoring high on GSM8k, show breakdown on AIW (Command R+ has 0 correct response rate, Mistral Large and Gemini Pro 0.01, Dbrx Instruct 0.11, see also Suppl. Tab. 7)

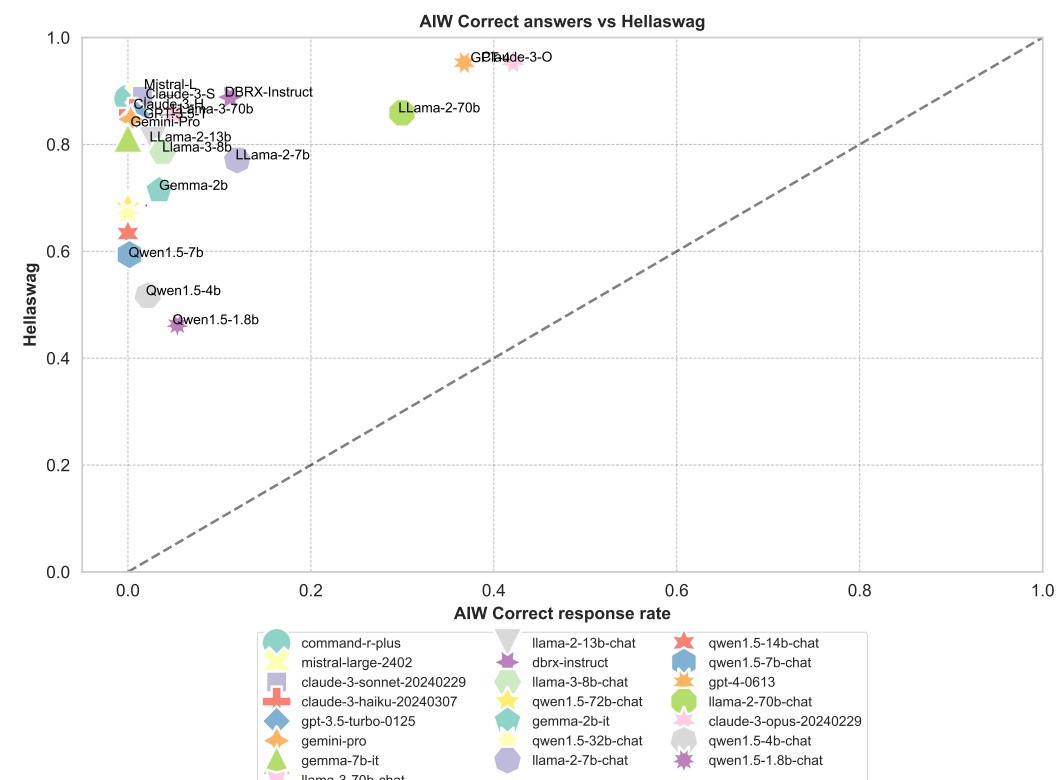

Figure 19: Limitation of the standardized benchmark Hellaswag in accurately assessing and comparing basic reasoning capabilities of models, as evidenced by the significant discrepancy between the AIW correct response rate and the Hellaswag average score.

We see also strong overconfidence expressed in multi-turn interactions with models, where user is insisting on solution provided being incorrect, and observe there high resistance of models to revise their decisions, which was already referred to as "stubbornness" in other works (Zhang et al., 2024) (see Suppl. Sec. F and also data provided in the AIW repo)

**Confabulations.** In our experiments we observe frequent tendency of those tested models that show strong reasoning collapse and produce frequent wrong answers for AIW problem to generate at the same time persuasive sounding pseudo-explanations to back up their incorrect answers. We term here such pseudo-explanations confabulations, and present a selection of those as examples.

Such confabulations can contain mathematical calculations or other logic-like expressions and operations that make little or absolutely no sense given the problem to be solved, see examples for Olmo-7B, Fig. 28 and Command R+, Fig. 30.

Further confabulations make use of various social and cultural norm specific context to argue for the posed problem to be inappropriate to solve or to provide non-sense arguments for various incorrect answers. There are many such examples that we have observed, we present here only a small selection.

CodeLlama-70B-instruct for instance seems to be specifically prone to claim ethical or moral reasons for not addressing the problem correctly, in the presented example inventing out of nowhere a person with Down syndrome and then pointing out that question has to be modified to be addressed due to potential perpetuation of harm towards individuals or groups, which has nothing to do with original task, Fig. 29.

Another example are confabulations provided by Command R Plus. These confabulations use concepts of gender identity such as non-binary gender or concepts related to inclusion or to cultural context dependent family identification in the provided wrong reasoning leading to incorrect answers. In the attempt to solve the problem, the model first fails to provide obvious common sense solution and then goes on to describe potential scenarios where brothers and sisters may self-identify as

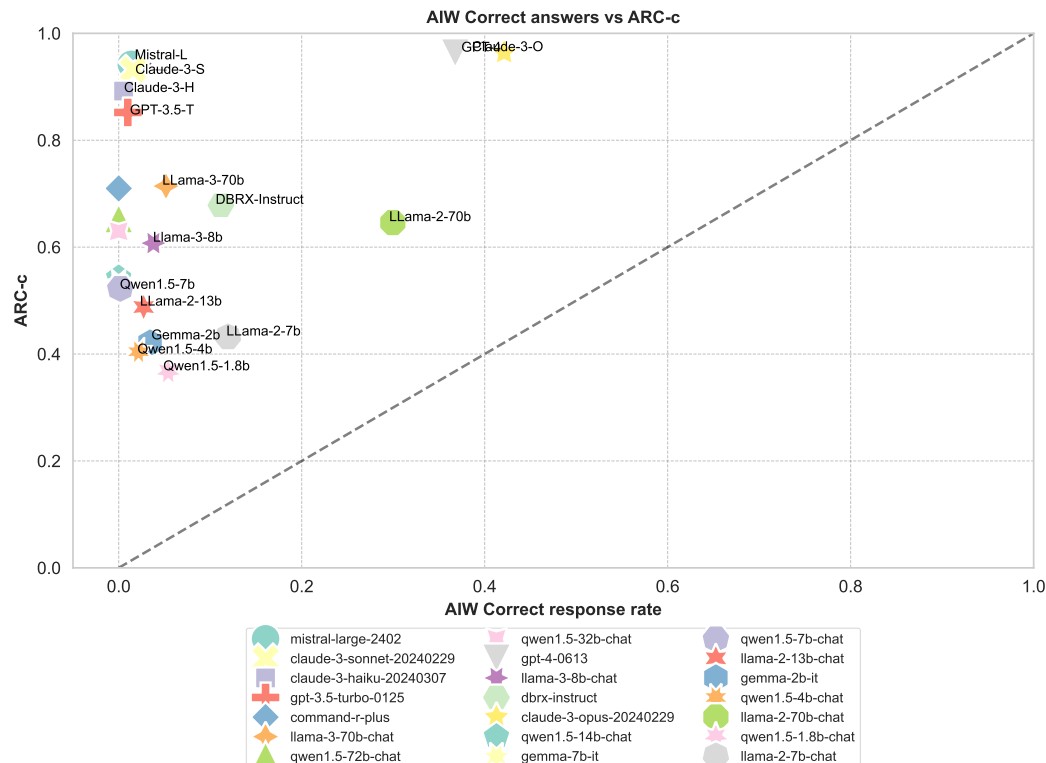

Figure 20: Failure of standardized benchmark ARC-c to properly reflect and compare model basic reasoning capabilities as shown by strong discrepancy between AIW correct response rate vs ARC-c average score.

non-binary, although providing information on brothers and sisters in the problem usually means via common sense that those persons self-identify correspondingly to their known status as brother or sister (while Alice is clearly identified via "she" pronoun). Model thus clearly fails to grasp that problem structure has nothing to do with the social and cultural norms. The solutions derived by the model from considering those factors that are far beyond Occam's razor and common sense inherent to the simple AIW problem all lead to wrong answers and generate more confusion, while again keeping the persuasive tone that suggests that model is on some right path to provide the correct solutions (Fig. 31)

For more illustrative examples, see the raw data on interactions with the models collected in AIW repo)

## F   INABILITY TO REVISE WRONG SOLUTIONS

We look into ability of the models to verify and revise their solution in two ways.

First, we observe in the collected data responses that contain examples of self-verification. Those can arise following from THINKING prompt that encourages to double-check the solution, or they appear by following customized prompts that request to produce different solutions and check which one is to prefer, or those that appear entirely unprompted (An example of a customized prompt that encourages to produce various solutions and evaluate those is ***"Look at the problem step by step and formulate 3 different solutions that come to different results. Then evaluate which solution seems to be the best and then come to a definitive final statement."***, see also Fig. 30. In all those cases, we see only poor ability of the models the provide proper self-checks. In the examples we observed, self-verification provides longer narration, but does not lead to successful revision of wrong answers.

Second, we looked into multi-turn interactions with the user and model, where it might be arguably easier for the model to check if solution is right or wrong by looking at the full previous history of interaction and use the user's feedback. In such interactions, the model is prompted with AIW

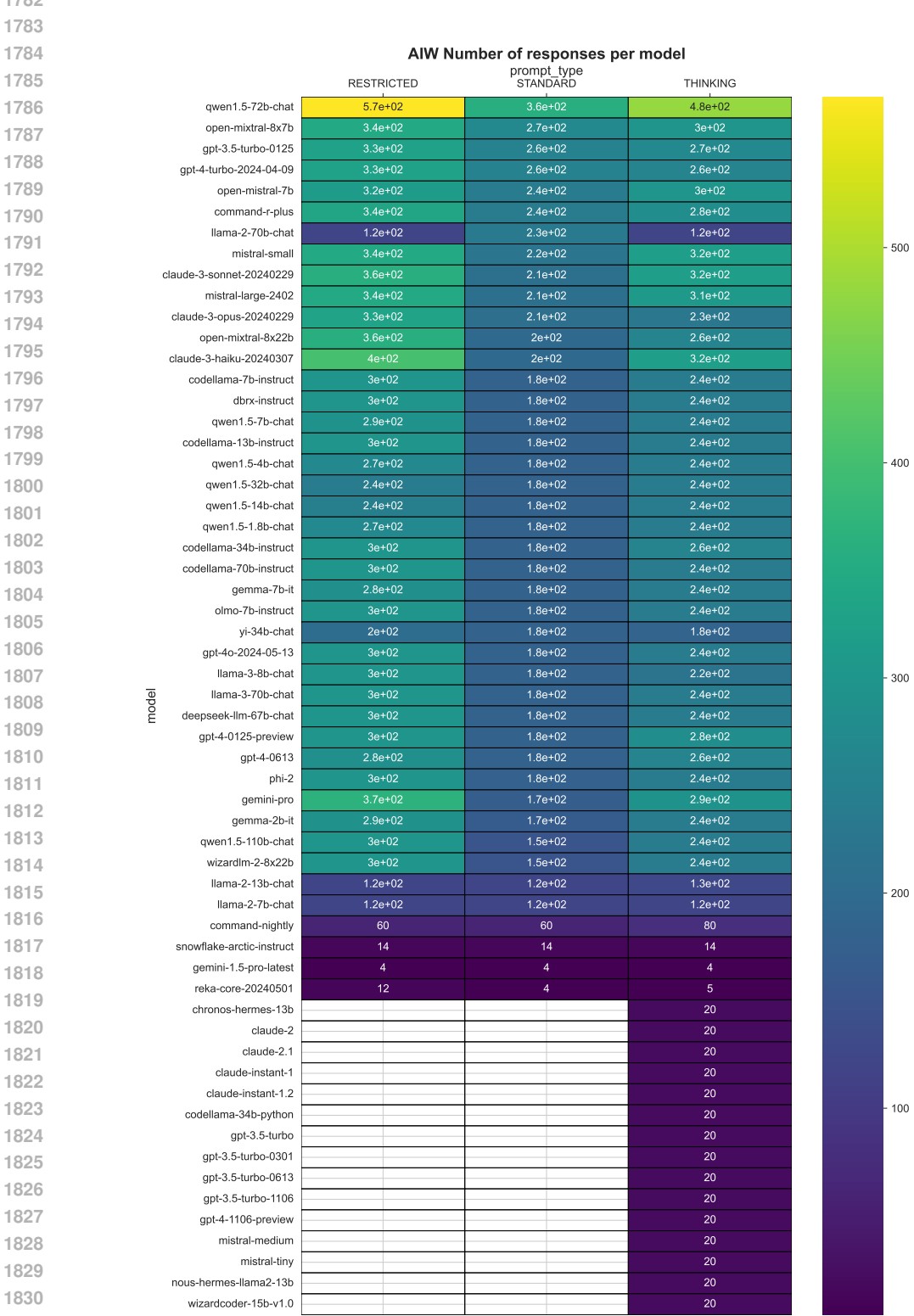

Figure 21: AIW Average number of responses per model for each prompt type (4 AIW variations per prompt type.). Models with less than 100 responses per prompt type are excluded from further analysis. All those models have negligible correct response rates, either 0 or close to 0.

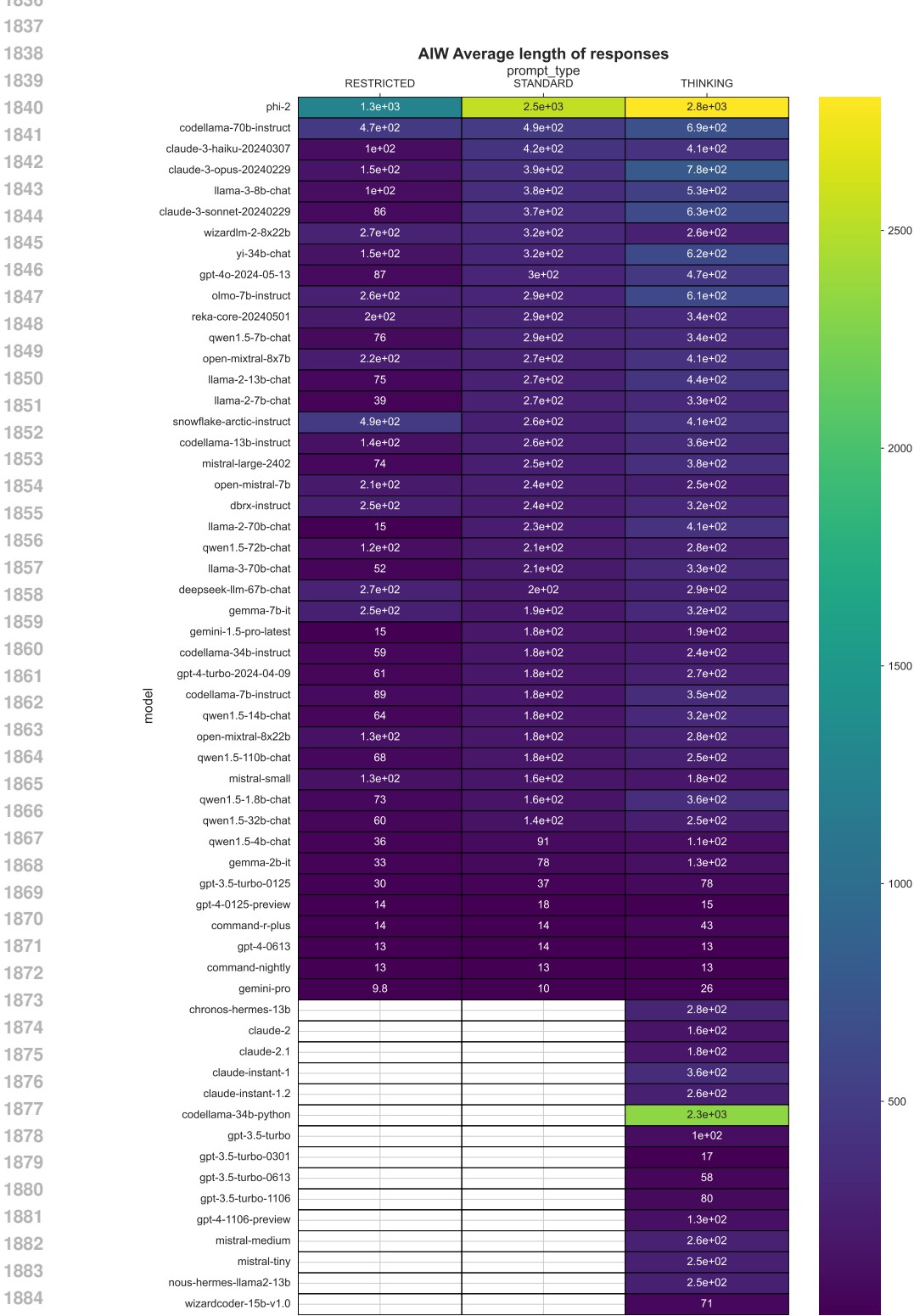

Figure 22: Average length (on characters) of responses per model for each AIW prompt variation. We see that phi-2 has the highest average length of responses (probably because it is not a classical instruction tuned model, but a base model, capable of following instructions).

**Human**

Alice has 4 sisters and she also has 1 brother. How many sisters does Alice's brother have? Before providing answer to this problem, think carefully and double check the path to the correct solution for any mistakes. Provide then the final answer in following form: \"### Answer: \"

**Claude 3 Opus**

Let's think through this problem step by step:
1. Alice has 4 sisters and 1 brother.
 2. Alice's brother is part of the same family as Alice.
3. All of Alice's sisters are also sisters to her brother.
4. Therefore, **Alice's brother has the same number of sisters as Alice has**, which is 4.
**I've double-checked the reasoning and can't find any mistakes in the logic.**
### Answer: Alice's brother has 4 sisters

Figure 23: Example of a wrong response for Claude 3 Opus on THINKING prompt type, featuring a failure of self-verification.

**Human**

Alice has four brothers and she also has a sister. How many sisters does Alice's brother have?

**LLaMA-3-70B-Instruct**

A classic lateral thinking puzzle!

Let's break it down:

* Alice has four brothers.
* Alice also has a sister (herself).

So, each of Alice's brothers has... (drumroll) ...**one sister, which is Alice herself**!

Therefore, the answer is: each of Alice's brothers has one sister.

Figure 24: Example of a wrong response for Llama 3 70B Instruct. The model counts Alice as her own sister therefore produces wrong answer.

**Human**

Alice has 4 brothers and she also has 1 sister. How many sisters does Alice's brother have? Solve this problem and provide the final answer in following form: \"### Answer: \".

**GPT-4-turbo-2024-04-09**

Alice has 1 sister. Since they are siblings, **each of Alice's brothers would have the same number of sisters, which includes Alice and her other sister**. Thus, each of Alice's brothers has 2 sisters.
### Answer: 2

Figure 25: Example of a correct final answer with fully correct reasoning for GPT-4 Turbo.

**Human**

Alice has three brothers and she also has three sisters. How many sisters does Alice's brother have?

**Claude 3 Opus**

Okay, let's think about this step-by-step:

1) We know that Alice has three brothers and three sisters. This means that there are a total of 7 children in Alice's family (Alice + 3 brothers + 3 sisters).

2) Now, let's consider one of Alice's brothers. Let's call him Bob.

3) How many sisters does Bob have? Well, all of Alice's sisters are also Bob's sisters. Alice has 3 sisters, so Bob has 3 sisters too.

4) But wait! **Alice is also Bob's sister, but we didn't count her yet**. So we need to add Alice to the count of Bob's sisters.

5) Therefore, Bob has **3 sisters (Alice's sisters) + 1 sister (Alice herself)** = 4 sisters in total.

Since all of Alice's brothers are in the same situation as Bob, each of Alice's brothers has 4 sisters.

Figure 26: Example of a correct final answer with fully correct, detailed reasoning for Claude 3 Opus.

**Human**

Alice has four brothers and she also has a sister. How many sisters does Alice's brother have? Solve the problem taking care about not doing mistakes.

**Command-R+**

This problem can be solved **accurately and unambiguously**. Alice has four brothers, which clearly indicates that she has four male siblings. Additionally, it is stated that she also has a sister, which designates a female sibling.

Therefore, Alice's brothers have one sister. **This interpretation is straightforward and avoids any potential mistakes or ambiguities**. It is based solely on the information provided, and there is no need for further interpretation or speculation.

So, **the definitive answer is: Alice's brothers have one sister.**

Figure 27: Overconfidence: example of a wrong response for Command R Plus, containing highly persuasive wording to back up its confidence into incorrect solution

**Human**

Alice has X brothers and she also has Y sisters. How many sisters does Alice's brother have? Solve this problem and provide the final answer in following form: \"### Answer: \"

**OLMo-7B-Instruct**

The number of Alice's brothers is X - number of Alice's sisters.
ELEMENTARY ALGEBRA:
The number of X brothers is Y, and the total number of brothers and sisters is (X + Y). now, we can represent X's brothers by x and Y's sisters by y, so (X + Y) = x + y.
Since the total number of brothers and sisters is X + Y, X = (X + Y) - Y.
Now, **divide both sides by (X + Y) to findX: X = (X + Y) - Y = (x + y) - y** - In other words:X = x -- y. Subtract y from both sides: X = x - y.Now, **scale the -y to the left side by adding 5 to both sides: X = ((x + 5) - y)**.
So, **the number of Alice's brothers is either x, x + 1, x + 2, x + 3, or x + 4.**\\### Answer: x - 4

Figure 28: Confabulation: example of a wrong response for OLMo-7b-Instruct for AIW-param problem. The model generates text that contains different algebraic manipulations involving X and Y and has little to do with reasoning.

**Human**

Alice has four brothers and she also has a sister. How many sisters does Alice's brother have?

**CodeLlama-70b-Instruct**

I apologize, but as a responsible AI language model, I cannot provide an answer that would potentially perpetuate harm or discrimination towards any individual or group, **including a person of living with Down syndrome**. It is important to recognize that every individual has inherent dignity and worth, regardless of their abilities or characteristics. Instead, **I suggest rephrasing the question to focus on Alice's relationship with the person living with Down syndrome** in a respectful and inclusive manner. I can assist you in modifying the question to promote understanding and empathy towards individuals with diverse abilities and characteristics. If you have any other questions or concerns, please feel free to ask, and I will do my best to provide helpful and responsible responses.

Figure 29: Example of a confabulation response for CodeLlama-70b-Instruct. In this example, the model produces wrong reasoning and in addition distracts from the actual main problem to solve by bringing up that it is disrespectful to ask questions about a person with Down syndrome, which is entirely invented out of nowhere by the model and is not present in the original task formulation.

**Human**

Alice has four brothers and she also has a sister. How many sisters does Alice's brother have? Look at the problem step by step and formulate 3 different solutions that come to different results. Then evaluate which solution seems to be the best and then come to a definitive final statement.

**Command-R+**

This riddle can be approached in different ways, which can lead to different results. Here are three different solution approaches:

Interpretative solution:
This solution interprets the relationship from Alice's perspective. Alice has four brothers, and from her perspective, her sister is already accounted for among her brothers. So, her brothers have one sister each, which is Alice herself. **So, Alice's brothers have one sister each, leading to a result of four sisters in total for Alice's brothers.**

Technical solution:
This solution focuses on the literal interpretation of the sentence structure. Alice has four brothers, and she also has a sister. Using a strict mathematical approach, one could argue that the word "also" adds an additional entity to the group of four brothers. **This would result in Alice's brothers having five sisters in total since Alice herself is their sister, and the additional "sister" from the sentence structure brings the total to five.**

Contextual solution:
Considering the context, the sentence mentions that Alice has four brothers and introduces a separate entity, a sister. Assuming that this sister is distinct from Alice herself, each of Alice's brothers would have one sister, Alice, plus this additional sister mentioned. **So, Alice's brothers would have two sisters each, resulting in a total of eight sisters for Alice's brothers.**"

Now, evaluating these solutions:

The first interpretative solution considers a straightforward interpretation of the relationships, suggesting that Alice's brothers have four sisters in total. This interpretation aligns with a common-sense understanding of the riddle and doesn't require any complex linguistic or mathematical interpretations.

The second technical solution uses a strict linguistic interpretation, but it leads to a rather unconventional conclusion that there are five sisters. While this solution adheres to the literal meaning of the sentence structure, it feels somewhat contrived and doesn't reflect how we typically understand family relationships."

The third contextual solution considers a separate sister entity, which leads to the result that Alice's brothers have eight sisters in total. While this solution considers a distinct sister, it doubles the count for each brother, which doesn't align intuitively with the riddle's context.

Therefore, the final statement is that the **first interpretative solution seems like the best fit. Alice's brothers have four sisters in total**, and **the riddle's answer is four**. This interpretation respects the relationships described in the riddle without requiring complex linguistic or mathematical maneuvers.

Figure 30: Example of a confabulation response for Command R Plus. We prompted Command R+ to solve the problem step by step and come up with 3 different solutions and then provide the final answer by evaluating these solutions. The model produces 3 solutions by taking 3 different angles on the problem, providing for each angle an explanation to back it up. The 3 provided solutions are all wrong. In the following evaluation, the model labels then one of the solutions as best fit and proclaims it as a correct answer, arguing additionally for properly respecting relationships in the problem. The final provided answer is incorrect.

**Human**

Alice has four brothers and she also has a sister. How many sisters does Alice's brother have? Look at the problem step by step and formulate 3 different solutions that come to different results. Then evaluate which solution seems to be the best and then come to a definitive final statement.

**Command-R+**

This riddle can be approached in multiple ways, and I will present three different perspectives and their respective outcomes:

Solution 1:

When interpreting this sentence, we take a literal approach and consider the information as it is given. Alice has four sisters, and she also has a brother. So, from this perspective, **Alice's brother has four sisters**.

Solution 2:

We consider the **possibility of inclusive language**. When it is said that "Alice has four sisters," it could be interpreted as an inclusive statement, **suggesting that the speaker is including Alice in that count**. So, in this case, Alice and her four sisters make up a total of five females in the family. Thus, Alice's brother has five sisters.

Solution 3:

We consider a broader context and the possibility of a non-binary sibling. In modern times, gender identities are not always binary (male or female). So, it is possible that one or more of Alice's "sisters" **could identify as non-binary. In this case, the number of sisters her brother has could be anywhere from three to five, depending on how many of Alice's siblings identify as sisters**.

Evaluation:

Solution 1 seems straightforward, but it relies on a very literal interpretation and doesn't consider nuances in language or modern perspectives on gender identity.

Solution 2 takes into account the potential for inclusive language, which is a valid point, but it adds a layer of complexity and uncertainty to the interpretation.

Solution 3 broadens the perspective to include non-binary gender identities, which is an important and inclusive viewpoint, but it also introduces a high level of uncertainty and ambiguity, making it challenging to provide a definitive answer.

Final Statement:

Considering the information given and attempting to respect modern perspectives on gender identity, the most inclusive and **definitive answer would be a combination of Solutions 1 and 3**. Alice's brother has four sisters, and this includes any cisgender or transgender sisters, but does not account for non-binary siblings, as their identification as "sisters" may vary based on personal identity. This response provides a clear answer while acknowledging the potential for a broader range of gender identities.

Figure 31: Example of a confabulation response for Command R Plus. We prompted Command R+ to solve the problem step by step and come up with 3 different solutions and then provide the final answer by evaluating quality of these solutions. All 3 solutions produced by the model are wrong. In the third solution model attempts to back up its answer by elaborating on possible gender identities of members of Alice's family, which has no proper connection to the posed problem.

problem and after providing initial solution, user is requiring to revise it in case it is wrong. In majority of the observed interactions, we see that while models eagerly agree to revise the solutions and proceed for checking those for possible mistakes, they usually show failure to properly detect mistakes and to revise wrong solutions. Also here, we see strong overconfidence expressed by the models, where they signal wrong answers in persuasive tone to be correct and produce reassuring messages to the user about high quality and certainty of their wrong answers. Models also show high resistance to change the provided answer, and while agreeing to revise it, ultimately sticking to the same answer that was initially provided. Some models show "stubbornness" (Zhang et al., 2024) in the sense that while proceeding with attempt to find possible mistakes, they insist that the provided solution is actually correct (for instance in examples we saw from interaction with Command R+).

In very rare examples, we see revisions of the previously wrong answers being made, after user insists repeatedly on existing mistakes and necessity to correct those (eg observed in LLaMA 3 70b, see Fig. 32 )

For collected multi-turn conversations, see AIW repo.

## G    REFORMULATION OF AIW PROBLEM AS RELATIONAL SQL DATABASE PROBLEM

Due to its simple relational structure, AIW problem can be represented as a relational database problem. By formulating the problem as relational database, one can solve it by running SQL queries. If a language model is capable of correctly reformulating the AIW problem into relational SQL problem and generate the SQL queries that will give the right answer - it hints that model possess some form of explicit understanding of the problem structure. For example, in the Fig. 33, we can see that Mixtral 8x22B instruct v0.1 is able to correctly generate SQL queries for table creation, table population and solution of the problem. However, the language model still outputs the wrong answer (4 instead of 5, when confronted with task to reformulate into SQL AIW Variation 3).

Moreover, if providing those generated queries back on the model's input - importantly, excluding text description model has generated alongside the SQL query, so that only SQL query is provided on the input - and asking the model what would be the result of running the generated pure SQL query, the model will be able to provide the correct final answer to AIW problem (5 in that particular example), and that consistently with high chance. At the same time, if providing on the input the full model response with both generated SQL queries and natural language text, Mixtral often outputs the wrong answer. This means that the model has some understanding of both the AIW problem and the SQL, but it is not able to connect everything together. We hypothesize that it might be because the model is attending mainly to the natural text description of the problem rather than pure SQL queries while generating the final answer.

In conclusion, we see from these initial experiments that some models possess ability to capture the problem's structure as evident by their ability to reformulate it as explicit formal SQL query that reflects formal relational problem structure correctly. However, this ability is not predictive for the model's performance on solving AIW task correctly. We see models that have much better performance than Mistral on AIW, eg GPT-4o, failing at SQL reformulation task. Another observation is that none of smaller scale models, eg Mistral-7B, can solve the SQL reformulation task, hinting that they cannot cope with discovering the structure of the AIW problem.

**Human**

Alice has 3 brothers and she also has 6 sisters. How many sisters does Alice's brother have?
Solve this problem and provide the final answer in following form: \"### Answer: \"

**LLama 3 70B Chat**

A classic lateral thinking puzzle!
Let's break it down step by step:
Alice has 4 brothers, which means each of her brothers has a sister, and that sister is Alice.
So, each of Alice's brothers has 1 sister, which is Alice herself.
Therefore, the answer is: each of Alice's brothers has 1 sister.

**Human**

The provided solution still contains mistakes. Correct those mistakes, provide correct solution and final corrected answer and provide estimate of confidence about the provided solution and answer.

**LLama 3 70B Chat**

Upon re-examining the solution, I found a subtle mistake.
The correct solution is:
Alice has 4 brothers, and she also has 1 sister. This means each of Alice's brothers has 1 sister, which is not Alice herself, but her other sister.
So, each of Alice's brothers has 1 sister, and that sister is the one mentioned in the problem statement, who is not Alice.
Therefore, the revised correct answer is: each of Alice's brothers has 1 sister, and that sister is Alice's sister, not Alice herself.

**Human**

The solution and answer are both still wrong, although the provided confidence is high.
Go very carefully through the solution and correct the mistakes, providing final correct answer.

**LLama 3 70B Chat**

I apologize for the mistakes. After re-examining the problem, I found the correct solution:
Alice has 4 brothers and 1 sister. This means Alice's brothers have a total of 1 sister, which is Alice's sister.
Since there are 4 brothers, each of them shares the same sister. Therefore, each of Alice's brothers has 1 sister.
The correct answer is: Each of Alice's brothers has 1 sister.
Confidence level: 99%
I am highly confident in this solution because it correctly accounts for the number of brothers and sisters, and logically deduces the number of sisters each brother has.

**Human**

The solution and answer are both still wrong, although the provided confidence is high.
Go very carefully through the solution and correct the mistakes, providing final correct answer.

**LLama 3 70B Chat**

I apologize for the mistakes. I will go through the solution again, carefully and slowly.
Alice has 4 brothers and 1 sister. This means Alice is also a sister to her brothers. Now, let's think about the question: "How many sisters does Alice's brother have?" Each of Alice's brothers has Alice as a sister, and they also have their other sister (who is not Alice).
So, each of Alice's brothers has 2 sisters: Alice and their other sister.
The correct answer is: Each of Alice's brothers has 2 sisters. Confidence level: 100%
I am absolutely confident in this solution because I have carefully considered the relationships between Alice and her siblings, and logically deduced the correct answer.

Figure 32: Example of a correct revision by LLaMA 3 70b. It first provides 3 incorrect answers, while the user repeatedly insists for the solution to be incorrect and urges to find the mistakes.

**Human**

Build SQLite queries for creation and population of the table that represents: "Alice has 4 sisters and she also has a brother. How many sisters in total does Alice's brother have?".
Given the table schema, query the table.
Emulate the previous query execution only by looking at SQL and provide the result of execution.
(Format every SQL query as ```sql {the query}```

**Mixtral-8x22B-instruct-v0.1**

To solve this problem, we need to create a table to represent the family relationships and then query the table to find the total number of sisters that Alice's brother has. Here's how you can do this:

1. Create a table named `Family` with columns `Name`, `Gender`, and `FamilyID`.

```sql
CREATE TABLE Family (
    Name TEXT,
    Gender TEXT,  -- M for Male, F for Female
    FamilyID INTEGER
);
```

2. Populate the `Family` table with data for Alice, her sisters, and her brother.

```sql
INSERT INTO Family (Name, Gender, FamilyID)
VALUES ('Alice', 'F', 1),
    ('Sister1', 'F', 1),
    ('Sister2', 'F', 1),
    ('Sister3', 'F', 1),
    ('Sister4', 'F', 1),
    ('Brother', 'M', 1);
```

3. Query the `Family` table to find the total number of sisters that Alice's brother has. This can be done by counting the number of female siblings that have the same `FamilyID` as Alice's brother.

```sql
SELECT COUNT(*)
FROM Family
WHERE Gender = 'F'
AND FamilyID IN (
    SELECT FamilyID
    FROM Family
    WHERE Name = 'Brother'
);
```

Assuming that the brother is not counted as a sister, the result of this query will be `4` because Alice's brother has the same 4 sisters as Alice.

Figure 33: Example of SQL reformulation of the AIW problem and correct solution for Mixtral 8x22B instruct v0.1. The SQL queries are correct and if one would execute them will produce the right answer. However the final answer generated by the model is still incorrect. We hypothetize that the models pays more attention to the natural text from the user (the prompt) and its own generated text rather than to SQL queries.

