# OpenReview forum: "Alice in Wonderland: Simple Tasks Reveal Severe Generalization and Basic Reasoning Deficits in State-Of-the-Art Large Language Models"
_ICLR.cc/2025/Conference — Submitted to ICLR 2025_

### Official Review · Reviewer_hx2d · 2024-10-29

**Soundness:** 3
**Presentation:** 3
**Contribution:** 3
**Rating:** 6
**Confidence:** 4

**Summary:**

Authors study reasoning capabilities of various SOTA LLMs in a controlled environment, by synthesizing a very simple yet efficient task, Alice in Wonderland (AIW). This task is composed of multiple variations of the following template: "Alice has N brothers and she also has M sisters. How many sisters does Alice’s brother have?". Authors systematically study more than 20 models by varying M/N, changing family relations, introducing redundant information, and varying between prompt templates. Authors showed that models not only fail on this simple task showing high variation between prompt templates, but also that their failure can not be attributed to arithmetic or commonsense knowledge errors, but occur due to the lack of generalization and basic reasoning abilities.

**Strengths:**

- Authors discuss an important problem of LLM reasoning abilities
- Paper is clearly written
- Proposed evaluation framework is novel and simple to implement and verify
- Authors perform extensive experiments across various models and prompt templates
- Detailed ablation studies on AIW variations support main claims of the paper

**Weaknesses:**

- Even though authors prove that SOTA LLMs fail on AIW task, I don't think we can claim that they are not capable of robust reasoning. On the contrary, paper shows that LLMs are capable of some types reasoning (like arithmetic, or basic family relations), but fail on the others (logical reasoning).
- There are multiple formatting issues in the paper, probably caused by moving text between templates, that hurt overall presentation of the paper (see Questions section for example).

**Questions:**

1. Why do you think Llama-3 performs so much worse then Llama-2 model? What framework did you use to run evaluations for models with  open weights? I wonder, if there might be any issues with prompts, as Llama-3 models require different special symbols in chat template than Llama-2.

2. There are some issues with citations across the paper where brackets are missing in most of the citations, for ex. in lines 41-42: "...visual recognition Radford et al. (2021) or language understanding Devlin et al.(2018); Raffel et al. (2020); Brown et al. (2020), l..." should be  "...visual recognition (Radford et al., 2021) or language understanding (Devlin et al., 2018; Raffel et al., 2020 ...".  Multiple periods are missing: lines 111, 117, 192, 309, 458, 460, 463.  Chapter 3.1.1 "original. like" -> "original, like" in multiple places.

---

> ### Author Response · Authors · 2024-11-17
>
> We thank the reviewer for going through our work. We appreciate comments on novelty of the evaluation, on value of extensive experiments, on simple reproducibility and on clarity of the writing.
>
> We further appreciate various points made by the reviewer and would like to address those in following:
>
> > Even though authors prove that SOTA LLMs fail on AIW task, I don't think we can claim that they are not capable of robust reasoning. On the contrary, paper shows that LLMs are capable of some types reasoning (like arithmetic, or basic family relations), but fail on the others (logical reasoning).
>
> Based on evidence from our work, we do not claim that LLMs are entirely incapable of reasoning (see also Sec.5, p.10). As shown by AIW Light control experiments (Fig. 3, 4, 5), where problem structure is simpler than AIW original, tested models are able to infer problem structure and successfully select and execute operations across AIW Light variations - handling basic family relations, elementary arithmetic and set operations, binding female attributes to entities (Alice, sisters) -  operations also required to solve AIW original. We do claim that all SOTA LLMs fail to generalize robustly and to perform robust reasoning. This is evident from strong performance fluctuations exhibited by all SOTA LLMs (Fig. 1, 2, 6) on AIW original variations that merely change numbers in otherwise the same fixed AIW problem template (Fig. B https://hackmd.io/_uploads/rJPFH38Gyl.png). Although problem structure and difficulty stays the same across AIW variations 1-4,  and same operations are required for AIW solution as for control AIW Light problems, correct response rates vary strongly across variations. Even for most advanced models like GPT-4, rates can jump wildly from 1 to 0 despite entirely unchanged problem structure and difficulty. This cannot be called robust reasoning, especially not given the simplicity of AIW problem that can be arguably easily handled by elementary school children. It also though cannot be called entire absence of reasoning, as correct responses are still present across variations, albeit with strongly different frequency.  We argue thus that what we observe is generic deficit in generalization consistently present in all SOTA LLMs, leading to lack of model robustness and fluctuations that cannot be explained alone by failures of specific low-level operations (required eg for arithmetical reasoning). Such failures would manifest equally across AIW variations - merely changing numbers N, M does not affect the way of operation execution, and fluctuations would not happen. Models claiming robust generalization and reasoning should be expected to solve simple problems like AIW across all variations with nearly 100% success rate without strong fluctuations, which is not what we observe. Open question for future work is what breaks problem structure inference and consequently proper selection and composition of low-level operations (like elementary arithmetics or set operations) to be executed already in such simple scenarios.
>
> > Why do you think Llama-3 performs so much worse then Llama-2 model? I wonder, if there might be any issues with prompts, as Llama-3 models require different special symbols in chat template than Llama-2.
>
> Llama-3 does not perform much worse than Llama-2 - they rather perform equally bad. Misleading here is to look only at average correct response rates (Fig. 1), where the average is over AIW variations 1-4 and all RESTRICTED, STANDARD and THINKING prompt types. There, it might seem Llama 2 70B (p=0.3) outperforms Llama 3 70B (p=0.05) (see also Suppl. Tab. 7). However, a glance at the full distribution of correct response rates reveals both models are equally bad in handling the problem (Fig. D https://hackmd.io/_uploads/S1QjnkuMyx.png; paper Fig. 2). Eg Llama 2 70B shows extreme lack of robustness - significant performance is shown only on single AIW variation 3, all others being 0 or close to 0 (important to emphasize again that variations do not change problem structure and difficulty). Due to that single outlier, average correct rate is significant, but robust problem handling is obviously broken. Thus, Llama 2 has the same severe deficits as Llama 3. Llama 3 behavior cannot be explained by requiring special instruction templates, as also without templates it handles very well all the AIW Light control problems (Fig. 3,4,5, also fluctuations vanish) and shows competitive performance to Qwen 2 72B on female boost version (Fig. 6). Prompts are available at https://anonymous.4open.science/r/AITW_anonymous-69A6/prompts/prompts.json, collected data https://anonymous.4open.science/r/AITW_anonymous-69A6/collected_responses/raw_data_inspection/AIW_AIW_plus.json
>
> > formatting of the citations is wrong, punctuation issues
>
> Thanks for helping to catch those. With regard to citations, using \cite instead of \citep caused havoc. We will correct all of that in the updated manuscript version.

---

> > ### Comment · Reviewer_hx2d · 2024-11-26
> >
> > Thanks for your detailed response. I will maintain my score.

---

> > > ### Author Response · Authors · 2024-11-28
> > > **Follow-up**
> > >
> > > We would like to thank the reviewer again for time and involvement in the discussions.
> > >
> > > As follow-up, we would like to emphasize that in contrast to testing models on a set of various problems that do not have much in common and where source of variations is not controlled (which is the usual case for standardized benchmarks. eg GSM8K, Fig. A https://hackmd.io/_uploads/HJULH2IG1l.png), we confront the models with instances of the same simple problem, which allows us to see how sensitive models are to perturbations that keep problem structure and difficulty unchanged and thus should NOT affect ability to cope with the problem IF generalization were intact. Performance fluctuations across problem structure and difficulty preserving variations observed in this experimental setup can thus provide direct evidence for the degree of generalization breakdown, additionally backed up by control experiments to rule out low level issues (Fig. E https://hackmd.io/_uploads/ByCpjM9M1x.png). To gather robust statistics, we execute many trials (> 30) for each AIW variation, estimating correct response rates for each variation from those multiple trials (Fig. J, https://hackmd.io/_uploads/B1MRvATzye.png) and also checking whether behavior is consistent independent of prompting conditions using 3 various prompt types (Fig. B, https://hackmd.io/_uploads/rJPFH38Gyl.png). See Fig. K https://hackmd.io/_uploads/rkPA9H-myg.png for overview.
> > >
> > > To confirm that the same observations hold for other simple problems of related kind, per reviewers’ request we conducted further additional experiments with AIW versions with modified problem templates that differ from AIW Original. For instance, we either introduce Alice and Bob as two entities in the problem structure to deal with, or we replace brothers and sisters entities with male/female friends, abandoning family specific frame. Using same experimental procedure to create variations of these problem versions, we observe the same pattern as for the AIW original, especially the strong fluctuations across variations, confirming the existence of the same generalization deficits for further problem examples (Fig. I https://hackmd.io/_uploads/BJ1nqj3MJx.png)
> > >
> > > We hope this amounts to convincing evidence that experimental setup we describe in our work is useful for community as systematically reproducible measurement method for falsification of the strong function hypothesis and for debunking overblown claims that rely on standardized benchmarks and overlook such clear deficits.
> > >
> > > We also would like to provide another illustrative example of such a debunking procedure on an example of recent case of NuminaMath-7B that was ranked 1st at the recent AIMO competition, solving 29/50 private set problems of olympiad math level (https://huggingface.co/AI-MO/NuminaMath-7B-TIR). Based on that evals, the claim was widely put forward that the model is capable of solving high school olympiad math problems (https://www.aimodels.fyi/models/huggingFace/numinamath-7b-tir-ai-mo). AIW problem has average elementary school level and does not require any advanced math knowledge. We tested NuminaMath-7B on AIW and observed a strong collapse of this model on AIW problem, with correct response rates close to 0 across AIW variations 1-4. Using AIW Light control problems, we can also see that NuminaMath-7B can handle all the low level operations and knowledge required to deal with family structure, ruling out that those are the issues. Using the AIW problem setting, we thus can contradict the strong claim of being capable to robustly deal with math problems (Fig F, https://hackmd.io/_uploads/SybG2hqz1x.png). Especially, breakdown in such a simple setting rules out that model will be able to deal robustly with olympiad level math tasks, debunking the claim and raising further questions about AIMO benchmark procedure used to measure model performance.
> > >
> > > For the collected data for this debunking experiment, see anonymous repo https://anonymous.4open.science/r/AITW_anonymous-69A6/collected_responses/raw_data_inspection/NuminaMath-7B_AIW_versions_problem_set_checked.json

---

### Official Review · Reviewer_5S6M · 2024-11-02

**Soundness:** 3
**Presentation:** 3
**Contribution:** 3
**Rating:** 6
**Confidence:** 3

**Summary:**

This paper demonstrate a dramatic breakdown of generalization and basic reasoning of all SOTA models (ncluding advanced models like GPT-4 or Claude 3 Opus) which claim strong function, using a simple, short, conventional common sense problem formulated in concise natural language (AIW problem). The authors observe that large language models (LLMs) exhibit significant performance fluctuations on simple problems across minor variations that should not impact problem-solving ability at all. Additionally, various standard interventions, such as chain-of-thought prompting, failed to yield correct solutions in the AIW problem. These observations highlight the need to re-evaluate the claimed capabilities of the current generation of LLMs.

**Strengths:**

* This paper is well-written and presents clear ideas.

* The authors conduct extensive experiments on over 36 LLMs to demonstrate the breakdown of SOTA LLMs on the somple AIW problem.

* Fully Open-sourced code and data to reproduce the result.

**Weaknesses:**

* The problem setting of AIW has certain interfering factors. After some attempts, I found that the main reason LLMs perform poorly on AIW origin is due to easy thinking. For example, "Alice has 3 brothers, and each of these brothers has the same sisters, who are Alice's sisters. Alice has 6 sisters, so each of her brothers has 6 sisters." The issue here is that the LLM overlooks counting Alice herself, rather than lacking reasoning ability. I believe that testing similar problems in mathematical reasoning tasks (like GSM8K or MATH) would be more convincing.

* Quite a few typo errors. line 016, few-show -> few-shot; format of most of the citations is wrong.

**Questions:**

See weaknesses.

Recently, I came across another paper that is similar in content to this study. I know this paper was published online prior to [1],  I'm just curious about what advantages the authors believe the AIW dataset presented in this paper has compared to [1], which focuses on the mathematical reasoning task.

[1] GSM-Symbolic: Understanding the Limitations of Mathematical Reasoning in Large Language Models

---

> ### Author Response · Authors · 2024-11-17
>
> We thank the reviewer for the time dealing with our work. We appreciate reviewer’s remarks on the paper being well written, and also pointing to the clarity of the ideas presented and the reproducibility of the findings by open-sourcing data and code.
>
> We also appreciate various points made by the reviewer and would like to address those in following:
>
> > The problem setting of AIW has certain interfering factors … I believe that testing similar problems in mathematical reasoning tasks (like GSM8K or MATH) would be more convincing.
>
> We think that problems formulated in GSM8K and MATH have on the contrary lengthier, overloaded formulations and thus a lot more interfering factors than the simple AIW problem (Fig. A, https://hackmd.io/_uploads/HJULH2IG1l.png ; Suppl. Fig. 16, 17 for bench failures). Main goal of our study was to find a “minimal” problem to convincingly falsify the hypothesis that current SOTA LLMs possess strong generalization and robust reasoning in zero-shot regime. Our intention was thus to reduce problem complexity as much as possible compared to standardized benchmarks and come up with a simple, short problem, unambiguously expressible in natural language, which despite its simplicity still could be used to stress test the core model capability for zero-shot generalization and reasoning. Contrary to problems used in GSM8K and MATH, AIW problem is a much simpler, short, common sense math problem that can be arguably easily solved by elementary school children. As we obtain strong evidence that all tested SOTA LLMs, including most advanced ones, cannot handle variations of that simple problem robustly and exhibit strong performance fluctuations, with correct response rate going from 1 to 0, despite variations being just different number instantiations (Fig. B, https://hackmd.io/_uploads/rJPFH38Gyl.png; paper Fig 1, 2), we can convincingly falsify the hypothesis without relying on more complex problems from GSM8K or MATH. Simple AIW structure also allows us to conduct control experiments (Fig. 3, 4, 5), ruling out failure in elementary operations as cause of the breakdown, which is unclear how to perform with GSM8K or MATH problems.
>
> > The issue here is that the LLM overlooks counting Alice herself, rather than lacking reasoning ability
>
> While we observe various occasional failing operations that lead to wrong AIW solutions (and we agree that overlooking counting Alice herself is one of those), we think that we can convincingly rule out low-level operation failure like those as the main cause of the observed breakdown. We obtain clear evidence for strong performance fluctuations on problem irrelevant AIW variations 1-4 (instantiating different brothers and sisters numbers N,M) that all tested SOTA LLMs consistently exhibit (paper Fig 2, 6, Suppl. Fig 10; Fig. B https://hackmd.io/_uploads/rJPFH38Gyl.png). These fluctuations are impossible to explain when assuming the same low-level operations behind the performance  - eg, failure to count Alice should then affect all variations the same way, as merely changing numbers N, M does not change the way of counting. Another evidence is obtained by our AIW Light control experiments (Fig. 3, 4, 5), where problems are constructed to contain operations that are also required to solve AIW original, though asking different questions to make the problem further simpler. We see fluctuations largely disappear, and models collapsing on AIW original show high correct response rates across all variations. Thus, we can prove low-level operations - handling family relations, binding female attributes, elementary arithmetics like counting - are actually intact. Remaining hypothesis is then generic generalization and basic reasoning deficits that result in failure to robustly infer problem structure. This explains the strong performance difference across different AIW variations despite the problem structure being the same, assuming different operations are inferred and executed.
>
> > AIW and GSM-Symbolic comparison
>
> GSM-Symbolic (GSM-S) work follows our method to use problem templates and to introduce variations, measuring then how sensitive models are wrt. to variations with the same intention - drawing conclusions from observed model robustness or lack thereof about generalization and reasoning capabilities. We think GSM-S went the right way, however the evidence obtained there is rather weak to claim severe deficits like done in AIW (Fig. C, https://hackmd.io/_uploads/Sk7aBh8Gyx.png). There are two clear differences: 1) in our work, we observe strong fluctuations even for most advanced models like GPT-4, with scores going up to 1 and down to 0 across variations. GSM-S sees only weak fluctuation, eg. for GPT-4o being around 0.07. 2) we see many models collapsing to very low rates, while in GSM-S rates are still high. Eg Llama-3-8B scores $< 0.15$ on AIW, while $>0.69$ on GSM-S. Thus, while AIW provides clear breakdown evidence, evidence on GSM-S is inconclusive.

---

> > ### Author Response · Authors · 2024-11-17
> > **Rebuttal response 2**
> >
> > > typos, format of most of the citations is wrong
> >
> > Thanks for catching those. With regard to citations, seems using \cite instead of \citep caused havoc. We will correct all of that in the updated manuscript version.

---

> > ### Comment · Reviewer_5S6M · 2024-11-22
> >
> > Thanks for your detailed response. I will maintain my score.

---

> > > ### Author Response · Authors · 2024-11-28
> > >
> > > We would like to thank the reviewer again for time and involvement in the discussions.
> > >
> > > As follow-up, we would like to emphasize that in contrast to testing models on a set of various problems that do not have much in common and where source of variations is not controlled (which is the usual case for standardized benchmarks. eg GSM8K, Fig. A https://hackmd.io/_uploads/HJULH2IG1l.png), we confront the models with instances of the same simple problem, which allows us to see how sensitive models are to perturbations that keep problem structure and difficulty unchanged and thus should NOT affect ability to cope with the problem IF generalization were intact. Performance fluctuations across problem structure and difficulty preserving variations observed in this experimental setup can thus provide direct evidence for the degree of generalization breakdown, additionally backed up by control experiments to rule out low level issues (Fig. E https://hackmd.io/_uploads/ByCpjM9M1x.png). To gather robust statistics, we execute many trials (> 30) for each AIW variation, estimating correct response rates for each variation from those multiple trials (Fig. J, https://hackmd.io/_uploads/B1MRvATzye.png) and also checking whether behavior is consistent independent of prompting conditions using 3 various prompt types (Fig. B, https://hackmd.io/_uploads/rJPFH38Gyl.png). See Fig. K https://hackmd.io/_uploads/rkPA9H-myg.png for overview.
> > >
> > > To confirm that the same observations hold for other simple problems of related kind, per reviewers’ request we conducted further additional experiments with AIW versions with modified problem templates that differ from AIW Original. For instance, we either introduce Alice and Bob as two entities in the problem structure to deal with, or we replace brothers and sisters entities with male/female friends, abandoning family specific frame. Using same experimental procedure to create variations of these problem versions, we observe the same pattern as for the AIW original, especially the strong fluctuations across variations, confirming the existence of the same generalization deficits for further problem examples (Fig. I https://hackmd.io/_uploads/BJ1nqj3MJx.png)
> > >
> > > We hope this amounts to convincing evidence that experimental setup we describe in our work is useful for community as systematically reproducible measurement method for falsification of the strong function hypothesis and for debunking overblown claims that rely on standardized benchmarks and overlook such clear deficits.
> > >
> > > We also would like to provide another illustrative example of such a debunking procedure on an example of recent case of NuminaMath-7B that was ranked 1st at the recent AIMO competition, solving 29/50 private set problems of olympiad math level (https://huggingface.co/AI-MO/NuminaMath-7B-TIR). Based on that evals, the claim was widely put forward that the model is capable of solving high school olympiad math problems (https://www.aimodels.fyi/models/huggingFace/numinamath-7b-tir-ai-mo). AIW problem has average elementary school level and does not require any advanced math knowledge. We tested NuminaMath-7B on AIW and observed a strong collapse of this model on AIW problem, with correct response rates close to 0 across AIW variations 1-4. Using AIW Light control problems, we can also see that NuminaMath-7B can handle all the low level operations and knowledge required to deal with family structure, ruling out that those are the issues. Using the AIW problem setting, we thus can contradict the strong claim of being capable to robustly deal with math problems (Fig F, https://hackmd.io/_uploads/SybG2hqz1x.png). Especially, breakdown in such a simple setting rules out that model will be able to deal robustly with olympiad level math tasks, debunking the claim and raising further questions about AIMO benchmark procedure used to measure model performance.
> > >
> > > For the collected data for this debunking experiment, see anonymous repo https://anonymous.4open.science/r/AITW_anonymous-69A6/collected_responses/raw_data_inspection/NuminaMath-7B_AIW_versions_problem_set_checked.json

---

### Official Review · Reviewer_NZax · 2024-11-02

**Soundness:** 2
**Presentation:** 3
**Contribution:** 1
**Rating:** 3
**Confidence:** 4

**Summary:**

This paper tests SOTA LLMs’ reasoning abilities by testing them with a bunch of variants of the AIW problem.

**Strengths:**

1. The tests are across different SOTA LLMs.

**Weaknesses:**

1. The whole paper is about one type of questions:  “Alice has N brothers and she also has M sisters. How many sisters does Alice’s brother have”. I personally feel like it is hard to judge model’s capabilities based on one type of question alone. Model’s generalization and reasoning abilities are maybe on a spectrum and with only one question, it is hard to tell where the model falls on this spectrum.

2. GPT-4o has superior performance, possible suggesting this might have to do with model size or training?

3. I don’t consider “female boost” as totally redundant information. For one thing, if you are testing the model’s reasoning abilities, it should disentangle model’s syntactic understanding as a separate thing. “She” as a sole indicator of Alice being a female is more a syntactic problem, which shouldn’t part of model’s burden if one’s goal is simply to test reasoning abilities.

4. Personally, I feel like this paper adds nothing significantly interesting to the existing discussion on whether LLM can reason or generalize. For one thing, a pure test on reasoning should not rely much on extra knowledge. The test in the paper (AIW) needs model’s understanding of “she” as Alice (syntactic) and basic family structure (external knowledge). The actual reasoning, on the other hand, is in my opinion, perhaps not the main bottleneck. This is also supported in the paper where “female boost” variants can improve performance.

**Questions:**

1. For figure 1, what do the numbers like 55, 56, 63, 69 mean?

---

> ### Author Response · Authors · 2024-11-19
>
> We thank the reviewer for the comprehensive feedback raising number of points, which would like to address those in following:
>
> > 1. The whole paper is about one type of questions … hard to judge model’s capabilities based on one type of question alone.
>
> Main goal of our paper is to provide convincing evidence for the falsification of the hypothesis stating current SOTA LLMs possess robust generalization and reasoning which is backed up by current standardized benchmarks. To do this, we follow the scientific method and search for a “minimal” experiment/problem - as simple as possible to already provide sufficient evidence -  to convincingly falsify the strong function hypothesis. AIW problem satisfies this requirement - it is so simple that it can be arguably solved by elementary school children, having short, concise, unambiguous formulation in natural language (Fig. A https://hackmd.io/_uploads/HJULH2IG1l.png, Suppl. Tab. 2). Any system that claims robust zero-shot generalization and basic reasoning should be able to solve it across variations in numbers in the problem template with correct response rate close to 100%. As advanced SOTA LLMs like GPT-4/4o or Claude Opus/Sonnet claim generalization and reasoning on high school or even PhD level problems, observed lack of robustness and strong performance fluctuation across variations of such a simple problem (Fig. B https://hackmd.io/_uploads/rJPFH38Gyl.png; paper Fig 1,2,6) clearly disproves the claim, also revealing the deficits in standardized benchmarks to detect such clear generalization failures (Fig 7, Suppl. Fig. 16,17). Goal of hypothesis falsification is thus fulfilled by using one problem type that delivers sufficient evidence.
>
> > 2. GPT-4o superior performance, model size or training?
>
> We see clear effect of pre-training scale on the observed AIW performance. The only stronger performers that show correct response rates > 0.3 across variations, are large scale pretraining models like GPT-4 and Claude Opus (Fig. 1; Suppl. Tab. 7). All those stronger performers suffer though from strong fluctuations across AIW variations (Fig 2, 6). Models at smaller scales, eg Llama 3 8B, are staying well below 0.2, most of them residing close to 0 or collapsing entirely to 0 across all variations (Suppl. Fig. 8).
>
> > 3. I don’t consider “female boost” as totally redundant information. … “She” as a sole indicator of Alice being a female is more a syntactic problem, which shouldn’t part of model’s burden …
>
> We agree that to solve the AIW problem, SOTA LLMs have to infer the problem relevant information from natural language description and this requires also handling properly handling language syntax. Either using “Alice is female” or using “she” conveys same information - Alice being female - if language handling is successful, therefore we state that adding “Alice is female” does NOT add new information to natural language problem description. To rule out that such low level language parsing/handling issues are the problem behind observed failures,  we conducted control experiments using AIW Light Arithmetic Total Girls problem version (Sec. 2.1, 3.1.1; Fig. E https://hackmd.io/_uploads/ByCpjM9M1x.png), which tests whether binding of female attribute via “she” is handled properly. We see most models that suffer clear breakdown on AIW solve AIW Light successfully, with high correct response rates and - importantly - vanishing fluctuations (Fig. 5). Thus, “female boost” (Fig. 6) is evidence for performance change due to entirely redundant information, again pointing to lack of model robustness and generalization deficit.
>
> > 4. … paper adds nothing significantly to the existing discussion on whether LLM can reason or generalize. … a pure test on reasoning should not rely much on extra knowledge. The AIW test ... needs model’s understanding of “she” as Alice (syntactic) and basic family structure (external knowledge). The actual reasoning … is in my opinion … not the main bottleneck ... also supported in the paper where “female boost” can improve performance.
>
> AIW problem is constructed exactly in a way to minimize the amount of knowledge necessary to handle it - in contrast to problems overloaded with various contexts used in standardized benchmarks (Fig. A). Using AIW Light control experiments (Sec. 2.1, 3.1.1), we prove that handling basic family structure, binding female attributes via “she”, elementary arithmetics like counting - are all intact (Fig. E https://hackmd.io/_uploads/ByCpjM9M1x.png; paper Fig. 3,4,5), ruling out that this additional knowledge poses an issue. Given the models show the ability to cope successfully with all required knowledge and operations to solve AIW, remaining hypothesis assumes generic generalization deficits responsible for observed strong fluctuations on AIW. “Female boost” is in line with that - behavior changes despite just adding redundant information, with strong fluctuations across variations still persisting (Fig. 6).

---

> > ### Author Response · Authors · 2024-11-19
> > **Rebuttal Continuation, 2**
> >
> > One main contribution of ours to existing discussion on whether LLM can reason or generalize is thus novel experimental method to detect generalization breakdown by measuring model sensitivity to problem irrelevant variations in a fixed, minimalistic problem template. This provides a tool to properly check whether a given model can indeed generalize well or whether claims are overblown by relying on high scores of standardized benchmarks that obviously do not manage to reflect such deficits properly (see Fig. 7, Suppl. Fig. 16, 17, 18, 19 for bench failures).
> >
> > Another contribution is providing clear evidence that also most advanced SOTA LLMs do not stand this test for robust zero shot generalization. This sends a strong warning  to technological community to put too much trust in existing benchmarks and to put models exhibiting high scores on those benchs to end applications that require robust generalization and reasoning under various conditions. It is clear that if strong fluctuations and sensitivity is exhibited even on such a simple problem as AIW, this will be only one problem of many that leads to strong sensitivity and lack of robustness to problem variations, and more complex problems will expectedly lead to even stronger lack of robustness. Our research shows that installing robust generalization is still an open question for basic research, and provides a guide for constructing simple, well reproducible measurement devices to probe generalization, paving the road for benchmarks that do measure this core property correctly and allow for measurable progress towards models with stronger generalization.
> >
> > > For figure 1, what do the numbers like 55, 56, 63, 69 mean?
> >
> > For figures that show correct response rates across AIW variations 1-4, the numbers in the legend mean the prompt IDs used in the experiments, for better reproducibility. Specifically for Fig. 1 inlay, correct response rates for each AIW variation 1-4 on STANDARD prompt type are shown, corresponding to prompt IDs 55, 56, 63, 69.  Prompts are available at https://anonymous.4open.science/r/AITW_anonymous-69A6/prompts/prompts.json, collected data https://anonymous.4open.science/r/AITW_anonymous-69A6/collected_responses/raw_data_inspection/AIW_AIW_plus.json

---

> > > ### Comment · Reviewer_NZax · 2024-11-22
> > >
> > > Thanks for the rebuttal. But my concern remains. It's hard to judge a model's reasoning capabilities from one type of tests. It just doesn't make sense to me. You can always find something that the model is bad at. To evaluate reasoning, one has to look at the performance on average, otherwise the results hold no statistical value.
> > >
> > > To put this in another way, if I test human a bunch of tricky questions, and they get one question wrong, I wouldn't argue that human are fundamentally flawed at reasoning.

---

> > > > ### Author Response · Authors · 2024-11-30
> > > > **Follow-up**
> > > >
> > > > Dear reviewer,
> > > >
> > > > we would like to thank again for the time dealing with our work. As the author-reviewer discussion comes near its end, we would like to draw attention to the ongoing discussion (which seems still unfinished to us) where we attempted to resolve what we think might be a misunderstanding how our work proceeded in obtaining evidence for generalization breakdown of current SOTA LLMs. We are curious whether it helped to illuminate the raised issues and looking forward any response. Should this discussion be insightful and resolve some of the concerns, please consider reflecting this in the scores.

---

> ### Author Response · Authors · 2024-11-22
>
> Thanks for the swift response. We think there is still a general misunderstanding how our work proceeds to demonstrate severe generalization deficit, as evident from the arguments in the response:
>
> > If I test human a bunch of tricky questions, and they get one question wrong, I wouldn't argue that human are fundamentally flawed at reasoning.
>
> This is an entirely wrong analogy wrt. to our work, and we agree that works using such an approach would not be able to draw proper conclusions. We do not have an arbitrary bunch of tricky questions. We have an opposite situation. We have one simple question which can be arguably handled easily by elementary school children. We then generate variations using the question template, such that variations are natural for the posed problem and none of them changes the problem structure and difficulty. Thus, the bunch consists of problem instances generated from same simple problem where merely numbers are varied in the corresponding placeholder variables. We estimate correct response rate from many trials (at least 30) for each of problem instances in the bunch, such that we obtain correct response rate for each of AIW variations 1-4 (thus here, "get one question wrong" notion is actually not appropriate - we can only speak about lower or higher correct response rates over many trials). Puzzling is here exactly why there should be ANY difference in performance across such variations at all, as rate of getting "one question wrong", or right, across many executed trials should be roughly equal for all variations as they pose the very same problem  - IF generalization were intact.
>
> If we imagine how human would perform on such a task, we definitely DO NOT expect that their performance to solve such a bunch would wildly vary between very high and very low across variations (Fig. J https://hackmd.io/_uploads/B1MRvATzye.png). Imagine a test with human probands where a person is confronted with these variations of the AIW in many repetitive trials
>
> > Variation 1: Alice has 3 brothers and she also has 6 sisters. [Correct answer: 7 ] \
> > Variation 2: Alice has 2 sisters and she also has 4 brothers. [Correct answer: 3 ] \
> > Variation 3: Alice has 4 sisters and she also has 1 brother.   [Correct answer: 5 ] \
> > Variation 4: Alice has 4 brothers and she also has 1 sister.   [Correct answer: 2 ] \
> > \
> >  How many sisters does Alice’s brother have?
>
> Arguably, it would be shocking to observe persons having consistently over many trials close to 100% correct responses on Variation 4, while entirely breaking down on Variation 3 close to 0 (Fig. J; paper Fig. 2), as variations actually pose the very same simple problem without altering problem structure, only difference being in instantiated numbers. That is because human would be able to generalize, extracting the actual problem structure behind the bunch, and thus solve the problem independent of instantiated numbers equally well. This is how we also tested generalization of the SOTA LLMs, and see them consistently failing (Fig. B https://hackmd.io/_uploads/rJPFH38Gyl.png; paper Fig. 1,2,6), which we think given all the persisting claims about those models successfully handling math problems at olympiad level equally shocking and points to clear flaws not only in models, but also in benchmarks.
>
> > To evaluate reasoning, one has to look at the performance on average, otherwise the results hold no statistical value.
>
> In our work, we test generic kind of reasoning - ability to infer underlying problem structure, which enables generalization and thus handling problems despite variations in their formulation. It matters which average to take to test this. Looking at performance averaged over all variations would be misleading, as breakdown of generalization is manifested in models' sensitivity to variations that is NOT visible if averaging over all variations. Eg, as evident in Fig. 1, 2, average correct response rates of better performers like GPT-4/4o or Claude 3 Opus, can be substantial. Only when looking at correct response rates of each variation separately, it becomes apparent that such average results from high performance on some, and low or very low close to 0 performance on other variations - despite variations being instances of the same problem (Fig. J https://hackmd.io/_uploads/B1MRvATzye.png). This breakdown in generalization would remain undetected if averaging over variations, and this is what also happens with standardized benchmarks that do not look into measuring performance on variations of problem instances (Fig. 7, Suppl. Fig. 16,17,18,19), stating high performance across static questions and creating misleading illusion of strong function where there are clear deficits. We thus average over right/wrong responses for each variation separately (>30 trials), obtaining statistics that allows conclusion about model sensitivity to variations, pointing to robustness or lack thereof (see again Fig. J for instructive overview).

---

> ### Author Response · Authors · 2024-11-28
> **Additional experiments measuring generalization breakdown using further problem examples**
>
> We would like to emphasize again that our work introduces a measurement procedure for model generalization that makes use of the same simple problem template to generate problem structure and difficulty preserving variations (AIW variations 1-4; see Fig. K for overview https://hackmd.io/_uploads/rkPA9H-myg.png). Thus, we confront the models with instances of the same simple problem, which allows us to see how sensitive model is to problem perturbations that should NOT affect ability to cope with the problem IF generalization is intact. It is in contrast to testing models on a set of various problems that do not have much in common and where source of variations is not controlled, which is the usual case for standardized benchmarks (eg GSM8K, Fig. A https://hackmd.io/_uploads/HJULH2IG1l.png). To gather robust statistics, we execute many trials (> 30) for each such variation, estimating correct response rates for each variation from those multiple trials (Fig. J, https://hackmd.io/_uploads/B1MRvATzye.png). Amount of existing fluctuations in correct response rates across the variations reveals then model sensitivity/robustness and enables conclusions about generalization or its breakdown.
>
> To further strengthen evidence that technique of introducing problem structure \& difficulty preserving variations into same simple problem template can be used to measure generalization breakdown, we executed per reviewers request a number of additional experiments using various problem examples that we think provide conclusive evidence that measurement procedure works independent of a chosen problem type.
>
> To confirm that the same observations hold for other simple problems of related kind, we conducted additional experiments with AIW versions with modified problem templates that differ from AIW Original. For instance, we either introduce Alice and Bob as two entities in the problem structure to deal with, or we replace brothers and sisters entities with male/female friends, abandoning family specific frame. Using same experimental procedure to create variations of these problem versions, we observe the same pattern as for the AIW original, especially the strong fluctuations across variations, confirming the existence of the same generalization deficits for further problem examples (Fig. I https://hackmd.io/_uploads/BJ1nqj3MJx.png)
>
> We hope this amounts to convincing evidence that experimental setup we describe in our work is useful for community as systematically reproducible measurement method for falsification of the strong function hypothesis and for debunking overblown claims that rely on standardized benchmarks and overlook such clear deficits.
>
> We also provide another illustrative example of such a debunking procedure on an example of recent case of NuminaMath-7B that was ranked 1st at the recent AIMO competition, solving 29/50 private set problems of olympiad math level (https://huggingface.co/AI-MO/NuminaMath-7B-TIR). Based on that evals, the claim was widely put forward that the model is capable of solving high school olympiad math problems (https://www.aimodels.fyi/models/huggingFace/numinamath-7b-tir-ai-mo). AIW problem has average elementary school level and does not require any advanced math knowledge. We tested NuminaMath-7B on AIW and observed a strong collapse of this model on AIW problem, with correct response rates close to 0 across AIW variations 1-4. Using AIW Light control problems, we can also see that NuminaMath-7B can handle all the low level operations and knowledge required to deal with family structure, ruling out that those are the issues. Using the AIW problem setting, we thus can contradict the strong claim of being capable to robustly deal with math problems (Fig F, https://hackmd.io/_uploads/SybG2hqz1x.png). Especially, breakdown in such a simple setting rules out that model will be able to deal robustly with olympiad level math tasks, debunking the claim and raising further questions about AIMO benchmark procedure used to measure model performance.
>
> (For the collected data for this debunking experiment, see anonymous repo https://anonymous.4open.science/r/AITW_anonymous-69A6/collected_responses/raw_data_inspection/NuminaMath-7B_AIW_versions_problem_set_checked.json)

---

### Official Review · Reviewer_3mYn · 2024-11-03

**Soundness:** 2
**Presentation:** 3
**Contribution:** 2
**Rating:** 5
**Confidence:** 4

**Summary:**

The authors discovered a surprisingly simple and concise problem that makes most LLMs, including state-of-the-art models, fail. The problem, namely AIW, belongs to the class of basic reasoning problems where humans excel. The authors show that LLMs fail on the vanilla version of the AIW problem and semantically equivalent variations; techniques such as chain-of-thoughts or more advanced prompting methods fail at mitigating such issues.

**Strengths:**

I am really surprised that such a simple example makes LLMs fail, so I consider this discovery valuable. I tried to prompt a few models, and I agree with the authors that this problem is indeed hard even for models like GPT-4 (though I tried a few times with GPT-o1-preview, and it correctly solves the task, but it surprisingly fails to reply with nonsense when we input negative numbers!).

The article is well written and easy to follow, and the AIW results are robust enough to support the claim that most LLMs fail on such problems.

**Weaknesses:**

The authors did not try to add illustrations (e.g., k-shot) to mitigate the issue. I tried to add an illustration, but some models still failed. That analysis would add value to the work. While more expensive, fine-tuning a small model would also add value to the consistency of the case study they present.

Beyond that, my biggest concern is that the authors tried and reported only one example of failure across multiple models.
To make an analogy with the adversarial robustness literature, this is equivalent to finding a single adversarial example in computer vision that makes most models misclassify an input (an example of a ‘universal trigger’).
The paper lacks a consistent analysis of other examples, and, in this form, it reduces the contribution to an exciting yet anecdotal showcase of failure. Plenty of articles show failure cases of LLMs on *many* examples and variations; one very popular is [1].

Furthermore, the authors do not provide a solution or a tentative plan to mitigate the problem (but that is not necessarily a limitation).
The authors do not give a reasonable rationale behind why LLMs fail on the AIW problem. Interestingly, it points out that LLMs fail on the AIW Light Arithmetic Total Girls, but that is another anecdotal showcase of failure that does not tell us much about why LLMs fail on such simple problems.
For example, a model that solves the vanilla AIW would possibly “create” a graphical representation of Alice and her brothers and sisters; then, the LLM can count the number of edges from one of the brothers connected to all the sisters. That means a sufficient (but not necessary) condition to solve the problem is being able to perform 2-hops reasoning and counting (from one of the brothers to all the sisters). LLMs seem to lack such capability.

[1] Large Language Models Fail on Trivial Alterations to Theory-of-Mind Tasks, Tomer Ullam.

**Questions:**

1) What happens when N and M grow larger than 7, and why do they decide to set that as the upper bound on such variables?

2) Have the authors tried with floating and/or negative values for N and M? If a model still replies with the consistent (yet wrong) reasoning, that is a strong hint a model does not understand the task under consideration (i.e., it cannot connect numerical and graphical reasoning with family relationships).

3) Have the authors tried to ask the model to generate a graphical representation of Alice’s family and then solve the task?

4) Why do the authors focus on a single example and not on a consistent range of variations of similar problems?

**Details Of Ethics Concerns:**

No ethical concerns.

---

> ### Author Response · Authors · 2024-11-21
>
> We thank the reviewer for dealing with our work and taking the time to try out the experiments with the AIW problem and test the findings. We appreciate positive feedback emphasizing the surprising nature of the discovery being valuable, the robustness of the findings across various models and well-written text easy to follow. We also appreciate further points raised, which we address in following:
>
> > The authors did not try to add illustrations (e.g., k-shot) to mitigate the issue. I tried to add an illustration, but some models still failed. That analysis would add value to the work. While more expensive, fine-tuning a small model would also add value to the consistency of the case study they present.
>
> In our work, we focus on zero-shot generalization as an important mode of foundation/instruction fine-tuned model operation as it reveals a lot about its core capabilities to handle novel scenarios. Therefore, we leave the very interesting questions of either few-shot in-context learning or few-shot fine-tuning for future work. We were though too curious and ran preliminary experiments with few shot in-context learning. There, we were presenting few shots of solved AIW problem variations, posing then an unseen AIW variation as test problem. We observe that models often discover and stick with the “shortcut” solution C = M + 1 (M being number of Alice’s sisters) without learning to extract the true underlying problem structure. This becomes visible when switching in the test problem the question such that correct answer is not anymore obtainable by adding 1. Models often still respond to the altered test question by just adding 1 to the queried variable (Fig. G https://hackmd.io/_uploads/SyvWuj3fkx.png). This preliminary evidence hints that in-context few shot learning may have the same issues with obtaining strong robust generalization as observed for zero-shot case. Furthermore, we think that given the widespread claims of advanced models like GPT-4/4o, Claude 3 Opus  to possess strong generalization and reasoning, such an embarrassingly  simple problem as AIW should be handled with ease in zero-shot mode, were the claim to be defended.
>
> > … biggest concern is that the authors tried and reported only one example of failure across multiple models. To make an analogy with the adversarial robustness literature, this is equivalent to finding a single adversarial example in computer vision that makes most models misclassify an input (an example of a ‘universal trigger’).
>
> We think that the analogy with the adversarial examples is correct only in one sense - adversarial examples also reveal lack of model robustness to variations that should not affect model function/performance, pointing to generalization deficits. However, our work is not of an adversarial kind. We do not start from a simple, solvable problem and probe various tweaks that do not correspond to “natural” problem variations to search for a problem input alteration that breaks the model. On the contrary, variations introduced into AIW problem template are simple, “natural” variations entirely in the sense of the problem structure, corresponding to instantiation of various numbers which leave problem structure and difficulty unchanged (Fig. A https://hackmd.io/_uploads/HJULH2IG1l.png). Those naturally generated problem instances do not contain any backdoors or tweaks aiming to trigger model glitches. We additionally make use of AIW Light control experiments to make sure that observed breakdowns and strong fluctuations do not origin in problem formulation specific issues, eg failing to parse language/numbers or to execute required low-level operations or access specific knowledge necessary for problem solution (Fig. E https://hackmd.io/_uploads/ByCpjM9M1x.png; Fig 3,4,5). Thus, our approach is almost the opposite of adversarial, as we attempt to make sure that problem formulations and variations are actually easy for the models to handle.
>
> Further, to confirm that same observations hold for other simple problems of related kind, we conducted experiments with various AIW versions. For instance, we introduce Alice and Bob as two entities in the problem structure to deal with, or we replace brothers and sisters entities with male/female friends, abandoning family specific frame. Using same experimental procedure to create variations of these problem versions, we observe same pattern as for AIW original, especially the strong fluctuations across variations, confirming existence of same generalization deficits for further problem examples (Fig. I https://hackmd.io/_uploads/BJ1nqj3MJx.png)

---

> > ### Comment · Reviewer_3mYn · 2024-11-26
> > **Response to comment 1**
> >
> > I only agree partially with the second point mentioned above. The analogy with adversarial robustness is not entirely correct, I agree, but the notion of adversarial trigger is not necessarily connected with a variation of existing input points. See [1], for example.
> > As regards the first point, adding experiments with illustrations can cause the model to find a 'shortcut': nonetheless, it would have been an interesting observation to make. But I understand your motivation.
> > [1] Universal Adversarial Triggers for Attacking and Analyzing NLP

---

> ### Author Response · Authors · 2024-11-21
> **Rebuttal Continuation, 2**
>
> > The paper lacks a consistent analysis of other examples, and, in this form, it reduces the contribution to an exciting yet anecdotal showcase of failure.
>
> We respectfully disagree with notion that failures that we study are anecdotal. Contrary to various indeed anecdotal examples, we conduct systematic evaluation of correct response rates of various SOTA models across variations while controlling variation source (Fig 1,2,6). We also execute control experiments to rule out various low level issues as source of observed failures and fluctuations (Fig. E https://hackmd.io/_uploads/ByCpjM9M1x.png; Fig 3,4,5). The failures and exact full statistics of model behavior can be reproduced following our approach of using fixed problem template while introducing problem compatible variations, which lead to strong fluctuations.  As our main goal was to falsify the still widespread hypothesis stating current SOTA LLMs possess robust generalization and reasoning, which is usually backed up by current standardized benchmarks (for bench failures, see Fig. 7, Suppl. Fig. 16,17,18,19) , we searched for a minimal sufficient problem setting that will allow such falsification. AIW problem that is made so simple that arguably elementary school children can easily handle it (Fig. A https://hackmd.io/_uploads/HJULH2IG1l.png, Suppl. Tab. 2), allowed us to obtain such evidence falsifying the strong function claim, as it would be expected from models with even basic generalization and reasoning capabilities to solve AIW without strong fluctuations with correct response rates close to 100% across all variations. Given this clear evidence, no further other examples were necessary.
>
> > Plenty of articles show failure cases of LLMs on many examples and variations; one very popular is [1].
>
> In line with preceding response, to our knowledge, ours is the first study that systematically quantifies severe lack of model robustness pointing to generalization deficit in a simple, well reproducible scenario, gathering statistics executing experiments across many repeated trials (at least 30 per each variation and prompt type combination, Suppl. Fig. 20), controlling for conditions and source of variations, obtaining evidence consistently  across many SOTA LLMs including the most advanced large scale ones, and using control experiments to rule out various low level causes like natural language/numbers parsing, tokenization, problem specific ambiguities, failures to access specific knowledge (Fig. E https://hackmd.io/_uploads/ByCpjM9M1x.png; Fig 3,4,5).
>
> We would also like to emphasize that variations we use are not rooted in prompt engineering or other arbitrary problem modifications that put model into vastly different operation modes - we make use of variations in a simple problem template that are natural part of problem instantiation (Fig. A https://hackmd.io/_uploads/HJULH2IG1l.png, Suppl. Tab. 2) and do not change problem structure, its difficulty or aim to heavily alter processing mode in general (like done in prompt engineering). Our approach makes it thus possible to draw from collected evidence about model sensitivity conclusions about generalization deficits, contrary to various previous anecdotal examples where such systematic measurements were not performed, one or only few models were used, control experiments were not executed, problem setting was ambiguous or prone to low level issues like tokenization, findings were hard to reproduce and thus no clear conclusions about conditions, extent and nature of model failures were possible.
>
> We think the work done in [1] (https://arxiv.org/abs/2302.08399) is exactly of this "anecdotal" kind - it takes single model (GPT-3.5), goes through selected cases of failure without executing many trials under well controlled conditions  to gather statistics on average failure rates related to various conditions, so that it is indeed impossible to conclude whether failures were due to specific problem type, input presentation, prompt type etc, or indeed due to deficits in core model function, such that those examples remain in anecdotal realm. One of motivations behind our work was to depart from anecdotal failure reports and do systematic evaluation study showing how such failures manifest in reproducible manner (all data available in the repo https://anonymous.4open.science/r/AITW_anonymous-69A6/README.md) and whether they indeed relate to core generalization deficits in al SOTA LLMs, which we hope to have succeeded in.

---

> > ### Author Response · Authors · 2024-11-21
> > **Rebuttal Continuation, 3**
> >
> > > The authors do not give a reasonable rationale behind why LLMs fail on the AIW problem.
> >
> >
> > Our work shows that all SOTA LLMs suffer from lack of robustness pointing to severe generalization deficits. Why generalization deficits exist in current SOTA LLMs is a tough question to answer, we hope to have made one step in direction to clarify it by developing a method and with AIW problem a tool that actually allows to measure existence of such deficits, contrary to existing standardized benchmarks (Fig 7; Suppl Fig 16,17,18,19).  Our control experiments show further how to rule out other low level issues as potential causes of observed breakdowns (Fig E https://hackmd.io/_uploads/ByCpjM9M1x.png; Fig 3,4,5), which helps to narrow the search in the future work.
> >
> > > Interestingly, it points out that LLMs fail on the AIW Light Arithmetic Total Girls but that is another anecdotal showcase of failure that does not tell us much about why LLMs fail on such simple problems.
> >
> > It seems to be a misunderstanding as to what AIW Light control experiments are revealing. LLMs do not fail, but successfully solve AIW Light control problems across variations, achieving high correct response rates across variations without strong performance fluctuations, in contrast to AIW original (Fig E https://hackmd.io/_uploads/ByCpjM9M1x.png). As AIW Light problems are constructed to test operations that are also required to solve AIW original, by leaving problem template unmodified and altering the question for checking specific operations, successful handling of AIW Light proves that tested models are able to handle well all "low-level" operation and specific knowledge required for AIW, like parsing natural language, numbers, tokenization in general, grasping basic family relations, binding attributes to entities (eg female attribute via she pronomen to Alice) or performing arithmetic operations. Importantly, fluctuations on AIW Light variations disappear almost completely (Fig 3,4,5). Specifically, success in solving AIW Light Arithmetic Total Girls (Fig. 5) proves that models do not have any issue with binding of female attribute to Alice via “she”, handling family structure and selecting and executing the correct arithmetic operation to count total girls number. Thus, contrary to anecdotal examples, these experiments allow to rule out low level trivial issues behind strong fluctuations and performance breakdown observed on AIW (Fig. 2,6), providing further evidence in favor of hypothesis stating generic generalization deficits behind the fluctuations.
> >
> > > … a sufficient (but not necessary) condition to solve the problem is being able to perform 2-hops reasoning and counting (from one of the brothers to all the sisters). LLMs seem to lack such capability.
> >
> > In similar line to preceding discussion, we think that 1) strong performance fluctuations observed on AIW variations (Fig B https://hackmd.io/_uploads/rJPFH38Gyl.png; Fig 2,6) effectively rule out that the breakdown is due to a particular same type of operation failure, as it would then manifest equally across variations where only differences are the differently instantiated numbers N,M, which means that if “2-hops reasoning and counting” would not work, then it would not be possible to observe at the same time high performance on one variation and breakdown on other, problem structure being the same - all variations would should equal breakdown, which we do not observe. 2) AIW Light control experiments rule out issues with failure of specific operations (Fig. E https://hackmd.io/_uploads/ByCpjM9M1x.png). For instance, models handle well AIW Light Family problem (Sec 2.1, Sec. 3.1.1, Fig. 4) which also requires to determine the entity of Alice’s sister and then to obtain the number of the brothers - a 2 hops operation. Again, evidence suggests that fluctuations on AIW are not due to failure of a specific operation required to handle AIW problem, but rather because of inability to robustly infer problem structure due to generic generalization deficits.
> >
> > > Question 1. What happens when N and M grow larger than 7, and why do they decide to set that as the upper bound on such variables?
> >
> > AIW problem was designed to be a simple common sense math problem where problem instantiations have realistic number values, thus N,M was chosen to correspond to values that are probable in a realistic family frame. We executed a control experiment with higher, exaggerated numbers (adding offset 60 to numbers used in AIW original), without observing unexpected qualitative differences (Fig. H https://hackmd.io/_uploads/H1sLdj2Mye.png). We see again rather low correct response rates and strong fluctuations across variations, with slightly lower correct response rates which might hint on further generalization deficits if numbers are outside of the expected problem specific range.

---

> > > ### Author Response · Authors · 2024-11-21
> > > **Rebuttal Continuation, 4**
> > >
> > > > Question 2. Have the authors tried with floating and/or negative values for N and M? If a model still replies with the consistent (yet wrong) reasoning, that is a strong hint a model does not understand the task under consideration (i.e., it cannot connect numerical and graphical reasoning with family relationships).
> > >
> > > We did not try to experiment with unrealistic numbers. Instead we conducted AIW Light control experiments with natural realistic formulations to check whether low level operations and additional family specific knowledge can be successfully handled by the tested models, confirming they do not have issues to perform arithmetic operations within family relationship structure (Fig E https://hackmd.io/_uploads/ByCpjM9M1x.png; Fig 3,4,5)
> > >
> > >
> > > > Question 3. Have the authors tried to ask the model to generate a graphical representation of Alice’s family and then solve the task?
> > >
> > > We indeed were experimenting with prompting models to generate various intermediate representations, including graphical representations and SQL code (Sec. 3.2, p.9, Supp.l Sec G, Suppl. Fig. 32). We observed similar behavior, where poor performing models cannot generate such representations and stronger performing models cannot generate correct representations robustly across variations. Thus, asking to work with explicit intermediate representations does not seem to improve the situation. It has to be also noted that hinting models to attempt a certain type of representation construction helpful for the problem can be considered as an implicit hint how to proceed with problem solution and cannot be thus compared directly to problem formulation that does not reveal such implicit hints.
> > >
> > > > Question 4. Why do the authors focus on a single example and not on a consistent range of variations of similar problems?
> > >
> > > We do provide a consistent range of variations, being the instantiations of numbers N,M in the AIW problem template (Fig. A https://hackmd.io/_uploads/HJULH2IG1l.png; Suppl. Tab. 2). Crucially, as we carefully control the source and type of variations that do not change problem structure and its difficulty, being natural part of problem formulation, it allows us to make conclusions from observed model sensitivity to those variations about existing generalization deficits. As discussed previously, AIW problem provides us thus the minimal setting delivering sufficient evidence (together with AIW Light control experiments) to falsify hypothesis of strong zero shot generalization claimed by SOTA LLMs, so that further problems are not required. We have also conducted further experiments with similar problems to have further confirmation of the observed behavior, replacing for instance brothers and sisters with male/female friends or introducing Alice and Bob as two entities in the family structure, while keeping the same procedure for creating variations by instantiating numbers N,M. We have observed the same pattern in those experiments as for AIW original, further confirming the original findings (Fig. I https://hackmd.io/_uploads/BJ1nqj3MJx.png)

---

> > ### Comment · Reviewer_3mYn · 2024-11-26
> > **Response 2**
> >
> > I still hold the opinion, shared by most reviewers in this batch, that one example is not sufficient to say much about a model's capabilities. It would be interesting to study what happens in a model's internal (e.g., Llama) when it fails on the problem. Especially looking the attention scores (we know there are severe limitations to this mechanistic approach, but it is something) when a model generates the numerical solution.
> >
> > Again, I believe the article is quite well written, but what lacks here is a consistent analysis of a few things, including whether a model failed because it cannot understand the notions of brother an sister, or it makes an assumption on Alice not being one of the sisters, or again the model cannot perform 2-hops reasoning in this specific case (while in many others, it can, see for example [2]).
> >
> > [2] Graph-enhanced Large Language Models in Asynchronous Plan Reasoning, ICML.

---

> > > ### Comment · Reviewer_3mYn · 2024-11-26
> > > **Response to this discussion**
> > >
> > > While my biggest concerns still hold (one example is not sufficient to show any lack of reasoning, there is no consistent analysis of why these models fail, not even for open-source LLMs), I believe the authors replied to many of my concerns and I thus increase the score to 5.

---

> ### Author Response · Authors · 2024-11-26
>
> We thank the reviewer for taking the time and involving intensively into discussion. We appreciate raising the score following our initial rebuttal. In following, we would like to iterate on points raised:
>
> > I still hold the opinion, shared by most reviewers in this batch, that one example is not sufficient to say much about a model's capabilities … my biggest concerns still hold (one example is not sufficient to show any lack of reasoning …)
>
> We think that there is still ongoing misunderstanding among reviewers when pointing out we deal with a single example only of what the study’s main goal is. Main goal of the study is to provide convincing falsification of the hypothesis that posits strong zero-shot generalization and robust basic reasoning exhibited by current SOTA LLMs (let’s call it “strong function hypothesis”). To falsify the hypothesis, following scientific method we construct a minimal experimental setup - as simple as possible to perform this duty - that gives us sufficient evidence to reject the strong function hypothesis.
>
> Thus, we do not aim to test all the various abilities of the models. We aim to test whether the claim of strong function holds, to falsify which it is sufficient to present one clear contradiction backed up by proper systematic measurements. AIW problem and measurement procedure using its variations (see Fig. K for overview https://hackmd.io/_uploads/rkPA9H-myg.png) have two crucial properties for providing this contradiction and thus falsification:
>
> 1. It is a very simple, unambiguous, common sense math problem, arguably simple enough to be solved by elementary school children. Any system that claims to have robust problem solving capabilities - and current advanced SOTA LLMs (e.g GPT-4, Claude 3 Opus) maintain quite a strong claim to be able to deal with math problems of graduate student level ! - should be able to solve such a simple problem with correct response rates close to 100% if measured across many trials.
>
> 2. It uses natural variations around fixed simple problem template to systematically probe models’ sensitivity to problem perturbations that keep problem structure and its difficulty unchanged. Controlling for source of variations in this way, it is possible to make conclusions about models’ generalization from the observed robustness to those variations in a simple scenario. Again, any system capable of robust generalization should be able to handle the problem independent of such “natural” variations, showing strong performance  on each variation as the problem structure is unchanged and so problem solving should be not affected. We would like to emphasize again that variations belong naturally to the problem, being mere number instantiations in the problem template (Fig. K https://hackmd.io/_uploads/rkPA9H-myg.png). Variations are not adversarial variations tweaked specifically to break the models, or prompt engineering like variations to tune the model performance. On the contrary, variations should keep the problem and model operation mode unchanged.
>
> We obtained following evidence:
>
> 1. Overall low average correct response rates, averaged across many trials (> 30 trials for each combination of AIW variations 1-4 and 3 various prompt types, STANDARD, THINKING, RESTRICTED) (Fig. K; paper Fig. 1)
>
> 2. Strong fluctuations across variations (Fig. K; paper Fig. 2,6) - despite each variation being instance of the very same problem, with the only difference between variations being the instantiated numbers, performance can vary wildly from high to low, also for most advanced tested models like GPT-4 and Claude 3 Opus.
>
> We thus think this clear evidence from experiments using single problem type is indeed sufficient to debunk strong function claim and to point to generic lack of reasoning - reasoning about problem structure that would allow robust problem structure inference and would enable robust generalization despite problem irrelevant variations. As pointed out in previous discussions, it is otherwise impossible to explain strong performance fluctuations observed across variations - assuming any operation or set of operations necessary to solve AIW being broken, performance should have been affected equally across variations, as those pose the very same problem.  Via control experiments, we also provide additional evidence to rule out that the observed breakdown is due to failures in executing low level operations specific to AIW (Fig. E https://hackmd.io/_uploads/ByCpjM9M1x.png)

---

### Official Review · Reviewer_7c6D · 2024-11-03

**Soundness:** 3
**Presentation:** 3
**Contribution:** 2
**Rating:** 6
**Confidence:** 4

**Summary:**

The paper demonstrates that LLMs struggle with generalisation on a simple problem. Specifically the authors construct a question and various variations thereof of a `simple' family relationship question. (i.e. in short the question is: Alice has N brother and M sisters. How many sisters does Alice's brother have?) The paper demonstrates that small variations of this question break state-of-the-art LLMs, even across different prompting techniques.

**Strengths:**

The strengths of the paper:
1. Good number of experiments specific to the question demonstrated by the authors
2. Careful analysis across a wide variety of models.
3. Interesting finding that breaks models.

**Weaknesses:**

The weakness of the paper:
1. The study while interesting and definitely highlights a problems with modern LLMs is quite limited by the actual test set (basically being based on a single question and variations thereof).
2. The study is quite limited into the actual limitations of the model.

Concretely, although many models were run on this small dataset (and it is understandable that so many models can only be run on smaller datasets [reasonably]), the contribution is quite limited regardless. **Finding specific phrasings that break a model is very common** to most problems and to most people that have done prompt engineering.

Furthermore, the actual analysis while removes high-level doubts in the approach (such as the female boost, or the control question) do not provide deeper insights into what might be going on. Very interesting work in this regard would be "Physics of Language Models", which provides excellent analysis of how model perform and why they generalise poorly. https://physics.allen-zhu.com/

Generally, your work is very interesting and should be pursued further. Well done, however, in terms of research contribution it requires more interesting datasets (than single examples that work poorly and then others that work well, as mentioned earlier this is very common for most tasks). Also, your analysis should be much more detailed in terms of how and why models perform poorly or well on these tasks. (Again, the Physics of Language Models is an amazing work (not ours, unfortunately ;)).

**Questions:**

Some questions that could help you with you research:
1. What specifically do you think can discovered about LLMs using your research (going beyond that LLMs perform poorly on specific examples, but perform better on others?)
2. How could you construct a dataset that measures that specific quality?
3. How could you then propose methods to overcome a fundamental problem that you have identified?

---

> ### Author Response · Authors · 2024-11-20
>
> We thank the reviewer for taking the time to deal with our work. We appreciate feedback on its positive aspects emphasizing interesting finding of model breakdown, extensive experiments and analysis involving a large set of tested models. We also appreciate further points raised, which we address in following:
>
> > The study while interesting and definitely highlights a problem  with modern LLMs is quite limited by the actual test set (basically being based on a single question and variations thereof).
>
> Main goal of our paper is to convincingly falsify the hypothesis stating current SOTA LLMs possess robust generalization and reasoning, which is backed up by current standardized benchmarks. Following the scientific method to obtain such falsification, we search for a “minimal” experiment/problem, as simple as possible to provide sufficient evidence for hypothesis falsification. AIW problem, despite being a specific problem, together with the developed technique to measure models’ sensitivity to variations introduced into the problem template without changing problem structure or difficulty, satisfies this requirement (Fig. A https://hackmd.io/_uploads/HJULH2IG1l.png, Suppl. Tab. 2). For us, it is surprising to observe i) low correct response rates (Fig. 1, Suppl. Fig. 8) and even more importantly ii) strong performance fluctuations (Fig 2, 6) across AIW variations that do nothing but only change numbers N, M in such a simple problem setting. Models claiming robust zero-shot generalization and basic reasoning should have been able to solve AIW variations without fluctuations with correct response rate close to 100%. Together with control experiments that rule out failures in low-level issues like natural language/numbers parsing or handling specific knowledge about family structure, the evidence we obtain by using AIW problem setting is thus sufficient to disprove strong function hypothesis and also reveal generic generalization deficits beyond specific AIW problem scenario that are not detected by standardized benchmarks (Fig 7, Suppl. Fig. 16,17,18,19).
>
> > Finding specific phrasings that break a model is very common to most problems and to most people that have done prompt engineering. … single examples that work poorly and then others that work well … very common for most tasks
>
> In our work, we do not deal with finding specific phrasings or formulations that break models. On the contrary, we intentionally design AIW problem formulation to be simple, short and unambiguous to focus on the effect of variations built into the template, which are as well constructed to be simple, natural variations (Fig. A, Suppl. Tab. 2). We think it is the most intriguing part of our work that strong fluctuations observed in all tested SOTA LLM are not caused by prompt engineering (which is indeed well known), but appear across problem variations that change neither problem structure nor its difficulty, observed behavior being consistent across various prompt types (Fig. B https://hackmd.io/_uploads/rJPFH38Gyl.png; paper Fig 2,6). The examples that work poorly and examples that work well differ in nothing else than instantiated numbers in the template, which allows conclusion about severe lack of robustness and generic generalization deficits. To our knowledge, ours is the first study that systematically quantifies such lack of robustness in a simple scenario, seeing all SOTA LLMs exhibiting this phenomena and using control experiments to rule out other causes. This approach makes it possible to draw conclusions about generalization deficits, contrary to various previous anecdotal examples where such systematic measurements were not performed, findings were hardly reproducible and no clear conclusions about extent and nature of model failures were possible.
>
> > … analysis should be much more detailed in terms of how and why models perform poorly or well on these tasks … "Physics of Language Models" [line of work] provides excellent analysis of how model perform and why they generalise poorly.
>
> While we agree that looking into various types of model breakdowns and their causes is very fruitful in general, we would like to note that science is an iterative process. We hope to have made with our work one important iteration, discovering a minimalistic AIW problem and a technique to measure performance fluctuations across variations in the problem template that do not change problem structure or difficulty that allowed us to obtain clear evidence for severe generalization deficits in all SOTA LLMs  (https://hackmd.io/_uploads/rJPFH38Gyl.png; paper Fig 2,6), contrary to previous claims. Follow up work can be to take this newly obtained evidence and conduct systematic studies elucidating origins of the fluctuations (where it helps that our control experiments could rule out low level issues; Fig. E https://hackmd.io/_uploads/ByCpjM9M1x.png; Fig. 3,4,5), which in turn may give hints what is required to get robust generalization.

---

> > ### Author Response · Authors · 2024-11-20
> > **Rebuttal Continuation, 2**
> >
> > (Continuation)
> >
> > We agree that recent works like Physics of Language Models (which consist of many parts, each dozen pages large) can provide helpful hints how to conduct search for the failure origins, while  AIW like problems can serve as tools to detect failures and measure the progress of interventions.
> >
> > > Questions:  1, What specifically do you think can be discovered about LLMs using your research (going beyond that LLMs perform poorly on specific examples, but perform better on others?)
> >
> > Our research gives a measurement tool to test for generalization failures. It gives also the possibility to compare models looking at the distribution of fluctuations (Fig. 2, Fig 6), creating ranking not reflected in standardized benchmarks. Currently, our research refutes the claim of advanced SOTA LLMs to possess robust strong generalization by showing clear lack of robustness on simple AIW problem, where models with robust zero-shot generalization should exhibit only small fluctuations and close to 100% correct response rates across variations. Our research allows in general to refute overblown claims about model capabilities that rely on standardized benchmarks and other benchmarks that claim to measure advanced functions.
> >
> > Eg, in case of NuminaMath-7B that was ranked 1st at the recent AIMO competition, solving 29/50 private set problems of olympiad math level, the claim was widely put forward that the model is capable of solving high school olympiad math problems. AIW has average elementary school level and does not require any advanced math knowledge. We tested NuminaMath-7B on AIW and observed a strong collapse of this model on AIW problem, with correct response rates close to 0 across AIW variations 1-4. Using AIW Light control problems, we can also see that NuminaMath-7B can handle all the low level operations and knowledge required to deal with family structure, ruling out that those are the issues. Using the AIW problem setting, we thus can contradict the strong claim of being capable to deal with olympiad level high school math (Fig F, https://hackmd.io/_uploads/SybG2hqz1x.png).
> >
> > We would like to emphasize again that examples in our study that lead to poor or better performance of tested models are constructed in certain way to probe model sensitivity to variations that keep problem structure and its difficulty unchanged and thus can be used to measure generalization - in contrast to other studies that has entirely different aim, eg performance optimization by prompt tuning.
> >
> > > 2. How could you construct a dataset that measures that specific quality?
> >
> > In the simplest form, a benchmark for measuring generalization can be constructed relying on already existing AIW problem variations. Using a robustness score computed over the shape of measured fluctuations distribution, models can be ranked in their generalization capability. A larger and more diverse dataset can be created by procedurally generating further AIW versions, where further variations can be introduced, e.g. varying names of entities, relational structure of the problem, and so on. The same evaluation would apply  - given common template, models can be evaluated on all possible problem instances, measuring distribution of fluctuations. This can be done for problem templates with increasing difficulty level, such that generalization breakdown can be measured dependent also on problem difficulty.
> >
> > > 3. How could you then propose methods to overcome a fundamental problem that you have identified?
> >
> > There are different directions to improve zero-shot generalization and model robustness that currently are unsatisfactory as evident from measured AIW fluctuations. One direction would be to test whether inference time interventions like various self-verification approaches might improve generalization especially in larger scale models. Another direction would be to fine-tune models on procedurally generated AIW problem-like variations and test on a held out AIW test version set whether generalization improves. Yet another, much more compute expensive direction would be to improve the core model generalization capability by modifying pre-training, eg changing dataset mixture to contain synthetic data generated from various problem templates and their variations.

---

> ### Comment · Reviewer_7c6D · 2024-11-26
>
> Thank you for providing a detailed answer to our review.
>
> (We apologise for answering late - as the current workload is quite high).
>
> We raise the score to 5, however, we still have concerns and believe a revision is needed:
>
> ---
> 1. First study:
>
> > To our knowledge, ours is the first study that systematically quantifies such lack of robustness in a simple scenario, seeing all SOTA LLMs exhibiting this phenomena and using control experiments to rule out other causes.
>
> There are works such "length generalisation of LLMs" that study generalisation of input length. And more recently also "What Makes Math Word Problems Challenging for LLMs?" that studies math problems and variations based on input. (Among other such findings).
>
> ---
> 2. Accuracy in rebuttal
>
> > This approach makes it possible to draw conclusions about generalization deficits, contrary to various previous anecdotal examples where such systematic measurements were not performed, findings were hardly reproducible and no clear conclusions about extent and nature of model failures were possible.
>
> It would be good to be more precise when referring to previous work. (Which work was "hardly reproducible"?)
>
> ---
> 3. Size of the dataset
>
> > In the simplest form, a benchmark for measuring generalization can be constructed relying on already existing AIW problem variations. Using a robustness score computed over the shape of measured fluctuations distribution, models can be ranked in their generalization capability. A larger and more diverse dataset can be created by procedurally generating further AIW versions, where further variations can be introduced, e.g. varying names of entities, relational structure of the problem, and so on.
>
> We believe that this would be required for this work to be a meaningful contribution to our field. At this stage we think that it is too narrow of a study and dataset and a bigger more useful dataset would be interesting - especially since in the next update of LLMs this benchmarks might be already obsolete (due to its limited size and breadth).

---

> > ### Author Response · Authors · 2024-11-27
> >
> > We thank the reviewer for getting involved in the discussion and appreciate the expression to raise the score. In following, we would like to address the points mentioned:
> >
> > > First study : … works such as "length generalisation of LLMs” [1] … "What Makes Math Word Problems Challenging for LLMs?" [2]
> >
> > We think that our study is the first to provide convincing falsification of the hypothesis that posits strong zero-shot generalization and robust basic reasoning exhibited by current SOTA LLMs - including the advanced ones like GPT-4 and Claude-3 Opus. We do this by testing model generalization using AIW problem and its variations to measure model sensitivity to perturbations that keep both problem structure AND its difficulty unchanged (see for overview Fig. K https://hackmd.io/_uploads/rkPA9H-myg.png). AIW problem is formulated in simple natural language common sense manner, and for testing, we use large variety of recent SOTA LLMs. In contrast to this, [1] uses synthetic tasks where variations change both problem structure and difficulty (e,g., when changing problem length), which introduces confounds and makes it hard to draw clear conclusions about generalization deficits. Further, tests are done only on one single model (LamDA dated back to 2022), such that it is not clear whether observed effects are specific to that particular model and training procedure only (LamDA family also never put forward claim to be robust problem solver, in contrast to current SOTA LLMs). [2] does not study variations or generalization breakdown at all - it has entirely different focus, taking GSM8K problems, collecting responses to those problems by 4 different models and training classifiers to predict whether an LLM succeeds or fails given a problem, to identify what makes GSM8K problems easier or harder to solve. This does not allow any conclusions about model generalization, in contrast to focused falsification we perform.
> >
> > > … more precise when referring to previous work …  (Which work was "hardly reproducible"?)
> >
> > Here we were referring to various anecdotal examples that were widespread in the ML community but never received proper systematic treatment and evaluation.  For instance, the reports on the problems like “What is larger, 9.11 or 9.9” or “Count letters “r” in the word “Strawberry” ” were suffering from lack of systematic evaluation, reporting single cases of failure without executing multiple trials to estimating robust statistics (eg failure rates) and without controlling for various conditions, eg prompt types. “Hardly reproducible” refers here both to insufficient rigor behind the experimental procedure that independent parties can reproduce under same conditions and also lack of repositories containing raw data and code to execute procedures to obtain the same results. We provide the collected raw data and exact prompts used to execute experiments in anonymous repo for the review (https://anonymous.4open.science/r/AITW_anonymous-69A6/README.md), and public repo with all source code necessary to reproduce the experiments will be available after the review procedure.
> >
> > >  … a benchmark for measuring generalization can be constructed relying on already existing AIW problem variations … we believe that this would be required for this work to be a meaningful contribution to our field. At this stage we think that it is too narrow of a study and dataset and a bigger more useful dataset would be interesting - especially since in the next update of LLMs this benchmarks might be already obsolete (due to its limited size and breadth).
> >
> > We very much agree that a benchmark measuring generalization constructed from the insights won in our study is the way to go and we work on follow up aiming for such a benchmark. We would like to note however that science is an incremental process. Motivation for such a benchmark has to come from work that first properly clarifies that current SOTA LLMs generalization is not what it seems as reported by current standardized benchmarks, so it also becomes clear that current benchmarks are not good enough for measuring generalization and need to be replaced. Our work reveals both model and benchmark flaws (Fig. 7, Suppl. Fig. 16,17,18,19), sending a warning signal to the community to be careful in trusting strong function claims based on current benchmarks and sketching a potential procedure that can provide a better measurement tool. We hope this can be acknowledged as an important step worth publishing. We would also like to stress that AIW problem and its variations are not to be seen as a dataset - this is a focused minimal experimental setup and measurement procedure sufficient to induce model generalization breakdown and provide necessary evidence to falsify the hypothesis of the robust generalization in current SOTA LLMs.

---

> > > ### Author Response · Authors · 2024-11-27
> > >
> > > We would also like to note that following reviewers’ requests, we conducted further experiments with problems similar to AIW, to obtain further confirmation that same behavior can be observed on variety of problems, further substantiating the value of the measurement procedure assessing models’ generalization, beyond AIW original. We looked into AIW versions with modified problem templates, where we either introduce Alice and Bob as two entities in the problem structure to deal with, or we replace brothers and sisters entities with male/female friends, abandoning family specific frame.
> > >
> > > Using same experimental procedure to create variations of these problem versions, we observe the same pattern as for the AIW original, especially the strong fluctuations across variations, confirming the existence of the same generalization deficits using further problem examples (Fig. I https://hackmd.io/_uploads/BJ1nqj3MJx.png) This paves the road for benchmark construction in follow up work.

---

> > > > ### Comment · Reviewer_7c6D · 2024-11-28
> > > >
> > > > Thank you for running additional experiments. We have adjusted the score once more to 6. We hope that a slightly updated version of the code / data will be released that will allows for "procedural" augmentation of the data for better validation and use for the future.

---

### Author Response · Authors · 2024-11-25
**Rebuttal summary**

We would like to thank all reviewers for the feedback. We provide a summary of the rebuttal, containing points we think are helpful to emphasize for discussion.

# Testing generalization and falsifying SOTA LLMs strong function hypothesis

While reviewers stressed the clarity of the idea and extensive experiments in our work, we would like to clarify a potential misconception wrt. to the work that we think is repetitively echoing through the feedback. This misconception concerns the purpose of the AIW problem constructed in our work for testing generalization, relation of this problem to previous anecdotal attempts to induce model breakdowns of various kinds and the role of systematically controlled variations in properly measuring model sensitivity to problem structure preserving perturbations that reveals generalization deficits.

Main goal of our study was to test the claims of strong function attributed to current SOTA LLMs, specifically regarding strong zero-shot generalization and robust basic reasoning. Such claims are mainly based on standardized benchmarks (MMLU, HellaSwag, GSM8k, MATH, etc) where current SOTA LLMs obtain high scores. We were curious whether these claims can be put to test using much simpler, more minimalistic problems, as problems used in standardized benchmarks have often formulations overloaded with various additional context and knowledge. We aimed for a problem as simple as possible though still useful for measuring generalization of SOTA LLMs.

AIW problem, which has a short, unambiguous natural language formulation, was the outcome of this effort (Fig. A https://hackmd.io/_uploads/HJULH2IG1l.png;  Suppl. Tab. 2). Despite its simplicity, we observed surprisingly low correct response rates averaged across various conditions for all SOTA LLMs, including the most advanced ones like GPT-4/4o or Claude 3 Opus (Fig. 1,  Suppl. Fig. 8). The scores were substantially lower than the ones SOTA LLMs obtain on benchmarks with seemingly more complex problems, already posing an intriguing discrepancy to standardized benchmark measurements.

More importantly however, introducing controlled natural variations into AIW problem template by instantiating numbers N,M involved in its formulation, we created a procedure that measures model sensitivity to perturbations that preserve problem structure and difficulty. This allowed us to probe models’ generalization - should it be intact, model would be capable of handling the problem equally well across variations. This is in contrast to standardized benchmarks, where problems are static and there is no default way to measure model sensitivity to problem variations. Using this procedure, we observed strong fluctuations across problem variations (Overview: Fig. K https://hackmd.io/_uploads/rkPA9H-myg.png; Fig. 2, 6). This is surprising not only because the problem is simple, but also because AIW variations do not change problem structure or its difficulty, being just instantiations of different numbers natural to the problem. Puzzling is why there should be any difference in performance across AIW variations at all. Given the variations should be actually irrelevant for problem handling, expected would be either struggling equally to solve any of the variations or handling them equally successful - assuming generalization is intact. Thus, observing such strong fluctuations in all models that manage to get significant non-zero average correct response rates, we can conclude that models are not able to properly infer the same underlying problem structure which is behind each AIW variation, which, given simplicity of the posed problem, points to severe generalization deficits.

We think it can be helpful to imagine how humans would perform on such AIW variations, to stress what the observed breakdown of model robustness and overall low average correct response rates mean. Having a test with average human probands confronted with the AIW variations over many trials (we do > 30 trials per variation to compute correct response rates)

> Variation 1: Alice has 3 brothers and she also has 6 sisters. [Correct answer: 7 ] \
> Variation 2: Alice has 2 sisters and she also has 4 brothers. [Correct answer: 3 ] \
> Variation 3: Alice has 4 sisters and she also has 1 brother. [Correct answer: 5 ] \
> Variation 4: Alice has 4 brothers and she also has 1 sister. [Correct answer: 2 ] \
> \
> How many sisters does Alice’s brother have?

we think it would be highly shocking if obtained statistics would reveal performance across variations varying wildly between very high and very low rates, the way we observe it for SOTA LLMs. We expect humans to handle any variation equally well, as we expect generalization and basic reasoning to be intact. It therefore should be equally shocking to observe SOTA LLMs breakdown in such a simple scenario, given the claims of robust graduate level problem solving put forward for models of GPT-4 and Claude 3 Opus class.

---

> ### Author Response · Authors · 2024-11-25
> **Rebuttal summary, continuation 1**
>
> # Controlling for low-level issues, contrast to reports of anecdotal model failures
>
> We also would like to stress that one of the motivations behind our work was to depart from anecdotal reports on various failures of SOTA LLMs done in the past. To gather evidence for model breakdown and postulate existing generalization deficit, we performed systematic evaluation study across a large variety of SOTA LLMs with robust statistics accumulated over multiple trials (> 30 trials for each AIW variation 1-4 and each of 3 prompt types) under various conditions while controlling accurately for source of variation (Overview: Fig. K https://hackmd.io/_uploads/rkPA9H-myg.png). This is in contrast to previous observations attempting to showcase model breakdown on simple problems. For instance, the reports on the problems like “What is larger, 9.11 or 9.9” or “Count letters “r” in the word “Strawberry” ” were suffering from lack of systematic evaluation, reporting single cases of failure without executing multiple trials and estimating failure rates and without controlling for various conditions, eg prompt types. Often, such problems suffer from ambiguous character (eg, adding “real numbers” may disambiguate the question above avoiding interpretation as dates) that makes it unclear whether model failure is due to ill posed problem specification or indeed due to core function deficit. In contrast, AIW problem was made intentionally to be simple to handle, without any ambiguities or quirks in the formulation.
>
> Another issue common to anecdotal reports are confounds in form of low level issues, eg inability to parse the input properly due to tokenization issues, like it might be the case for “Strawberry”, or in general, inability to access and deal with highly specific problem knowledge, which might cause model function breakdown although actual generalization and basic reasoning might still be intact. In our study, we made an effort to rule out such low level issues causing observed model breakdown on AIW by conducting control experiments using AIW Light problems. AIW Light problems are constructed to test operations that are also required to solve AIW original, by keeping the problem template unmodified and altering the question for checking specific operations (Fig. E https://hackmd.io/_uploads/ByCpjM9M1x.png, Sec 2.1, Sec 3.1.1). By observing that models handle AIW Light problems successfully, we prove that tested models are able to handle well all "low-level" operation and specific knowledge required for AIW, like parsing natural language, numbers, tokenization in general, grasping basic family relations, binding attributes to entities (eg female attribute via she pronomen to Alice) or performing arithmetic operations. Importantly, fluctuations on AIW Light variations disappear almost completely (Fig. E; paper Fig. 3,4,5).
>
> Thus, contrary to anecdotal examples, our control experiments allow us to rule out low level trivial issues or other quirks behind strong fluctuations and performance breakdown observed on AIW (Fig. 1,2,6), strengthening evidence in favor of the hypothesis stating generic generalization deficits behind the fluctuations.
>
> # Additional experiments with evidence for generalization deficits on further problem examples
>
> Per reviewers request, we executed a number of additional experiments that we think further strengthen evidence pointing to severe generalization deficits in SOTA LLMs.
>
> One experiment concerns an AIW version where numbers for brothers and sisters are instantiated to be in an exaggerated range not realistic for a typical family scenario. In this AIW version, offset 60 is added to numbers in AIW original problem. We observe the same pattern as in AIW original (Fig. H https://hackmd.io/_uploads/H1sLdj2Mye.png) - low correct response rates and strong fluctuations across variations. We also see slightly lower correct response rates on average compared to AIW original. This might point to further generalization deficits becoming apparent when dealing with numbers outside the expected problem specific range, despite problem structure left unchanged.
>
> Further, to confirm that the same observations hold for other simple problems of related kind, we conducted experiments with AIW versions with modified problem templates that differ from AIW Original. For instance, we either introduce Alice and Bob as two entities in the problem structure to deal with, or we replace brothers and sisters entities with male/female friends, abandoning family specific frame. Using same experimental procedure to create variations of these problem versions, we observe the same pattern as for the AIW original, especially the strong fluctuations across variations, confirming the existence of the same generalization deficits for further problem examples (Fig. I https://hackmd.io/_uploads/BJ1nqj3MJx.png)

---

> > ### Author Response · Authors · 2024-11-25
> > **Rebuttal summary, continuation 2**
> >
> > # AIW as a tool to check for and warn against overblown claims of strong core function
> >
> > In our work, using the simple AIW problem that requires only elementary set and arithmetic operations and can be easily solved by adults and arguably even children, we observe a striking breakdown of SOTA LLMs performance when confronted with the AIW problem variations (Fig. B https://hackmd.io/_uploads/rJPFH38Gyl.png, Fig. K https://hackmd.io/_uploads/rkPA9H-myg.png; paper Fig. 1, 2,  Suppl. Tab. 2, 7). The breakdown is manifested in (i) overall low correct response rates (Fig. K https://hackmd.io/_uploads/rkPA9H-myg.png); paper Fig. 1) and (ii) strong performance fluctuation across natural variations of the same problem that do not affect problem structure or its difficulty, which reveals strong lack of robustness and hints at fundamental issues with the generalization capability of the models (Fig. K; paper Fig. 2,6). The observed breakdown is in dramatic contrast with claims about strong core functions of SOTA LLMs as backed up by standardized benchmarks, revealing benchmarks failure to properly measure core functions (Fig. 7, Suppl. Fig. 16,17,18,19).
> >
> > Relying on those benchmarks, it is still commonly-held position to attribute to SOTA LLMs advanced functions like robust zero-shot reasoning (e.g. [1], as one example of many), and in general to put high expectations of strong core functionality on released SOTA LLMs. Such claims extend beyond basic research artifacts and become pervasive in applied industry, where SOTA LLMs are advertised as robust problem solvers for various real world settings, explicitly emphasizing their value as robust reasoners, coders and math solvers, attesting "key business-critical capabilities" or suitability for "real-world enterprise use cases" (see announcements by Cohere on Command R-Plus [2], or by Mosaic on DBRX [3], as only few selected representative examples out of many - these models suffer collapse on simple AIW problem variations obtaining correct response rates close to 0 or 0 across variations (Fig. 1, Suppl. Fig. 8, Suppl. Tab. 7)., although our control experiments show that both models can handle all the operations necessary to solve the problem (Fig. 3,4,5) )
> >
> > Given the situation where standardized benchmarks fail to detect obvious failures on core function, AIW problem together with the measurement procedure using its variations provides a way to debunk claims of strong function by testing models that report high scores on standardized benchmarks.
> >
> > One particular scenario is debunking smaller scale models’ strong function claims. There is a persistent claim that overtraining smaller scale models leads to model performance that is almost on par with larger scale models, again relying on comparison with standardized benchmarks. On the contrary, we see in our experiments small scale models having much lower average correct response rates than larger scale ones, most of them collapsing close to 0 or having 0 rates across all variations. This discrepancy is not visible on standardized benchmarks. E.g., if testing GPT-4o-mini, the claimed close performance proximity to larger GPT-4o (backed up by standardized benchmarks, e.g. https://artificialanalysis.ai/models/gpt-4o-mini) falls apart (Fig. L https://hackmd.io/_uploads/BJ2M1-Mmke.png).
> >
> > Another example of debunking overblown claims is a case of NuminaMath-7B that was ranked 1st at the recent AIMO competition, solving 29/50 private set problems of olympiad math level. The claim was widely put forward that the model is capable of solving high school olympiad math problems. AIW has arguably average elementary school level and does not require any advanced math knowledge. We tested NuminaMath-7B on AIW and observed a strong collapse of this model on AIW problem, with correct response rates close to 0 across AIW variations 1-4 (Fig. F, https://hackmd.io/_uploads/SybG2hqz1x.png). Using AIW Light control problems, we can also see that NuminaMath-7B can handle all the low level operations and knowledge required to deal with family structure, ruling out that those are the issues. Using the AIW problem setting, we thus can contradict the strong claim of being capable to deal with olympiad level high school math
> >
> > Thus, AIW problem and its variations offer a measurement technique that can reveal lack of robustness and model weaknesses in generalization and core functions that remain undiscovered by current benchmarks. We think that our study can also serve as a vivid warning that many of the claims put forward for strong core functions of SOTA LLMs cannot be trusted, as they often rely on benchmarks that overlook clear function deficits, and AIW problem with its variations offers a tool for systematic, reproducible stress testing and debunking of such claims.
> >
> > References -> following final part

---

> > > ### Author Response · Authors · 2024-11-25
> > > **Rebuttal summary, references**
> > >
> > > ### References
> > >
> > > [1] Takeshi Kojima, Shixiang Shane Gu, Machel Reid, Yutaka Matsuo, and Yusuke Iwasawa. Large language models are zero-shot reasoners. Advances in neural information processing systems, 35:22199–22213, 2022.
> > >
> > > [2] Cohere. Introducing Command R+: A scalable LLM built for business. Apr 2024 https://cohere.com/blog/command-r-plus-microsoft-azure
> > >
> > > [3] Mosaic. Introducing DBRX: A new state-of-the-art open LLM. March 2024 https://www.databricks.com/blog/introducing-dbrx-new-state-art-open-llm

---

### Comment · Reviewer_7c6D · 2024-12-01
**Comment after Reviews**

Overall, this work is interesting and has good suggestions.

Our last comment as reviewer summarises our proposal how to improve this work. Specifically, extending the dataset to make it more robust to future changes (and we do not mean massive expansion, however, meaningful enough for this work to be valid next year).

---

### Author Response · Authors · 2024-12-03

Dear reviewers,

we would like to thank you all again for time and involvement in the discussions.

As the discussion time nears its end, we would like to encourage you to go through discussions, and should those be insightful and resolve some of the concerns, to consider reflecting this in the scores.

We would like to highlight again additional experiments performed in the rebuttal period upon reviewers' request:

- Experiments with modified problem templates that differ from AIW Original. For instance, we either introduce Alice and Bob as two entities in the problem structure to deal with, or we replace brothers and sisters entities with male/female friends, abandoning family specific frame. Using same experimental procedure to create variations of these problem versions, we observe the same pattern as for the AIW original, especially the strong fluctuations across variations, confirming the existence of the same generalization deficits for further problem examples (Fig. I https://hackmd.io/_uploads/BJ1nqj3MJx.png)

- An experiment concerning an AIW version where numbers for brothers and sisters are instantiated to be in an exaggerated range not realistic for a typical family scenario. In this AIW version, offset 60 is added to numbers in AIW original problem. We observe the same pattern as in AIW original (Fig. H https://hackmd.io/_uploads/H1sLdj2Mye.png) - low correct response rates and strong fluctuations across variations. We also see slightly lower correct response rates on average compared to AIW original. This might point to further generalization deficits becoming apparent when dealing with numbers outside the expected problem specific range, despite problem structure left unchanged.

- An illustrative example of a debunking procedure using the recent case of NuminaMath-7B that was ranked 1st at the recent AIMO competition, solving 29/50 private set problems of olympiad math level (https://huggingface.co/AI-MO/NuminaMath-7B-TIR). Based on that evals, the claim was widely put forward that the model is capable of solving high school olympiad math problems (https://www.aimodels.fyi/models/huggingFace/numinamath-7b-tir-ai-mo). AIW problem has average elementary school level and does not require any advanced math knowledge. We tested NuminaMath-7B on AIW and observed a strong collapse of this model on AIW problem, with correct response rates close to 0 across AIW variations 1-4. Using AIW Light control problems, we can also see that NuminaMath-7B can handle all the low level operations and knowledge required to deal with family structure, ruling out that those are the issues. Using the AIW problem setting, we thus can contradict the strong claim of being capable to robustly deal with math problems (Fig F, https://hackmd.io/_uploads/SybG2hqz1x.png). Especially, breakdown in such a simple setting rules out that model will be able to deal robustly with olympiad level math tasks, debunking the claim and raising further questions about AIMO benchmark procedure used to measure model performance.

For the collected data for this debunking experiment, see anonymous repo https://anonymous.4open.science/r/AITW_anonymous-69A6/collected_responses/raw_data_inspection/NuminaMath-7B_AIW_versions_problem_set_checked.json

We hope this together amounts to convincing evidence that experimental setup we describe in our work is useful for community as reproducible measurement method for systematically probing models' generalization using structure  preserving variations of simple problem setting. The method is also useful for falsification of the strong function hypothesis, to debunk overblown claims that rely on benchmarks which overlook such clear deficits as revealed by the AIW problem and its variations. We also hope that study provides impulses and clear roadmap to create novel benchmarks that use problem structure \& difficulty preserving perturbations to stress test model generalization systematically, so that such deficits do not remain hidden, also allowing for measurable progress in improving model generalization capability.

---

### Meta-Review · Area_Chair_Np52 · 2024-12-24

**Metareview:**

Summary: The paper investigates the generalization and reasoning deficits in state-of-the-art Large Language Models (LLMs) by employing a simple, common-sense reasoning problem called the "Alice in Wonderland" (AIW) problem. This problem reveals severe shortcomings in generalization and reasoning among advanced models like GPT-4 and Claude 3 Opus, even under variations of the problem that do not alter its difficulty or structure. The authors highlight the discrepancy between the claims of robust reasoning capabilities made by standardized benchmarks and the observed failures on simple tasks. Additionally, the paper introduces systematic variations and control tests to isolate the issue and provide evidence of fundamental deficits in LLMs’ ability to generalize across minimal perturbations.

Strengths:
- Important problem being studied and key shortcomings of LLMs being exposed
- Code and raw experimental data are made available for validation.

Weakness:
- Lukewarm response from all but one reviewer and the positive reviewer didn't champion the paper
- Limited Scope of Problem: The analysis is focused on a single problem type and its variations, restricting the generalizability of the findings.
- Lack of Deeper Diagnostic Insights: While generalization failures are observed, the paper does not delve into the underlying architectural or training-related causes of these deficits.
- Comparative Benchmarking: Insufficient discussion on why standardized benchmarks fail to detect these limitations and how alternative designs could address this.
- Broader Dataset: Reviewers suggested expanding the dataset to ensure robustness and broader applicability.
- Some formatting and citation issues in the manuscript

Decision: Given the lack of enthusiasm from the reviewers and limited scope, unfortunately, the paper can't be accepted in its current form and addressing all the concerns would warrant another round of reviewing.

**Additional Comments On Reviewer Discussion:**

We thank the authors and reviewers for engaging during the discussion phase towards improving the paper. Below are some of the highlights:

1. Single Problem Concern:
- Reviewers questioned whether conclusions could be drawn from one problem type
- Authors argued that their minimal test case was sufficient for falsifying strong generalization claims, supported by systematic variations and controls

2. Interpretation of Results:
- Debate about whether results showed complete lack of reasoning vs specific deficits
- Authors clarified they weren't claiming complete inability to reason, but rather inconsistent generalization

3. Technical Clarifications:
- Questions about model performance differences (e.g., Llama-2 vs Llama-3)
- Authors provided detailed analysis showing similar underlying issues despite surface differences

4. Additional Experiments:
- Authors conducted new experiments with modified problem templates and different entity types
Results reinforced original findings about generalization deficits
- The authors were highly responsive and provided detailed, evidence-based responses to all major concerns.

---

### Decision · Program_Chairs · 2025-01-22

Reject